developmental biology, microbiology

aposymbiotic, morphology, symbiosis, host–microbe interaction, bivalves, anatomy

**Authors for correspondence:**
Nicole Dubilier
e-mail: ndubilie@mpi-bremen.de
Nikolaus Leisch
e-mail: nleisch@mpi-bremen.de

# Coming together—symbiont acquisition and early development in deep-sea bathymodioline mussels

Maximilian Franke[1,3], Benedikt Geier[1], Jörg U. Hammel[2], Nicole Dubilier[1,3] and Nikolaus Leisch[1]

[1]Max Planck Institute for Marine Microbiology, Celsiusstrasse 1, 28359 Bremen, Germany
[2]Helmholtz-Zentrum Hereon, Institute of Materials Physics, Max-Planck-Strasse 1, 21502 Geesthacht, Germany
[3]MARUM—Zentrum für Marine Umweltwissenschaften, University of Bremen, Leobener Strasse 2, 28359 Bremen, Germany

MF, 0000-0002-1930-1036; BG, 0000-0002-2942-2624; JUH, 0000-0002-6744-6811; ND, 0000-0002-9394-825X; NL, 0000-0001-7375-3749

How and when symbionts are acquired by their animal hosts has a profound impact on the ecology and evolution of the symbiosis. Understanding symbiont acquisition is particularly challenging in deep-sea organisms because early life stages are so rarely found. Here, we collected early developmental stages of three deep-sea bathymodioline species from different habitats to identify when these acquire their symbionts and how their body plan adapts to a symbiotic lifestyle. These mussels gain their nutrition from chemosynthetic bacteria, allowing them to thrive at deep-sea vents and seeps worldwide. Correlative imaging analyses using synchrotron-radiation based microtomography together with light, fluorescence and electron microscopy revealed that the pediveliger larvae were aposymbiotic. Symbiont colonization began during metamorphosis from a planktonic to a benthic lifestyle, with the symbionts rapidly colonizing first the gills, the symbiotic organ of adults, followed by all other epithelia of their hosts. Once symbiont densities in plantigrades reached those of adults, the host's intestine changed from the looped anatomy typical for bivalves to a straightened form. Within the Mytilidae, this morphological change appears to be specific to *Bathymodiolus* and *Gigantidas*, and is probably linked to the decrease in the importance of filter feeding when these mussels switch to gaining their nutrition largely from their symbionts.

## 1. Introduction

Mutualistic interactions between hosts and their microbiota play a fundamental role in the ecology and evolution of animal phyla. By associating with microbial symbionts, animals benefit from the metabolic capabilities of their symbionts and gain fitness advantages that allow them thrive in habitats they could not live in on their own [1]. Prime examples for such symbioses are bathymodioline mussels, which occur worldwide at cold seeps, hot vents, and whale and wood falls in the deep sea. Mussels of the genera *Bathymodiolus* and *Gigantidas* house chemosynthetic bacteria in their gills, in cells called bacteriocytes [2]. In these nutritional symbioses, the bacteria use reduced compounds in the vent and seep fluids as an energy source for carbon fixation, which in turn provides nutrition to their hosts. Two types of symbionts dominate bathymodioline mussels, sulfur-oxidizing (SOX) symbionts, whose main source of energy are reduced sulfur compounds, and methane-oxidizing symbionts (MOX), which gain their energy from oxidizing methane [3].

The transmission of symbionts from one generation to the next plays a central role in the ecology and evolution of mutualistic associations [4]. Symbionts

can be transmitted vertically from parent to offspring, intimately tying them to the reproduction and development of their host [4], as known from vesicomyid clams [5]. Alternatively, in horizontal transmission, symbionts are recruited each generation anew from the environment, and the symbiotic partners are physically separate from each other before the symbiosis is established [4]. In many hosts that rely on horizontal transmission, the acquisition of symbionts triggers morphological and developmental changes in the host. These can range from tissue rearrangements, known to be symbiont-induced in the *Euprymna* squid–*Vibrio* symbiosis [6], to largescale modifications of host organs, typically through hypertrophy [7], or the development of a novel symbiont-housing organ, like the trophosome of the tubeworm *Riftia pachyptila* [8]. As most of these symbioses are uncultivable, it is still unclear if these developmental changes are mediated by the host, actively induced by the symbiont or a mix of both.

Although bathymodioline mussels have been studied for over 40 years, very little is known about how their symbionts are transmitted, at which developmental stage the symbionts colonize the mussels, and how symbiont acquisition affects the development and body plan of the mussels [9]. It is assumed the symbionts are transmitted horizontally, based on phylogenetic studies that showed a lack of cospeciation between hosts and symbionts, as well as morphological studies that found no evidence for symbionts in the mussels' reproductive tissues [10–14]. In bathymodiolins of the genus *Idas*, commonly found at organic falls as well as seeps, the larvae remained aposymbiotic until they settled and developed the dissoconch shell, indicating a heterotrophic lifestyle in the larval dispersal phase, and horizontal acquisition of symbionts during or after metamorphosis [7,15].

Given the high abundance of *Bathymodiolus* and *Gigantidas* at vents and seeps worldwide, it is surprising that definitive evidence of an aposymbiotic early life stage of these mussels is lacking. The earliest life stages described so far had undergone metamorphosis and were already colonized by symbionts, with a well-developed symbiotic habitus that was indistinguishable from adult mussels [16]. As the early life stages of aposymbiotic *Bathymodiolus* and *Gigantidas* mussels have not yet been described, fundamental questions in the acquisition of symbionts in these bathymodioline genera have remained unanswered, including at which developmental stage the mussels acquire their symbionts, whether the SOX and MOX symbionts colonize their hosts at the same time, and which developmental changes occur in the mussels at the onset of symbiont colonization.

In this study, we were fortuitous in discovering very early, aposymbiotic life stages of three bathymodioline species, two from hydrothermal vents on the Mid-Atlantic Ridge (MAR), *Bathymodiolus puteoserpentis* and *B. azoricus*, and one from cold seeps in the Gulf of Mexico, *Gigantidas childressi* (originally described as '*B*'. *childressi* [17]). We used a correlative imaging approach by combining synchrotron-radiation based micro-computed tomography (SRμCT), correlative light (LM) and transmission electron microscopy (TEM), and complemented it with fluorescence *in situ* hybridization (FISH) to analyse the early life stages and compare them to their shallow water relative *Mytilus edulis* [18]. This approach allowed an integrative analysis of symbiont colonization and its effects on the host body plan, from the whole animal down to single host and symbiont cells.

# 2. Methods

## (a) Sampling, fixation and sample preparation

Deep-sea mussels were collected from the sea floor with remotely operated vehicles (electronic supplementary material, table S1). *Mytilus edulis* were collected in the Baltic Sea at a site close to Kiel, Germany (electronic supplementary material, table S1). Samples were preserved for morphological, FISH and TEM analysis (electronic supplementary material, Methods a). All specimens were photographed and shell dimensions and shell margin limits were recorded as shown in electronic supplementary material, figure S1b–d (see also electronic supplementary material, Methods b, table S2). For histological analysis, all samples were post-fixed, embedded, serial sectioned and stained with a toluidine blue and sodium tetraborate solution. For TEM, semi-thin sections were re-sectioned according to [19] (electronic supplementary material, Methods c). Paraformaldehyde fixed samples were decalcified and DOPE-FISH was performed on embedded and sectioned samples using general and specific probes (electronic supplementary material, table S3, Methods d and e) [20]. Details of the light, fluorescence and electron microscopes used are found in electronic supplementary material, Methods f.

## (b) SRμCT measurements

SRμCT datasets were recorded at the DESY using the P05 beamline of PETRA III, operated by the Helmholtz-Zentrum Hereon (Geesthacht, Germany [21]). The X-ray microtomography setup at 15–30 keV and 5× to 40× magnification was used to scan resin-embedded samples with attenuation contrast and uncontrasted samples in PBS-filled capillaries [22] with propagation-based phase contrast. Scan parameters are summarized in electronic supplementary material, table S4. The tomography data were processed with custom scripts implemented in the ASTRA toolbox [23–25] (electronic supplementary material, Methods g).

## (c) Correlative workflow for SRμCT, light and electron microscopy, and FISH

Section series were screened by eye to identify individual host cells and predict symbiont colonization state based on three morphological characteristics: hypertrophy, loss of microvilli and loss of cilia (table 1, electronic supplementary material, Methods i). The location and symbiont colonization state of the analysed host cells was marked using Cell Counter in Fiji [26]. The analysed sections were re-sectioned and the same fields of view were recorded with TEM (electronic supplementary material, figure S2). To validate the LM-based predictions, symbionts were identified in TEM images based on their morphology (electronic supplementary material, video S5).

## (d) Image processing and three-dimensional visualization

Microscopy images were adjusted and figures composed using Fiji, Adobe Photoshop and Adobe Illustrator 2021. LM-images were stitched and aligned with TrackEM2 [27] in Fiji. Amira 2020.2 (ThermoFisher Scientific) was used to generate three-dimensional models from LM and μCT datasets. Co-registration between μCT, LM and TEM datasets was carried out after [19] (electronic supplementary material, Methods j).

# 3. Results

We analysed developmental stages of *B. puteoserpentis*, *B. azoricus* and *G. childressi* ranging from aposymbiotic pediveligers to symbiotic adults, to determine at which stage the

**Table 1.** Overview of number of individuals analysed for each species, developmental stage, and imaging method. The numbers correspond from left to right to *B. puteoserpentis* (black), *B. azoricus* (dark yellow) and *B. childressi* (blue).

| method | pediveliger | | | metamorphosis | | | plantigrade | | | juvenile | | | adult | | |
|---|---|---|---|---|---|---|---|---|---|---|---|---|---|---|---|
| SRμCT/μCT | — | — | — | — | — | — | — | — | — | 1 | 2 | — | — | 2 | 1 |
| SRμCT + serial sectioning + LM | 1 | — | — | 1 | — | — | 2 | 1 | — | 1 | — | — | — | — | — |
| SRμCT + serial sectioning + LM +TEM | — | — | 1 | — | — | — | — | — | — | — | — | — | — | — | — |
| serial sectioning + LM | 2 | 1 | — | 2 | 1 | 2 | — | 1 | 2 | — | — | — | — | — | — |
| serial sectioning + LM + TEM | 2 | 1 | — | 2 | — | — | 3 | — | — | — | — | — | — | — | — |
| FISH | 2 | — | — | 1 | — | — | 3 | — | — | 1 | — | — | — | — | — |
| total | 7 | 2 | 1 | 6 | 1 | 2 | 8 | 2 | 2 | 3 | 2 | — | — | 2 | 1 |

symbionts colonize their hosts, and the developmental modifications that these mussels have evolved to adapt to their symbiotic lifestyle (figure 1; electronic supplementary material, figures S3 and S4). Because the names for larval stages of bivalves have not always been used consistently, we define them as follows. The earliest life stages in our study were at the last planktonic larval stage—the pediveliger. Once settled on the seafloor, the animal initiates its metamorphosis from a planktonic to a benthic lifestyle, and we refer to this stage as being in metamorphosis. While metamorphosing, the mussel degrades its velum, the larval feeding and swimming organ, and develops into a plantigrade [28]. During the plantigrade stage, the mussel secretes the adult shell and once the ventral groove of the gills, which transports particles to the labial palps, is formed, it enters the juvenile stage [29]. When the gonads are developed, the mussel is considered an adult [18].

## (a) Identification of developmental stages

We measured the shell lengths of 259 specimens: 129 *B. puteoserpentis*, 124 *B. azoricus* and 6 *G. childressi* individuals. We assume that these had already settled or were in the process of settling, as we collected them from mussel beds on the sea floor. The specimens ranged from 370 μm to 4556 μm shell length (electronic supplementary material, tables S2 and S5). The earliest developmental stages were pediveligers, with shell lengths of 366–465 μm. Developmental stages could only be identified through detailed analyses of the mussels' soft body anatomy (electronic supplementary material, figure S1a,b). These analyses revealed that the shell sizes of 58 pediveligers and mussels in metamorphosis overlapped with those of the smallest plantigrade stages (electronic supplementary material, figure S1f).

## (b) Morphological characterization of *Bathymodiolus* and *Gigantidas* developmental stages

For our morphological analyses, we analysed 39 individuals (table 1). *Bathymodiolus puteoserpentis* specimens were best preserved and covered the widest range of developmental stages. We therefore focussed our detailed morphological analyses on *B. puteoserpentis*, and compared these with selected *B. azoricus* and *G. childressi* stages (electronic supplementary material, figures S3 and S4). In the following, we describe the shared morphological features of all three species unless specified otherwise.

The pediveliger larvae were characterized by the presence of a velum, a fully developed digestive system, a foot with two pairs of retractor muscles and two gill baskets (figure 1a; electronic supplementary material, table S6, figure S5a,d and video S1). The digestive system consisted of the mouth, oral labial palps, oesophagus, stomach, two digestive glands, gastric shield, the style sac with crystalline style, mid gut, s-shaped looped intestine and anal papillae (figure 1a; electronic supplementary material, figure S6). The diverticula of the digestive glands and the epithelia of the stomach contained membrane-bound lipid vesicles (figure 1; electronic supplementary material, figures S6 and S7). The gill baskets on each side of the foot (figure 1a; electronic supplementary material, figure S6) consisted of three to four single gill filaments in *B. puteoserpentis* and five in *B. azoricus* and *G. childressi* (electronic supplementary material, figure S3b,c). These filaments form the descending lamella of the inner demibranch in later life stages. For further details, see electronic supplementary material, Note 1.

In mussels undergoing metamorphosis, the first steps from a planktonic to a benthic lifestyle were visible in the degradation of the velum (electronic supplementary material, figure S5) and the appearance of byssus threads (electronic supplementary material, figure S8). Rearrangements of all organs occurred in this stage, for example, the alignment of the growth axis of the gill 'basket' with the length axis of the mussel (figure 1a–c; electronic supplementary material, figure S9 and video S2). The number of gill filaments increased by one in all species (electronic supplementary material, figure S6). Furthermore, gill filaments separated from each other, increasing the gaps between them from 47 μm to 120 μm (figure 1c). In the digestive gland and stomach epithelia, the number and volume of lipid vesicles decreased from 12.8% of the soft body volume in pediveligers, to 3.9% in metamorphosing mussels, 1.5% in plantigrades, and were no longer present in juveniles and adults (figure 1a–d; electronic supplementary material, table S6). The organs involved in filter feeding also changed. In pediveligers, the main feeding organ was the velum, which collects particles from the seawater and transports them to the oral labial palps and mouth. After the degradation of the velum, particle sorting was taken over by the highly ciliated foot (electronic supplementary material, figures S5, S8 and S9) in metamorphosing mussels, and by the gills in late plantigrades and juveniles. The plantigrade stage began once the mussels secreted the dissoconch and completed metamorphosis (electronic supplementary material, figures S1 and S10). As the mussels transitioned from the plantigrade to the juvenile stage, the digestive system straightened (figure 1c,d; electronic supplementary material, figure S11 and videos S3 and S4). This morphological change was most prominent in the intestine,

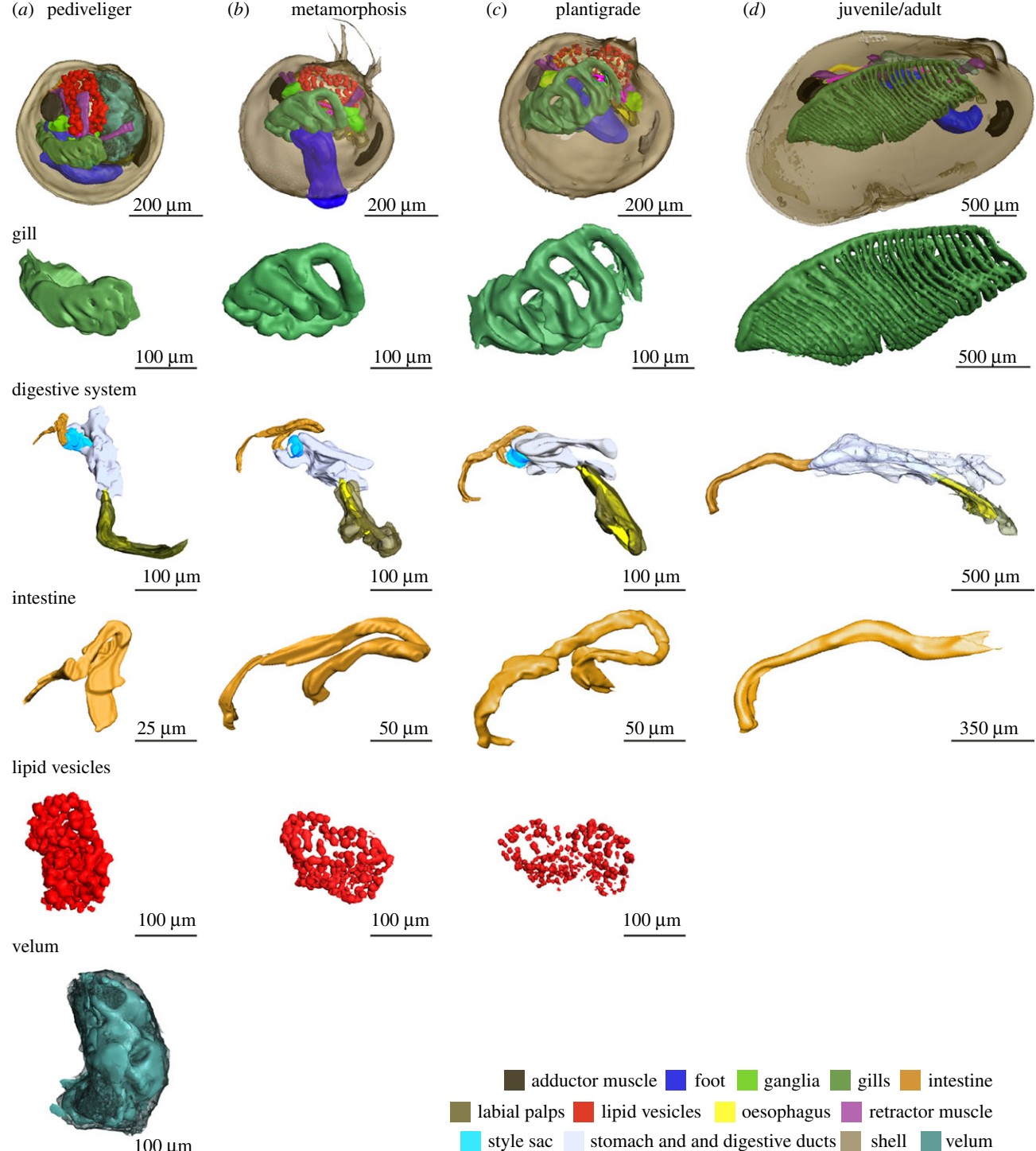

(*a*) pediveliger     (*b*) metamorphosis     (*c*) plantigrade     (*d*) juvenile/adult

gill

digestive system

intestine

lipid vesicles

velum

| adductor muscle | foot | ganglia | gills | intestine |
| labial palps | lipid vesicles | oesophagus | retractor muscle |
| style sac | stomach and and digestive ducts | shell | velum |

**Figure 1.** Three-dimensional visualization of section series and SRμCT measurements of *B. puteoserpentis* developmental stages based on analyses of four individuals for each stage. Note the different scale bars.

which went from a looped to a straight shape and remained straight in all later developmental stages (figure 1*c*,*d*; electronic supplementary material, figure S11). For further details see electronic supplementary material, Note 2.

## (c) Establishment of the symbiosis

Central to an accurate assessment of symbiont colonization and symbiont-mediated morphological changes was our correlative approach, which combined SRμCT, light and electron microscopy (electronic supplementary material, video S5 and figure S2) and was complemented with FISH. This allowed us to rapidly screen whole animals, yet achieve the resolution needed to identify the colonization of single eukaryotic cells

by symbiotic bacteria. We first searched for a morphological characteristic that was visible using light microscopy and reliably revealed the presence of symbionts in host cells. Previous studies [30,31] showed that in juvenile and adult *Bathymodiolus* mussels, the morphology of epithelial cells colonized by symbionts is fundamentally altered: (i) the microvilli that cover all epithelial cells are lost (known as microvillar effacement) and (ii) epithelial cells become hypertrophic (swollen) compared to aposymbiotic cells. We identified a third characteristic change in epithelial cells colonized by symbionts that has not received much attention, namely the loss of cilia (figure 2*c*, *f*,*i*). We tested if these three morphological characteristics had predictive power for symbiont colonization by analysing 1965 epithelial cells from a subset of seven *B. puteoserpentis*

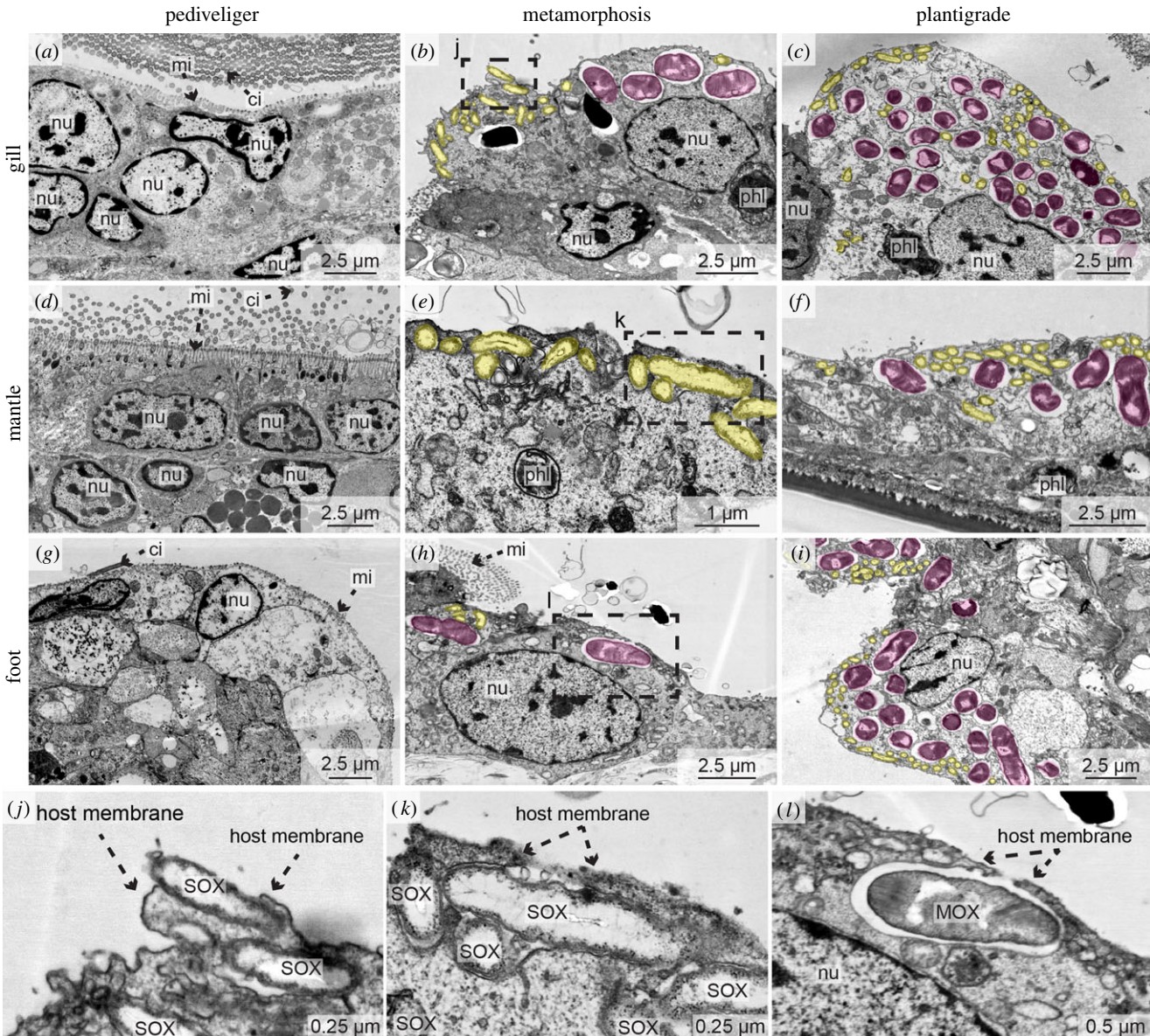

**Figure 2.** Symbionts first colonize *Bathymodiolus puteoserpentis* during metamorphosis. TEM micrographs of gills (*a–c*), mantle (*d–f*) and foot (*g–i*) epithelial tissues of pediveligers (*a,d,g*), metamorphosing mussels (*b,e,h*) and plantigrades (*c,f,i*). SOX (yellow) and MOX symbionts (magenta) are highlighted with colour overlays for visibility reasons (*a–i*). All epithelial tissues of pediveligers were aposymbiotic (*a,d,g*). Colonization by the SOX and MOX symbionts was first observed during metamorphosis (*b,e,h*). In plantigrades, all epithelial tissues were colonized by both symbiont types (*c,f,i*). Dashed boxes indicate regions in which symbionts were in the process of colonizing epithelial tissue (shown magnified in *j–l*). ci, cilia; mi, microvilli; MOX, methane-oxidizing symbiont; nu, nucleus; phl, phagolysosome; SOX, sulfur-oxidizing symbiont. For raw image data see electronic supplementary material, figure S14.

individuals (table 1) and comparing LM images of these cells with their correlated TEM images. Our analyses revealed that all cells predicted to have symbionts in the LM dataset were indeed colonized in the TEM dataset, and likewise, all cells predicted to be aposymbiotic were free of symbionts (electronic supplementary material, figure S2). Our approach allowed us to identify host cells that were colonized by only a few SOX symbionts based on the absence of microvilli and cilia, indicating that these are lost immediately after the first symbionts colonize host cells (figure 2*e*). We next used our verified morphological characters to reveal the onset of symbiont colonization in *B. puteoserpentis* in this subset of seven mussels. All pediveliger cells were free of symbionts (*n* = 797 host cells in two pediveliger). In the metamorphosing mussels examined with our correlative approach 1–15% (*n* = 488 host cells in two metamorphosing mussels) and in plantigrades 21–26% (*n* = 680 host cells in three plantigrades) of all analysed gill, mantle, foot and retractor muscle epithelia cells were colonized by symbionts.

We then expanded our analyses to LM on another 10 *B. puteoserpentis* individuals (table 1). These analyses confirmed our results from the correlative dataset: All additional pediveligers (*n* = 3) and metamorphosing mussels were aposymbiotic (*n* = 3), all additional plantigrades (*n* = 2) and juveniles (*n* = 2) were colonized by symbionts. Finally, we performed FISH on another seven *B. puteoserpentis* individuals (table 1). These analyses corroborated our LM data on the timing of symbiont colonization, with all pediveliger aposymbiotic and symbiont colonization beginning at metamorphosis (figure 3; electronic supplementary material, figures S12 and S13).

Our correlative workflow revealed that pediveliger were aposymbiotic in all three mussel species (figure 2*a,d,g*; electronic supplementary material, figure S15a,c). Interestingly, two of the aposymbiotic *B. puteoserpentis* pediveligers had bacterial morphotypes similar to the SOX and MOX symbionts attached to the outside of their shell (electronic supplementary material, figure S16). Mussels that were undergoing

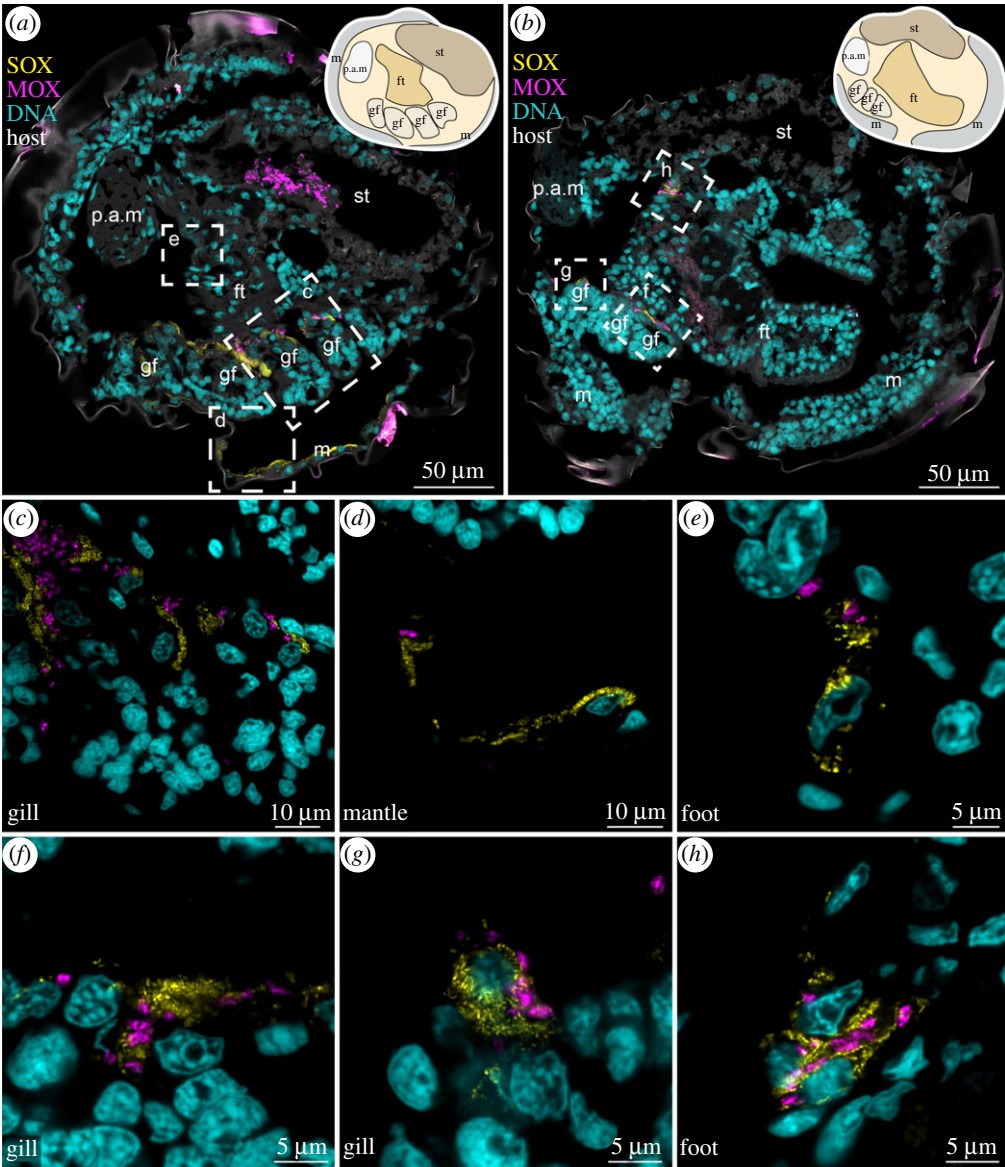

**Figure 3.** SOX and MOX symbionts colonize all epithelial tissues in *Bathymodiolus puteoserpentis* plantigrades. False-coloured FISH images show probes specific for SOX (yellow, BMARt-193) and MOX symbionts (magenta, BMARm-845) and host nuclei stained with DAPI (cyan). Sagittal cross-sections of two individuals (*a,b*) show SOX and MOX symbionts in the gill, foot and mantle epithelia. Schematic drawings of the anatomy are provided in the top right corners. To visualize host tissue autofluorescence is shown in grey in (*a*) and (*b*). Dashed boxes indicate magnified regions of the colonized gill, mantle and foot region shown in (*c–h*). ft, foot; m, mantle; gf, gill filament; p.a.m, posterior adductor muscle; st, stomach.

metamorphosis were the earliest developmental stage in which we found symbionts in all three host species, with a shell length of 432 µm in the smallest *B. puteoserpentis* individual, 510 µm in *B. azoricus*, and 383 µm in *G. childressi*. In these metamorphosing mussels, we observed symbionts in epithelial cells of the gill filaments, mantle, foot and retractor muscle (figure 2*b,e,h, j–l*; electronic supplementary material, figure S15). Once symbiont colonization began, gill tissue morphology was similar to that of adult mussels [30–32]: the majority of gill cells were symbiont-containing bacteriocytes without microvilli or cilia, whereas the only gill cells without symbionts were those at the ventral ends of the gill filaments and at the frontal-to-lateral zones along the length of the filaments, as well as the intercalary cells (electronic supplementary material, figures S6, S9 and S10). In addition to the gills, the epithelial tissues of the mantle, foot and retractor muscles also had symbionts in all three host species (figure 2*c,f,i*; electronic supplementary material, figure S15). We never observed other bacteria besides the two symbionts in any of the developmental stages, including the

intranuclear parasite that infects these mussels, based on FISH analyses with symbiont-specific and eubacterial probes of *B. puteoserpentis*, and TEM analyses of symbiont morphology of *B. puteoserpentis*, *B. azoricus* and *G. childressi* specimen (figures 2 and 3; electronic supplementary material, figures S12, S13 and S15).

The superior preservation of *B. puteoserpentis* specimens allowed us to analyse the process of symbiont colonization in this species in more detail. We first found evidence of symbiont colonization in metamorphosing mussels that had only a few bacteria in gill, mantle and foot epithelial (figure 2*b,e,h,j–l*). Bacterial density per gill cell was the lowest in metamorphosing mussels with 13.7% (±6.3) of the host cell area occupied by symbionts, and steadily increased in later developmental stages, reaching up to 29.0% (±5.0) in plantigrades and 32.1% (±6.1) in adult mussels (electronic supplementary material, table S7). With the onset of symbiont colonization, we observed phagolysosomal digestion of the symbionts in all epithelial cells, even those with only very few symbionts

(figure 2b,c,e,f). The process of symbiont colonization appeared to be extremely rapid. Nearly all metamorphosing mussels had either no symbionts at all, or all of their epithelial tissues were colonized. In only two out of six individuals, we occasionally observed SOX and MOX symbionts that were not completely engulfed by the host's apical cell membrane, which we interpreted as ongoing colonization (figure 2j–l). These first steps in colonization were particularly common in mantle epithelial cells, while in the same specimen the gill epithelial cells were already fully colonized (figure 2b,e,f). Furthermore, in host cells where colonization was ongoing, we observed that these were colonized only by SOX (figure 2e), or by both SOX and MOX (figure 2h), but we never observed host cells only colonized by MOX symbionts.

## 4. Discussion

### (a) Post-metamorphosis development in Bathymodiolus and Gigantidas deviates from the mytilid blueprint

Our study shows that the use of shell characteristics alone to determine developmental stages of deep-sea mussels is not reliable (e.g. [16]), particularly for shell lengths of settling pediveligers and early plantigrades. Our analyses of shell lengths in developmental stages of the three mussel species revealed an overlap in size of 50 µm between pediveligers and plantigrades. This inconsistency in shell lengths between developmental stages indicates that metamorphosis begin is not dependent on size, and provides further evidence for the ability of mussels to delay metamorphosis while continuing to grow, as previously suggested for G. childressi [33]. Such a delay could be due to a lack of settlement cues or limited nutrition, similar to what is known from M. edulis [34]. Delaying metamorphosis would favour dispersal, potentially leading to an increase in geographical distribution and the colonization of new and remote habitats [35].

The pre-metamorphosis development of Bathymodiolus and Gigantidas mussels is similar to that of their close relatives from the genus Idas [7,15] and shallow water mytilids such as M. edulis [18,29,36]. The pediveligers of Bathymodiolus, Gigantidas, Idas and Mytilus have a large velum, foot, mantle epithelium, digestive system, central nerve system and two preliminary gill baskets consisting of three to five gill buds [7,15,18]. During metamorphosis, the velum is degraded, organs within the mantle cavity are rearranged and the lipid vesicles in the digestive diverticula and stomach are reduced. In early developmental stages of mussels from the Mytilidae, lipid vesicles serve as storage compounds to fuel the energy-demanding process of metamorphosis [18] and they probably have a similar function in Bathymodiolus and Gigantidas. Furthermore, these lipid vesicles could provide energy for movement of the pediveligers during their searches for sites to settle [37,38].

Although early development appears to be conserved across Bathymodiolus, Gigantidas, Idas and their shallow water relatives, marked differences occur as soon as symbiont colonization begins in the deep-sea mytilids. All colonized epithelial cells lost not only their microvilli but also their cilia, and developed a hypertrophic habitus, as previously shown for gill bacteriocytes [7,30,31]. If the symbionts of Bathymodiolus and Gigantidas actively induce these cellular changes or if these are a response by the host to symbiont colonization remains unresolved, but our data shows that these processes were tightly linked spatially and temporally. Furthermore, our correlative analyses demonstrate that these cell surface modifications serve as reliable markers for the state of symbiont colonization. Observations of effacement of cilia and microvilli have been reported for a wide range of bacteria that invade epithelia, particularly pathogens, and for these it also remains to be shown if the bacteria or the host drive these processes [39–41].

Although it has been known for several decades that the symbionts of Bathymodiolus and Gigantidas mussels supply their host with nutrition, and that adults possess only a rudimentary gut, nothing was known about the development of the digestive system in these mussels. We observed a straightening of the digestive system after completion of metamorphosis in all three species (figure 4). The stomach and the intestine straightened and the digestive system changed from the complex looped type found in Mytilus to the straight type seen in adults of bathymodioline mussels (electronic supplementary material, table S8). This transformation is striking, as in mytilids like M. edulis, such drastic morphological changes after metamorphosis are not known [42]. In Bathymodiolus and Gigantidas, the straightening of the digestive system did not coincide with the first stages of symbiont colonization or metamorphosis, but rather occurred during the transition from the plantigrade to the juvenile stage, well after metamorphosis and only when these hosts had become fully colonized by their symbionts (figure 4).

In general, the morphology of an animal's gastrointestinal tract reflects its food sources. Animals that digest complex foods possess enlarged compartments and lengthened gastrointestinal structures to slow down the flow of digested material and increase the breakdown of complex molecules [43]. We hypothesize that the straightening of the digestive system in the vent and seep mussels we analysed here was induced by their shift from filter feeding to gaining nutrition from their symbionts. The straightening of the digestive tract could be an evolutionary adaptation to minimize the energy needed for maintaining the digestive tract once the majority of nutrition is gained through intracellular digestion of symbionts in the bacteriocytes.

Not all bathymodiolins have a straightened digestive system. Idas intestines are complex with one or more loops [7,15,44]. These small mussels are commonly found at organic falls, which typically have higher inputs of organic matter than vents and seeps. Also, symbiont abundances are lower in Idas than in the large vent and seep bathymodiolins, indicating that Idas mussels may depend more on filter feeding than their larger relatives. Intriguingly, the relatively large adults of the bathymodioline genus Vulcanidas have a pronounced looped intestine despite high symbiont abundances [45]. Vulcanidas inhabits relatively shallow hydrothermal vents close to the photic zone (140 m compared to 1000–2500 m depth for the mussels in our study) where more organic material from the photic zone is available for their nutrition. It is thus likely that filter feeding plays a greater role in the nutrition of Idas and Vulcanidas than in the three species analysed in this study. These findings raise the question whether the straightening of the digestive tract is a conserved developmental trait in only some genera, or if the environment and availability of organic matter and other non-symbiotic food sources drives the morphology of the digestive tract in bathymodioline mussels.

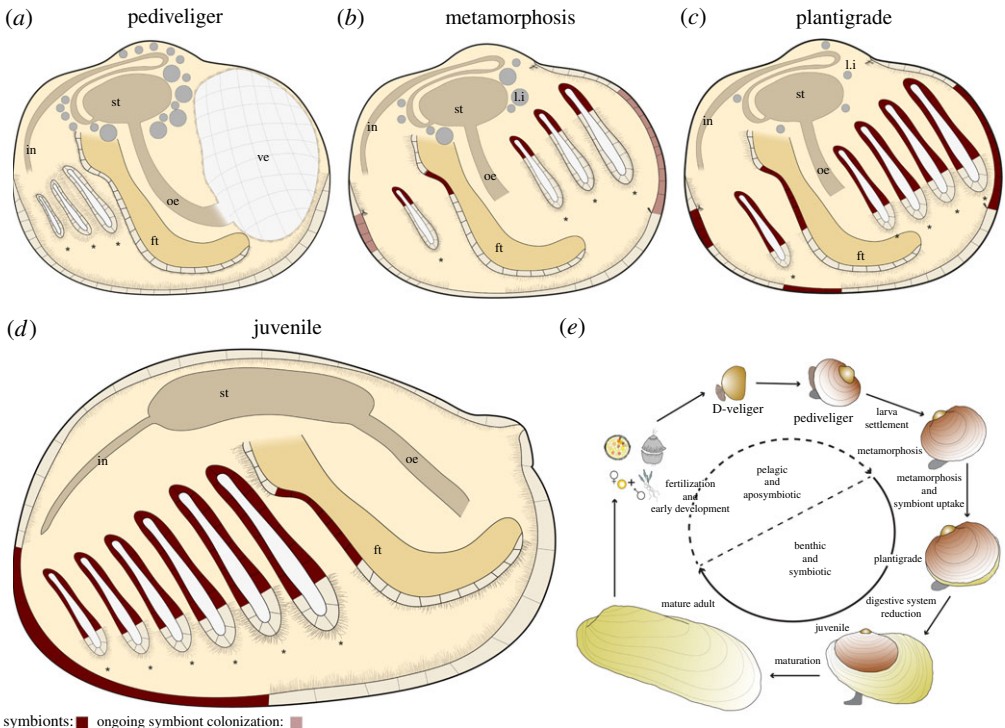

**Figure 4.** Summary of symbiont colonization and development of *Bathymodiolus* and *Gigantidas* mussels. Pediveliger are aposymbiotic (*a*), and symbiont colonization begins in metamorphosing mussels (*b*). By the plantigrade (*c*) and juvenile (*d*) stages, all epithelial tissues are fully colonized by symbionts. The digestive system is reduced between the plantigrade and juvenile stages, changing from a looped to a straight morphology. (*e*) Schematic of the hypothetical life cycle of bathymodioline mussels indicating the aposymbiotic pelagic developmental stages and the symbiotic benthic developmental stages. ft, foot; gf, gill filament; in, intestine; l.i, lipid inclusion; oe, oesophagus.

## (b) Symbiont colonization begins during the plantigrade stage as soon as the velum is degraded

As recently highlighted, how and when *Bathymodiolus* and *Gigantidas* mussels acquire their symbionts has remained, as yet, unclear [9]. Here, we used correlative imaging analyses to reveal that the early developmental stages of two *Bathymodiolus* and one *Gigantidas* species were aposymbiotic, and narrowed the window of symbiont acquisition to mussels undergoing metamorphosis (figure 4). Our analyses revealed that the symbionts colonize the gills only after the velum is lost. As long as the velum is present and active, particles from the surrounding seawater are either transported directly into the digestive system or expelled from the mussel. In *M. edulis*, once the velum is degraded the gill filaments separate further from each other and take over the task of sorting food particles and generating a water current [18,29]. Beginning in the late plantigrade stage of *Bathymodiolus*, we identified a similar increase of space between the gill filaments. Given that the developmental changes in the morphology of the bathymodiolin gills are so similar to those of *M. edulis* [7,15,18,29], it is likely that the functional roles of the gills in *Bathymodiolus* change in a similar manner as in *M. edulis*: The pediveligers' gills of *Bathymodiolus* are used for respiration only, and then become responsible for water current generation in late metamorphosing mussels, and the sorting of particles in plantigrades and juveniles. We hypothesize that this functional shift of the gills plays an important role in enabling the symbionts to adhere to the gill epithelium and initiate symbiont colonization.

The initial colonization of the host by its symbionts appears to be rapid, based on our observation that 22 out of 24 *B. puteoserpentis* individuals were either completely aposymbiotic or fully colonized, and the first stages of symbiont colonization were only visible in two metamorphosing mussels. As we were working with preserved samples that represent a snapshot of development, the chance of observing a process depends on how often it occurs and how long it takes. The less frequent or the faster a process happens, the smaller the chance of observing it. We therefore conclude that symbiont colonization occurs rapidly in *B. puteoserpentis*, given the small percentage of individuals in which we observed the first steps of colonization. Symbiont colonization seems to be even more rapid than in *I. simpsoni* and *I. modiolaeformis* [7,15]. This could reflect the importance for *Bathymodiolus* mussels to quickly acquire symbionts once they have nearly consumed their internal energy reserves and settled in an environment that lacks energy-rich planktonic nutrition [46].

We found symbionts in epithelial cells of the gills, mantle, foot and retractor muscle in the plantigrades and juveniles of all three bathymodioline species, similar to previous studies [16,47,48]. Previous work suggested that the symbionts first colonize the mantle epithelia, and from there colonize gill cells, as the first gill filaments are formed from mantle tissues [47]. Our data contradicts this assumption, as the gills had already begun to develop in pediveligers before the onset of symbiont colonization. Furthermore, in two *B. puteoserpentis* specimens we detected fully colonized gills, while the mantle tissue was still in the process of being colonized. Our findings indicate that symbionts first colonize gill cells before colonizing other epithelial tissues, and that the SOX symbionts colonize individual host cells first, before the MOX.

Our analyses revealed that *B. puteoserpentis*, *B. azoricus* and *G. childressi* larvae are aposymbiotic during their planktonic phase and first acquire their symbionts when they transition to a benthic lifestyle. The timing of symbiont acquisition and major developmental changes was similar in all three species, and, with the exception of the digestive system, corresponds to observations of early life stages of *Idas* [7,9,15]. These similarities in developmental biology and symbiont acquisition suggest that these traits are conserved in bathymodiolins. The acquisition of symbionts after settlement allows these mussels to recruit locally adapted symbionts. As the geochemistry of vent and seep environments varies strongly across spatial scales [49], recruiting locally adapted symbiont populations would confer a strong fitness advantage to these hosts. Indeed, recent studies have revealed that *Bathymodiolus* mussels host multiple strains of symbionts that vary in key functions, such as the use of energy and nutrient sources, electron acceptors and viral defence mechanisms [50,51]. By acquiring their symbionts from the sites where they settle, bathymodioline mussels can establish symbioses with those strains that are best adapted to the local environment.

## 5. Conclusion and outlook

Our correlative imaging workflow revealed the intricate developmental processes from the subcellular to the whole animal scale that are thought to be triggered when deep-sea mussels acquire their symbionts. Furthermore, we identified the narrow window in which symbiont acquisition begins and showed the morphological changes of the digestive system following symbiont uptake. Given that we never observed bacterial morpho- or phylotypes other than the known SOX and MOX symbionts, even in the earliest larval life stages, strong recognition mechanisms must ensure this high specificity. Now that we have identified when and how symbiont colonization occurs in *Bathymodiolus* and *Gigantidas*, a spatial and temporal transcriptomic approach could shed light on the underlying molecular mechanisms of symbiont recognition, acquisition and maintenance, and further our understanding of the entwined dialogue between animal hosts and their microbial symbionts.

Data accessibility. LM data, µCT-data and videos are available on figshare [52].

Authors' contributions. M.F.: conceptualization, data curation, formal analysis, investigation, methodology, visualization, writing—original draft, writing—review and editing; B.G.: data curation, formal analysis, visualization, writing—review and editing; J.U.H.: data curation, formal analysis, resources, writing—review and editing; N.D.: conceptualization, funding acquisition, resources, supervision, writing—review and editing; N.L.: conceptualization, data curation, formal analysis, investigation, methodology, project administration, supervision, visualization, writing—original draft, writing—review and editing.

All authors gave final approval for publication and agreed to be held accountable for the work performed therein.

Competing interests. We declare we have no competing interests.

Funding. Open access funding provided by the Max Planck Society.

Funding was provided by the MARUM Cluster of Excellence 'The Ocean Floor' (Deutsche Forschungsgemeinschaft (German Research Foundation) under Germany's Excellence Strategy—EXC-2077-39074603), a Gordon and Betty Moore Foundation Marine Microbial Initiative Investigator Award (grant no. GBMF3811 to N.D.) and a European Research Council Advanced Grant (Bathy-Biome, grant no. 340535 to N.D.). µCT measurements were performed at the DESY under the proposal IDs: 20170337 and 20180295.

Acknowledgements. We thank the captains, crew members and ROV pilots of the cruises M126, M82-3 and NA58. We are grateful to Christian Borowski and Stéphane Hourdez for their valuable contributions to collecting mussel larvae and Wiebke Ruschmeier for her help in the laboratory. We thank all involved in supporting us at the DESY at the P05 beamline of PETRA III (Helmholtz-Zentrum Hereon, Geesthacht, Germany). We also thank Benjamin Cooper (Max-Planck Institute for Experimental Medicine, Göttingen) for preliminary sample preparation, and Bernhard Ruthensteiner (Zoologische Staatsammlung München) and Frank Melzner (GEOMAR, Kiel) for fruitful discussions.

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
