## [Peer Review File · Proceedings of the Royal Society B: Biological Sciences]

Review History

RSPB-2021-1044.R0 (Original submission)

Review form: Reviewer 1 (Sven Laming)

Recommendation

Accept with minor revision (please list in comments)

Scientific importance: Is the manuscript an original and important contribution to its field?

Excellent

General interest: Is the paper of sufficient general interest?

Good

Quality of the paper: Is the overall quality of the paper suitable?

Acceptable

Is the length of the paper justified?

Yes

Should the paper be seen by a specialist statistical reviewer?

No

Do you have any concerns about statistical analyses in this paper? If so, please specify them explicitly in your report.

No

It is a condition of publication that authors make their supporting data, code and materials available - either as supplementary material or hosted in an external repository. Please rate, if applicable, the supporting data on the following criteria.

Is it accessible?

Yes

Is it clear?

No

Is it adequate?

No

Do you have any ethical concerns with this paper?

No

Comments to the Author

I attempted to post comments directly here but the website kept encountering errors, so I have attached the file instead. (See Appendix A)

Review form: Reviewer 2 (Stephanie Markert)

Recommendation

Accept with minor revision (please list in comments)

Scientific importance: Is the manuscript an original and important contribution to its field?

Excellent

General interest: Is the paper of sufficient general interest?

Good

Quality of the paper: Is the overall quality of the paper suitable?

Excellent

Is the length of the paper justified?

Yes

Should the paper be seen by a specialist statistical reviewer?

No

Do you have any concerns about statistical analyses in this paper? If so, please specify them explicitly in your report.

No

It is a condition of publication that authors make their supporting data, code and materials available - either as supplementary material or hosted in an external repository. Please rate, if applicable, the supporting data on the following criteria.

Is it accessible?

Yes

Is it clear?

Yes

Is it adequate?

Yes

Do you have any ethical concerns with this paper?

No

Comments to the Author

Please see attached Word file. (See Appendix B)

Review form: Reviewer 3

Recommendation

Accept with minor revision (please list in comments)

Scientific importance: Is the manuscript an original and important contribution to its field?

Excellent

General interest: Is the paper of sufficient general interest?

Excellent

Quality of the paper: Is the overall quality of the paper suitable?

Excellent

Is the length of the paper justified?

Yes

Should the paper be seen by a specialist statistical reviewer?

No

Do you have any concerns about statistical analyses in this paper? If so, please specify them explicitly in your report.

No

It is a condition of publication that authors make their supporting data, code and materials available - either as supplementary material or hosted in an external repository. Please rate, if applicable, the supporting data on the following criteria.

Is it accessible?

Yes

Is it clear?

Yes

Is it adequate?

Yes

Do you have any ethical concerns with this paper?

No

Comments to the Author

This is an important contribution to our understanding of morphological development in coordination with the timing of symbiont acquisition in a widespread and highly-studied group of chemosymbiotic mussels from deep-sea hydrothermal vents and cold seeps. Through meticulous study of an impressive number of specimens, and using a clever combination of microscopy techniques, the authors have provided clear descriptions of how organs develop from the late larval to the adult stage (through metamorphosis), and have managed to establish, for the first time, the timing and process of symbiont uptake in this group of bivalves. The investigative approach used combines detailed reconstructions of organs in whole specimens along with histological analysis, FISH and TEM, which allow confirmation of symbiont type, presence, and apparent endocytosis and intracellular lysis in various organs. In particular, the combination of histological images and reconstructed tomographs, as presented in the supplementary videos, provide exceptional detail and a clear understanding of the processes described in the text.

The analysis performed also yielded important data with regards to relationships between developmental stage and shell size in *Bathymodiolus*, as well as the relative timing of acquisition of sulfur-oxidizing and methanotrophic symbionts in *B. puteoserpentis*. Interestingly, the analysis revealed how the morphology of the digestive tract changes during development, likely due to a change in host nutrient acquisition modes. The observation of some SOX- and MOX-like bacterial morphotypes on the shells of *B. puteoserpentis* pediveliger and postlarvae is also of interest and should stimulate further work.

The text is clear, logical and sufficient detail is provided to enable non-experts to understand the study. The figures are of exceptional quality, and I particularly appreciate the highly informative Figure 4 which elegantly summarizes the study's findings.

This study is relevant to multiple fields of study (from animal-microbe symbiosis to marine ecology, developmental biology, and evolution, for example) and should inspire further research in this exciting area.

I have a few minor comments:

line 33: period missing at end of sentence.

line 36: these mussels still rely to some extent on filter-feeding in the adult stage. Perhaps reword to something like: "is likely linked to the decrease in importance of filter-feeding as intracellular digestion of symbionts becomes possible."

lines 42-43: bathymodiolins also occur in other habitats (whale falls and wood falls)

line 69: "late larval phase": perhaps the term "post-larval" would be more appropriate here?

line 77 (and 175): perhaps specify that these "very early, aposymbiotic life stages" were at the post-settlement stage (I was left with the impression that you had collected planktonic stages).

line 111: how did you mount semi-thin sections on a resin block? Using an adhesive?

line 117: were these PFA-fixed specimens decalcified prior to dehydration and embedding?

line 463: "and" instead of "und"

Supplementary materials:

line 44: nervous

line 149: "immersed" rather than "dissolved"

line 253 (legend, Fig S2): add v.g. and o.l.p.

Decision letter (RSPB-2021-1044.R0)

04-Jun-2021

Dear Mr Franke:

Your manuscript has now been peer reviewed and the reviews have been assessed by an Associate Editor. The reviewers' comments (not including confidential comments to the Editor) and the comments from the Associate Editor are included at the end of this email for your reference. As you will see, the reviewers and the Editors have raised some concerns with your manuscript and we would like to invite you to revise your manuscript to address them.

Research ethics:

Use of animals and field studies:

It is a condition of publication that you make available the data and research materials supporting the results in the article. Please see our Data Sharing Policies

(<https://royalsociety.org/journals/authors/author-guidelines/#data>). Datasets should be deposited in an appropriate publicly available repository and details of the associated accession number, link or DOI to the datasets must be included in the Data Accessibility section of the article (<https://royalsociety.org/journals/ethics-policies/data-sharing-mining/>). Reference(s) to datasets should also be included in the reference list of the article with DOIs (where available).

If you wish to submit your data to Dryad (<http://datadryad.org/>) and have not already done so you can submit your data via this link [http://datadryad.org/submit?journalID=RSPB&manu=\(Document not available\)](http://datadryad.org/submit?journalID=RSPB&manu=(Document%20not%20available)), which will take you to your unique entry in the Dryad repository.

Please submit a copy of your revised paper within three weeks. If we do not hear from you within this time your manuscript will be rejected. If you are unable to meet this deadline please let us know as soon as possible, as we may be able to grant a short extension.

Best wishes,
Dr Daniel Costa
<mailto:proceedingsb@royalsociety.org>

Associate Editor
Comments to Author:

We have now obtained three expert reviews of your manuscript, and I am happy to tell you that all three reviewers liked your paper. My own reading of your manuscript corroborates the reviews: this is an interesting piece of work that could become publishable in Proceedings B.

However, although the reviews were generally very positive, one of the reviewers came up with a long list of insightful and useful suggestions, which should lead to an improved manuscript. The comments by the other two reviewers are fewer and perhaps easier to deal with, but they too would likely improve your paper. Briefly, all reviewers felt that the methods section could contain more detail in places, including a clearer description of the specimen handling workflow, and in some cases methods that are described in the results would be more suitable in the

methods section. The methods and results surrounding replicates could be better explained. The discussion could be more developed with a fuller explanation of the proposed model. One reviewer also noted that the study organism should be considered two genera instead of one; if this is the case, the implications of this should be addressed. This reviewer also points out that more precise references should be made to original studies rather than review articles.

Please see attached Word file.

Referee: 3

Comments to the Author(s)

This is an important contribution to our understanding of morphological development in coordination with the timing of symbiont acquisition in a widespread and highly-studied group of chemosymbiotic mussels from deep-sea hydrothermal vents and cold seeps. Through meticulous study of an impressive number of specimens, and using a clever combination of microscopy techniques, the authors have provided clear descriptions of how organs develop from the late larval to the adult stage (through metamorphosis), and have managed to establish, for the first time, the timing and process of symbiont uptake in this group of bivalves. The investigative approach used combines detailed reconstructions of organs in whole specimens along with histological analysis, FISH and TEM, which allow confirmation of symbiont type, presence, and apparent endocytosis and intracellular lysis in various organs. In particular, the combination of histological images and reconstructed tomographs, as presented in the supplementary videos, provide exceptional detail and a clear understanding of the processes described in the text.

The analysis performed also yielded important data with regards to relationships between developmental stage and shell size in *Bathymodiolus*, as well as the relative timing of acquisition of sulfur-oxidizing and methanotrophic symbionts in *B. puteoserpentis*. Interestingly, the analysis revealed how the morphology of the digestive tract changes during development, likely due to a change in host nutrient acquisition modes. The observation of some SOX- and MOX-like bacterial morphotypes on the shells of *B. puteoserpentis* pediveliger and postlarvae is also of interest and should stimulate further work.

The text is clear, logical and sufficient detail is provided to enable non-experts to understand the study. The figures are of exceptional quality, and I particularly appreciate the highly informative Figure 4 which elegantly summarizes the study's findings.

This study is relevant to multiple fields of study (from animal-microbe symbiosis to marine ecology, developmental biology, and evolution, for example) and should inspire further research in this exciting area.

I have a few minor comments:

line 33: period missing at end of sentence.

line 36: these mussels still rely to some extent on filter-feeding in the adult stage. Perhaps reword to something like: "is likely linked to the decrease in importance of filter-feeding as intracellular digestion of symbionts becomes possible."

lines 42-43: bathymodiolins also occur in other habitats (whale falls and wood falls)

line 69: "late larval phase": perhaps the term "post-larval" would be more appropriate here?

line 77 (and 175): perhaps specify that these "very early, aposymbiotic life stages" were at the post-settlement stage (I was left with the impression that you had collected planktonic stages).

line 111: how did you mount semi-thin sections on a resin block? Using an adhesive?

line 117: were these PFA-fixed specimens decalcified prior to dehydration and embedding?

line 463: "and" instead of "und"

Supplementary materials:

line 44: nervous

line 149: "immersed" rather than "dissolved"

line 253 (legend, Fig S2): add v.g. and o.l.p.

Author's Response to Decision Letter for (RSPB-2021-1044.R0)

See Appendix C.

Decision letter (RSPB-2021-1044.R1)

26-Jul-2021

Dear Mr Franke

I am pleased to inform you that your manuscript entitled "Coming together – symbiont acquisition and early development in deep-sea bathymodioline mussels" has been accepted for publication in Proceedings B.

Data Accessibility section

Open Access

Paper charges

Sincerely,

Dr Daniel Costa

Associate Editor:

Board Member

Comments to Author:

(There are no comments.)

Appendix A

General observations

The manuscript has many strengths. Stand out among these are 1) its enterprising use of size-stratified series of specimens in three species of "Bathymodiolus" mussels, which is extremely rare in deep-sea biology (and, as the authors note, also highly fortuitous) and 2) the intelligent use of correlative approaches that allow some comparisons to be made across techniques. Sadly, a seriously underdeveloped discussion, a confused and at times vague methodology which lacks transparency and a narrative that infers results to be unprecedented when in places they are not, undermines the ingenious, intelligent use of samples. That said, I think the MS has the potential to be an outstanding piece of research.

Several issues first need to be addressed.

1) The MS repeatedly focuses on what is and is not known about *Bathymodiolus* mussels. I'd like to draw attention to the fact that you are speaking of *Bathymodiolus* as if it were a monophyletic genus. Sadly this is not the case. Until quite recently, it was of course polyphyletic (or paraphyletic, depending on your point of view) in that it included two genera. Putative "*Bathymodiolus*" spp. that strictly belong in *Gigantidas*, are now accepted as such, in the face of overwhelming evidence that points to this. Some care is needed with this as the nomenclatural issues underlie very real divergent evolutionary histories (albeit relatively recent). This has ramifications for the discussion as on the one hand, you have two sister species capable of hybridisation (and in that situation, they possess phylogenetically indistinct symbionts), and on the other hand, a species from a separate genus, *Gigantidas childressi*. If you are concerned about continuity, you could simply state that you are deviating from the use of "*Bathymodiolus*" as the genus, since the species is now considered to be from the genus *Gigantidas* (see WORMS database)

Also, changing the species name only in the MS is not sufficient, as this would fail to address the degree of relatedness among the three species, which is highly relevant to the ecology of symbiosis. Finally, please note that this problem also applies to the extensive supplementary materials so please ensure you make changes here as well

2) There are too many instances where review articles are used to support quite specific statements. It's considered poor form to use a review, when the original studies discussed therein are more appropriate as references, since the original authors work is not given due credit.

For example:

The role of symbionts in inducing developmental changes in the host in the squid-*Vibrio* symbiosis was described in 1994* and should be cited accordingly (<https://doi.org/10.1242/dev.120.7.1719>), rather than using a review paper published much later (McFall-Ngai 2014 reference, currently number 5 in the MS reference list)

There are even instances where review papers that broadly deal with the appropriate subject area are used to support statements for which no evidence exists within said review (as in the Bright & Bulgheresi review cited in Line 55/56). This is lax, at best, and should be avoided.

*Montgomery and McFall-Ngai 1994
<https://doi.org/10.1242/dev.120.7.1719>

3) Methods: The methods are not clear regarding the workflow employed (particularly the order) for a given specimen fixation and could benefit from being reordered into paragraphs that follow the analytical timeline, rather than paragraphs grouping vaguely similar methods together.

First, you need to define the term "morphological analyses" more explicitly for each fixation approach please. It may seem clear to you how the analyses were performed and in what order and upon which specimens for a given fixation. However, as a researcher who is very familiar with all the techniques individually and some (other) correlative approaches, I still find myself trying to piece together the order of analyses and decipher which specimens were subjected to what treatment. I know you have just put out a preprint on the "chemo-histo-tomography" approach but the method here should be clearly formulated and only supported by the considerable supplementary materials that accompany this MS rather than being dependent on them. Another person needs to at least be able to follow the logic of the analysis. This is currently not possible. More serious still, fundamental methodological details concerning what analyses were carried out on which species that seriously influence the interpretation of your results are buried in the results section (see

detailed comments) and the number of analyses in a given species is not detailed clearly. In short the MS lacks transparency and borders on misinformation at times.

Specific comments

The following comments deal with specific issues (and positive notes) in the MS. It is meant to be constructive and hope that the detail provided helps (I don't do short answers well). Disclaimer: you will probably have a good sense of the identity of this reviewer by the end.

Line 25 "three deep-sea *Bathymodiolus* species"

The MS needs to be restructured in order to acknowledge that you are not looking at *Bathymodiolus* species exclusively, see later comments on this.

Line 32/33

The colonisation of other epithelia in *Bathymodiolus* spp. juveniles following the establishment of symbioses is also already documented (Wentrup et al 2013). It's the timing of acquisition and the fact that gill tissues are among the first to be colonised rather than the last, that is novel to this study.

Line 33 a full stop is missing

Line 33/34 "...symbiont densities in post-larvae reached those of adults"

Careful.... In mytilids, post-larva and plantigrade are morphologically synonymous (but not functionally synonymous). They are terms used to describe individuals settling for the first time but employing markedly different locomotor behaviour. Early and Late plantigrades are terms to describe post-settlement individuals that have settled for the first time and the second time respectively (the latter, among adult conspecifics).

Do you mean to define all post-settlement specimens that have yet to form a ventral groove as post-larvae ?

I presume some of these individuals display dissoconch growth. Note that in deep-sea species, this is often considered the "juvenile dissoconch". If you have chosen to label all individuals that do not have a ventral groove on their gill as post-larvae, it flies in the face of many years of study on mussels, which use different staging conventions.

Line 35 "This morphological change appears to be specific to *Bathymodiolus*"

DO you mean within the mytilids or more generally? Be aware that similar morphological shifts have already been documented in Peltospirid gastropods as well.

Line 36/37

Indeed, though whether the change is mediated by the host or the symbionts still remains unclear. Inferences to the contrary (e.g. bacteria inducing the cryptic metamorphosis in *Bathymodiolus* juvenile from a functional gut and heterotrophy, to a linear ruduced gut and chemosymbiosis) can only ever be speculative without targeted experiments employing symbiont exposure versus aseptic conditions, akin to the squid-vibrio symbiosis (I revisit this again later).

Line 40/41 "By associating with microorganisms, animals can expand their metabolic capabilities and colonize habitats they could not live in on their own"

This is not accurate.

First, not to be pedantic but the host's metabolic capabilities remain unchanged. Instead they benefit from the alternative metabolic capabilities of their symbionts and access to an efficient means of acquiring energy and a source of organic carbon.

Second, engaging in symbioses does not "allow animals to colonise habitats they could not live in on their own". Instead, it typically provides a competitive fitness advantage, be it through: access to an energy source that would otherwise be inaccessible *directly*; the detoxification of a habitat; the realisation of a new (or augmentation of an existing) host trait; or several of these factors. It typically also allows increased somatic and reproductive growth (large sizes, high reproductive output). So the only way in which it aids in the colonisation of habitat is by increased larval output.

In reducing environments, where the resource in question is clearly nutritional, symbioses allow host organisms to survive – or indeed thrive – in a habitat where availability of organic carbon for nutrition is otherwise limiting and critically, through efficient assimilation of and reliably access to a sustained food source without **expending energy on active feeding** (e.g. bacterial mats / biofilms, or in the case of a filter feeders POC as bacterial flocculates for example), providing a clear fitness advantage over heterotrophic feeding strategies (while potentially also allowing the host species to tolerate otherwise toxic habitat conditions).

As a side note, the use of the word colonise here is also particularly counter intuitive for horizontally transmitted symbionts, since this often occurs in the absence of symbiont partners... (e.g. this study!). What allows animals to colonise a habitat is distinct from what allows them to survive/thrive in said habitat: In other words larval supply/settlement *versus* post-settlement mortality.

Thus I'd ask that you be clearer about how you convey your message here please.

Line 42 “bathymodiolin”

Not sure if this is a typo, but bathymodiolin is the noun, bathymodioline is the adjective. In other words it's either "bathymodiolins" or "bathymodioline mussels"

Line 43 *Bathymodiolus*

same issue as above (do you mean *sensu stricto* or *sensu lato*?)

Line 45/46 “the bacteria use reduced compounds in the vent and seep fluids to gain energy for the fixation of carbon and provide nutrition to their hosts”

consider changing to “the bacteria use reduced compounds in the vent and seep fluids as an energy source for carbon fixation, which in turn provides nutrition to their hosts”

The sentence as it currently stands suggests that bacteria actively seek to provide energy to their hosts.

Lines 50 – 55 Any chance we could have some references to support these statements please?

Line 54 change “entailing that” to “where”

Line 55/56 “In many hosts that rely on horizontal transmission, the acquisition of symbionts **induces** drastic morphological and developmental changes”

While there are certainly data on bacterium-mediated changes in host physiology and on bacteria-induced settlement in larvae, data on *demonstrable* bacterium-mediated changes in host morphology and/or on bacteria being a cue for developmental changes remain limited (e.g. certain nematode species, some insects e.g. *Aedes*-associated *Wolbachia* and of course the squid *Euprymna scolopes*). In marine organisms in particular, there is very little empirical evidence to support this statement (that is, it may prove to be a widespread phenomenon in symbioses but we don't yet have the data to say so), with the notable exception of the squid *Euprymna scolopes* and its bioluminescent bacterial symbiont, *Vibrio fischeri*, for which the symbionts role in cuing changes in squid morphology is unequivocal.

Even in *Riftia pachyptila*, for which the process of horizontal transmission during host development has been documented, no evidence is proposed for the bacterial role in cuing / inducing host trophosome development (other than noting that the observed epithelial apoptosis in the host following infection echoes that seen in the *Euprymna-Vibrio* symbiosis during infection).

Thus please change to "is believed to [induce / trigger / cue]... " or something similar, to allude to the fact that we do not yet know which partner mediates this process in most symbioses and, please find a more suitable reference (perhaps one that is more specific) to provide evidence that this is known to occur in certain model species.

You could, for example, write the following for remainder of this paragraph (multiple options provided in square brackets):

"In hosts that rely on horizontal transmission, the acquisition of symbionts may even [*induce / trigger / cue*] morphological and developmental changes in the host, which can often be drastic and typically occur upon

or after the arrival of symbiont partners. These can range from tissue rearrangement, known to be symbiont-induced in the *Euprymna* squid-*Vibrio* symbiosis [REF: Montgomery and McFall-Ngai. 1994], to the large-scale modification of existing organs, typically through hypertrophy (e.g. gill of the bathymodioline *Idas modiolaeformis* REF: Laming et al 2014; oesophageal gland of the peltospirid gastropod *Gigantopelta chessoia* REF Chong et al 2018*) or the development of a novel symbiont-dedicated organ, such as the trophosome of the siboglinid *Riftia pachyptila* (REF Nussbaumer et al 2006) where, in these latter examples, the role of symbionts as mediators remains hypothetical, though probable.”

Finally, the Bright & Bulgheresi review does not provide any evidence to support this statement either, being predominantly concerned with the subject of inter-partner "dialogue" and migration of symbiont partners to target host organs during horizontal transmission (notwithstanding the probable veracity of your statement, given future evidence).

*Chong et al 2018 reference here:
<https://doi.org/10.1098/rspb.2018.1099>

Line 65-67

The findings of these studies are not fully disclosed here (and they are highly relevant to the current study). In addition, transmission electron microscopy was also used in Laming et al 2014, the study by Gaudron et al 2012 (here: doi: 10.1086/BBLv222n1p6) on symbiont transmission in *Idas modiolaeformis* is overlooked entirely and finally, given the definition of juvenile in the current MS deviates significantly from other papers on deep-sea mussel species, it might be prudent not to use the term juvenile here at all, as it is not synonymous with the juveniles in the current study.

Thus the sentence should be rephrased as follows (or similar):

“In two smaller-sized bathymodioline mussels that colonise organic falls and seeps, histological analyses, and fluorescence and transmission electron microscopy showed that both were aposymbiotic as newly settled plantigrades. Symbionts were first identified shortly after the onset of dissoconch shell growth, providing evidence for heterotrophic larval dispersal, post-metamorphic acquisition and horizontal transmission in these species (Laming et al 2014; 2015) These findings corroborated earlier, targeted fluorescent in-situ hybridisations (FISH) of adult reproductive tissues, which did not find any evidence of transovarian (vertical) bacterial transmission (Gaudron et al 2012).”

Line 69

The term "late larval phase" is not accurate here. In Salerno et al 2005, the smallest stages referred to are "post-larvae" (i.e. post-metamorphic), which for deep-sea mussels is normally synonymous with plantigrade in shallow water mussels*. Post-larvae, by definition, are "after larvae". Thus, they were not from the late larval phase, which would be the pediveliger, but rather the earliest post-metamorphic stage, i.e. either postlarvae (as stated in that study) or, following your classification system plantigrades / early postlarva without dissoconch,.

*By the way, the definitions for shallow water mytilids are more complicated, due to the propensity for byssus-drifting postlarvae that become temporarily resuspended and thus planktonic, making them functionally distinct from crawling plantigrades but not necessarily morphologically distinct (differences in byssus type being excepted). I revisit this distinction in later comments

Line 79 “...“*B*”. *childressi* (also referred to as *Gigantidas childressi* e.g. [15]).”

In the face of overwhelming evidence from multiple papers (including some robust concatenated analyses), it is now probably more appropriate to refer to this species as *Gigantidas childressi*, accepted as such in WORMS.

Line 84 Suggestion for restructuring and rewording of sentence to be clearer and more succinct (multiple suggestions provided in square brackets):

“This approach allowed an integrative analysis of the processes involved in symbiont colonization and its effects on the host body plan, [from the organismal to the cellular level / from the whole animal down to single host and symbiont cells].

Line 104

Were measurements made of the Prodissoconch I and PII in all individuals or was only total shell length measured (i.e. equivalent to Prodissoconch II only in those specimens for which no dissoconch shell had developed)?

Line 109 low-viscosity resin
LR white, I presume?

Line 110 This is an ultramicrotome, not a microtome

Line 117

Until this point the MS reads like all methods were performed on all species (e.g. abstract, and the closing paragraph of the introduction). However, clearly FISH was only performed on *B. puteoserpentis*. In fact, examination of the supplementary materials (for which, by the way, I have created a separate section of corrections) suggests that *B. puteoserpentis* has been used as the model species for most analyses. I now see that this is ultimately alluded to in the results. You need to be MUCH clearer about which species have received which analyses in the main-text methods. To force a reader to dive into supplementary material to confirm this, is not acceptable and stating it in results is not appropriate. There is nothing inherently wrong with using one species as an example and making comparisons with other species, but it has to be clear that this is happening from the start. As a minimum you must provide details concerning the number of specimens actually analysed by various techniques, since the veracity and robustness of the findings can only be assessed knowing this information?

Line 120 "probe"

It was a cocktail of three general probes, was it not? Worth mentioning, since this is a clever way to achieve more "universal" coverage.

Line 137 abbreviation of STEM is not defined (Scanning Transmission Electron Microscopy, I presume?)

Line 149

Is this an actual average value, or a colloquialism? If so you need a measure of variance and define which type (presumably a mean value?)

Line 163

So, here is an example of where an honest inaccuracy veers towards misinformation. Figure 1 does not represent the results of your analysis of three species. It represents the results of a given (representative) specimen for each developmental stage from one species only subjected to either serial sectioning or SruCT and visualised in 3D following segmentation. While it might be representative of what you found in all three species (and I can't be sure if it is because the results and methodology do not provide sufficient information regarding the number of samples examined for each species /stage and by what means), it is not a suitable figure reference for this opening statement. The supplementary materials are the only data that truly encapsulate this statement.

Note also previous comments about the fact that you have three species from two genera of large bathymodiolin mussel, not three species of Bathymodiolus and that of the three, the most distantly related *G. childressi*, which also inhabits a different reducing habitat in a separate oceanographic region in the Atlantic, is the species for which you have least samples, unless I misunderstood?

Line 169 "post-larva"

Ok, so mytilid life stages... Postlarval forms are not so clear cut in mytilids as they are in other bivalves, and your use of these terms does not appear to follow the convention of many other shallow water studies:

As pointed out in Baker and Mann 1997, in mussels, there exist "...two functional life history stages intermediate between the pediveliger larva and the sedentary juvenile. These two stages, which may either alternate or occur exclusively, include the benthic plantigrade (Carriker, 1961) and the planktonic postlarva (Sigurdsson et al., 1976)."

So, the distinction between plantigrade and postlarva is basically a behavioural /functional one. The post-larva is post-metamorphic, despite also being planktonic. These post-larvae re-enter the plankton by way of byssus drifting using specialised gossamer-like threads that are structurally different from normal attachment byssuses.

Plantigrades are of a similar size to the postlarvae described above. However they refer to mid- or post-metamorphic individuals not considered juveniles (juvenile size-range definitions vary a lot) that retain high levels of motility but are benthic, akin to your definition of plantigrade and postlarva combined, in the current study). Note that the term plantigrade is the norm for mussels, subdivided into early and late, based predominantly on whether they are primary settlers (away from adult conspecifics) or secondary settlers

(gregarious settlement in association with adults). Post-larva is used more generally and has different meanings for different taxa.

All that said, I agree that there are issues with consistency in the literature. The delineation between late plantigrades and small juveniles is also highly subjective, as you may probably discovered, since some use behavioural shifts / relocation of late plantigrades as the benchmark for being juvenile, others use onset of dissoconch growth (particularly in the deep sea, where we lack evidence for secondary settlement behaviour).

The convention you are using appears to be based on Cannuel et al 2008., who basically invented their own definition because of these ambiguities. Just be aware that they aren't very consistent, as an earlier study by a subset of the authors on gill development in the Pacific Oyster *Crassostrea gigas* refers to all post-settlement specimens as juveniles (Cannuel and Beninger 2006)!

<https://doi.org/10.1007/s00227-005-0228-6>

In deep-sea species, for the symbiont acquisition / developmental studies examining much smaller bathymodiolin (Idas-like) species, "juvenile" dissoconch shell development was used as a benchmark for being juvenile (shortly after which the first inflexion of gill lamellae was identified, but not the appearance of the ventral groove). Given the maximum size of your study species which is obviously much greater, I understand the choice to use later stages of gill development (i.e. appearance / formation the ventral groove) as a benchmark for being juvenile, as in Cannuel et al 2008. Just be aware that Shawn Arellano's work on Gigantidas – "Bathymodiolus" – childressi larval and reproductive biology also follows the "juvenile = dissoconch growth" staging convention.

In any case, I recommend that you re-examine the literature, as your stage definitions appear to be heavily influenced by Cannuel et al 2008, who deviate from standard terminology. Regardless of your ultimate decision, you need to relate what you mean by each stage to what has been used in the past and you should provide a brief explanation of the *biological* reasoning behind this decision, as it currently reads as arbitrary. An obvious one relates to your earlier benchmark of velum absorption: i.e. shifts in filter-feeding strategy from predominantly velar (ending with pediveliger settlement and metamorphosis), to DIC and foot-driven (early / late plantigrades) and ultimately gill-driven, once the ventral groove has formed (juvenile onwards)? I personally would discourage the use of post-larva as a developmental stage, and use it in the more general context of post-larval processes.

Line 170 "once the ventral groove of the gills is formed"

The biological significance of which is... (I know, you know, but the reading public may not)

Line 172 -180

It feels a little like the number of individuals used for shell measurements is being stated explicitly for impact, while the number of specimens examined for EVERYTHING ELSE is not stated at all, presumably because only a very low number of individuals was actually examined using the correlative approach technique?

You need to state here in the results or in the methods (of the main text) the number of individuals examined for each stage by each technique for each species. The lack of detail regarding this and the disorganised methodology must be improved if this MS is to be published in Proceedings B, or elsewhere

Line 190 palp or palps (they occur in pairs usually?)

Line 190 "digestive glands"

currently your 3D data and the labelling of your serial sections indicate you don't fully appreciate the extent of the digestive glands, and have identified the main digestive ducts only (see comments below)

Line 191-192 was the style actually visible in histological sections or CT data? Also, please remove "an" before anal.

Line 192 "epithelial cells of the stomach contained membrane-bound lipid vesicles"

I may be mistaken but based on my experience with similar species at similar developmental stages, these aren't actually in the stomach epithelium. The CT segmentation in figure 1 indicates that the inclusions lie outside the region of the stomach entirely and are in fact closer to (but not inside) what appear to be the main digestive ducts in your "stomach and digestive gland" surface mesh. This part of the CT visualisation does not appear to include all the remaining digestive gland tissue (composed of a dense convoluted arrangement of tubules called diverticula) that is associated with these ducts, which would occupy the same

space as the lipid inclusions (which I agree are likely to be vacuolar lipid stores, based on their appearance in serial sections).

My interpretation is that these inclusions are almost certainly spread throughout these as-yet-unidentified digestive diverticula that communicate with the main ducts and thus the stomach (both of which appear to have been segmented correctly). This is confirmed unequivocally in the supplementary animations where the areas occupied by lipid inclusions match exactly the expected areas that digestive diverticula would be found (please see Laming et al 2014 and 2015 for analogous examples in *Idas*-like mussels, but also the work of the deep-sea team based in the Azores for examples from adult *B. azoricus*).

Even in the toluidine-stained serial sections, there are examples where this "epithelium" would have to be unaccountable thick (aka it's actually developing digestive gland). Unfortunately it's not possible to see the fine-scale structure in these histological images to confirm this one way or another. In your defence, this oversight is understandable, as in my experience, digestive gland tissues typically display a less coherent histological arrangement in early post-settlement mussels. I also have experience with issues relating to thresholding of greyscale signal associated with digestive glands when using standard or synchrotron-amplified CT scanning techniques, even with suitable contrast staining. This makes segmentation a headache and I'm not surprised to find that the tissue surrounding the inclusions was not attributed a clear function.

However, this needs to be re-examined, with some greater care paid to the functional anatomy of this region. You need to either revisit the CT data and update the segmentation of the different tissues present (though this represents a lot of work as all the animations would need to be recreated too) or indicate clearly that the CT data only includes the stomach and main digestive ducts of the digestive gland and not the main mass (diverticula). Your supplementary tissue sections in which the digestive gland appears also need to be corrected as the labelling is not accurate (any tissue in this region that is not obviously the line of epithelial nuclei that surround the stomach lumen, in which the lipid stores appear). Finally, references to lipid inclusions stomach epithelial cells or digestive tract (you actually write both) must be changed.

For comparison, here are the main morphological features of *Mytilus edulis pediveliger* larvae.

Large velum for swimming and feeding

Ciliated palp that sorts food particles

A few pairs of gill filaments

Mouth, oesophagus, stomach with style sac and **large digestive gland**, simple intestine

Thin mantle that secretes shell

Foot used in crawling and byssus secretion

Cerebral, pedal and visceral ganglia, sensory pigment spots, pedal statocysts

from:

Gosling 2015. Reproduction, settlement and recruitment. *Marine Bivalve Molluscs*, 157–202.

doi:10.1002/9781119045212.ch5

Line 193 average = mean ? Also, you need a measure of variance around this value

Line 210/211 "These changes in gill morphology indicate a functional shift from a purely respiratory organ to a filter feeding and respiratory organ"

I don't agree with your reasoning here. Increased inter-filamentary space doesn't speak to a change in function but rather an increase in the animals size and resulting spread of filaments; a necessary process to accommodate the budding of new filaments posteriorly. In fact, filter feeding relies on relatively tight packed filaments aligned to one another in order to work efficiently, since the ventral-ward transport of particles is achieved by the action of opposing lateral cilia between filaments, before being directed anteriorly along the ventral border of the gill. Overly wide gaps would actually interfere with the fluid dynamics of this process.

Examination of live plantigrades in shallow water mussels confirms that at these sizes, gill coordination is poor, marked by random contractions rather than coordinated movement of cilia and that the ciliated foot plays a central role in current generation (I strongly recommend reading sections on larval-to-postlarval development found in "Marine Mussels: Their Ecology and Physiology" by Bayne, 1976 (see pages 94/95 in particular).

I can also confirm that even in juveniles of the wood-colonising mussel *Idas argenteus* the foot remains highly ciliated and these cilia remain active, though the role of particle transport to the mouth by this stage is assumed by the gills, which possess ventral grooves.

For an example video of particle transport in a bathymodiolin mussel (in this case, "Idas" simpsoni), please see Supplementary material 2 of the 2018 Review article on the Lifecycle ecology of Bathymodioline mussels.

available here: <https://www.frontiersin.org/articles/10.3389/fmars.2018.00282/full#supplementary-material>

Line 211 "digestive tract"

Alarming, these vesicles appear to have moved from the "stomach epithelium" to the "digestive tract"! However, as mentioned elsewhere my feeling is that they are in fact located in as-yet-unidentified digestive diverticula, though higher resolution histological images would be useful to confirm this.

Line 212-214

If these are average percentage values then measures of variance are needed, unless some are calculated from < 3 specimens, in which case it's more appropriate to provide percentage ranges. Regardless, the number of specimens examined for each stage should be stated

Also:

Why don't you use CT volume calculations more extensively in the main text? I eventually discovered in the supplementary materials that you had actually examined gut volumes for example but bafflingly this does not feature in the main text. For example you've missed a glaring opportunity to examine how relative gill-volume changes with development (as a proportion of total soft body volume) in the context of symbiont carrying capacity with increasing host size. This would be a welcome addition. Taken with a parallel calculation of the relative volume of the gut with increasing size, this would provide a more quantitative indication of the "importance" of each system in the trophic biology of the host throughout development and presumably provide support for some of your (valid) conclusions which relate to the shape and arrangement of the gut only.

Line 214 "no longer present in juveniles and adults"

This is a very nice finding! It's very satisfying to see the transition from lipid-supported heterotrophy, to mixotrophy, to predominantly chemosymbiotic nutritional modes.

Line 221 – 246

A lot of this section is methodology and does not belong in the results, except only to reiterate the method used (eg. see bold text)

Central to an accurate assessment of symbiont colonization and symbiont-mediated morphological changes was our combined approach of correlative SR μ CT, FISH, and light and microscopy (Video S5, Figure S10). This allowed us to rapidly screen whole animals yet achieve the resolution needed to identify the colonization of single eukaryotic cells by symbiotic bacteria. **We first searched for a morphological characteristic that was visible using light microscopy and reliably revealed the presence of symbionts in host cells. Previous studies [28, 29] showed that in juvenile and adult Bathymodiolus mussels, the morphology of epithelial cells colonized by symbionts is remodelled: i) The microvilli that cover all epithelial cells are lost (known as microvillar effacement), and ii) epithelial cells become hypertrophic (swollen) compared to aposymbiotic cells. We identified a third characteristic change in epithelial cells colonized by symbionts that has not received much attention, namely the loss of cilia (Figure 2 c, f and i). We tested if these three morphological characteristics had predictive power for symbiont colonization by analysing 1813 epithelial cells from a subset of six *B. puteoserpentis* individuals (2 pediveliger, 1 plantigrade, 3 post-larvae) and comparing LM images of these cells with their correlated TEM images (average cell count per sample 302).** Our analyses showed that all cells predicted to have symbionts in the LM dataset were indeed colonized in the TEM dataset, and likewise, all cells predicted to be aposymbiotic were free of symbionts (Figure S10). **We next used our verified morphological characters to reveal the onset of colonization in *B. puteoserpentis* in the subset of six mussel individuals.** All pediveliger cells were free of symbionts ($n = 797$ host cells in 2 pediveliger). In plantigrades 15% and in post-larvae 23% of all analysed gill, mantle, foot and retractor muscle epithelia cells were colonized by symbionts ($n = 336$ host cells in 1 plantigrade; $n = 680$ host cells in 3 post-larvae). **Finally, we expanded our analysis to LM analyses of serially sectioned individuals,** and these confirmed our results from the correlative dataset (all pediveligers were aposymbiotic, $n=5$; 50% of the plantigrades were colonized by symbionts, $n=6$; all post-larvae were colonized by symbionts, $n=5$).

Line 222 "symbiont-mediated"

Remove this term:

You do not provide evidence that the changes in the gill are symbiont mediated (though this is of course quite possible), just that the acquisition of symbionts coincides with the onset of hypertrophy and an apparent cessation in further gut development. The mechanism is not demonstrated, only an observation of the pattern.

Line 224-226

In my view this is the main strength of the MS.

Line 229

consider changing "remodelled" to something less suggestive of interior design e.g. "fundamentally altered"

Line 233 – 239

Credit, where credit is due: this is ingenious and a stand-out aspect of the MS. That said this is

One question: the example provided of the correlative light-electron microscopy approach in Figure S10 concerns the presence of MOX symbionts only (which is fine as they are always accompanied by SOX symbionts, right?). However, as you likely know, MOX are considerably larger than SOX (exemplified by your TEM images). Was this predictive power thus restricted to the presence of MOX symbionts and if so, how do you know that epithelial cells infected by SOX symbionts only haven't been overlooked by this extrapolative approach? In other words, is the degree of hypertrophy / loss of microvilli/cilia similar in cells colonised by SOX only comparable to those with SOX/MOX?

Line 239-240

These sorts of details must be also be provided in the methodology as mentioned above

Line 241 "In plantigrades"

change to "In the plantigrade measured"

(since n = 1)

Also if the percentage value for "post-larvae" is based on the three specimens, is it a mean? If so provide measure of variance or alternatively provide the range of percentages calculated for the three specimens

Line 250

Are color overlays drawn by eye based on bacteria ultrastructure I presume, rather than formally identified by a truly correlative approach (e.g. CLEM-based FISH?). Given the focus of correlative approaches in the MS, this needs to be clear.

Line 256 – 257 "Expanding our analyses to all three *Bathymodiolus* species, we found that the pediveligers of all three host species were aposymbiotic"

Here I finally get a sense of the approach used, but you don't explicitly express it. In fact, A lot of unspoken methodology appears to be alluded to in this one sentence. Effectively what you are saying, I presume, is that histological sections of the other two species were interrogated in the same way using the same LM cell morphologies, based on the validated method used in *B. puteoserpentis*?

Was this carried out manually, by eye or using a semi-automated, machine- learning-type approach?

Please correct me if I am wrong, but this suggests that no actual correlative microscopy with TEM, SRuCT or FISH was performed in the other two species, correct? This is not clear in the methods AT ALL, which is where this information needs to be detailed (in the main text, not the supplementary materials). It changes the interpretation of the entire MS and calls the reliability of your analyses for the other two species into doubt (particularly when this revelation is buried in the results).

If the above interpretation is correct, I have very genuine concerns about this extrapolative approach, in the absence of any cross validation using either (S)TEM or FISH in the remaining species. If you have carried out such validation, evidence of this needs to be included. If not, then the narrative of the MS must change to reflect the degree of confidence you have in the results for each species. I would also expect to see the word "putative" (or equivalent) being used, since these are inter-host inferences of symbiont distributions / acquisition patterns in multiple epithelia in each host, based solely (unless I've misunderstood) on comparative histology.

This is made more troubling by specific concerns:

For *B. azoricus*, while this comparative approach should work well, as the appearance of hypertrophied, non-ciliated, microvilli-devoid, epithelial cells in this species is likely to be very similar to that of its sister species, it has come to my attention that you may have used different tissue thicknesses and / or embedding procedures for this species (based on the tell-tale differences in tissue appearance in the Supplementary figures). This doesn't necessarily mean the proposed comparative approach won't work if carried out manually by eye. It does pose a problem if the approach is somehow automated.

For *G. childressi*, I see three possible issues. First, there are far fewer specimens used (I believe?), the section quality is either poorer or the staining is less differentiated in some sections (e.g. Figure S2) and the host species is less closely related and as a seep species, the relative proportions of MOX and SOX symbionts is likely to be heavily biased to MOX.

Can you provide assurances/evidence that these species have bacteria-colonised hypertrophied epithelial cells that are roughly equivalent to those found in *B. puteoserpentis* please. The whole-animal histological section comparisons provided in the Supplementary materials do not really demonstrate this well

Line 257-259

This is interesting, but I'm glad the speculation around this is cautious since I believe there are plenty of free living SOX bacteria at vents and seeps, with some evidence for free-living MOX bacteria in seeps (at vents it's less clear) and unless I have missed something in the literature, the positive identification of known symbiont OTUs in the environment, as free-living bacteria, remains elusive.

Line 265 Reference needed

Line 274 "The good preservation"

This sounds innately odd in English, as a native speaker. Consider changing to "superior". Also, I believe it was also the "better coverage of developmental stages" too, was it not?

Line 276

Please change to "... a few bacteria in gill, mantle and foot epithelial cells..."

Line 283-285

This has proven notoriously difficult to capture in still-image analyses in deep-sea species and is a credit to your sampling effort for this species. Well done!

Line 289-290

Worth providing a possible explanation as to why this might be in the discussion?

Line 292 Figure 3

Great FISH images. Question though:

Is there additional MOX signal in the centre of the foot in b, in a region where nuclei densities are lowest (i.e. potentially not epithelium, but more internal) and in the lumen of the stomach in a?

Line 303

"Bivalves" is a very large group of animals and this is actually taxon specific. It's well established that what works for staging oysters does not hold true for mussels, for example.

As alluded to already, in mussels and many other bivalves groups it's actually a combination of shell size and morphology and environmental context that's normally used to estimate developmental stage, in the absence of internal anatomy (i.e. relative proportions of PII and Dissoconch growth and – in mussels – whether plantigrades are settling away from, or among conspecific adults). However, in bathmodioline species, it's not yet known if primary and secondary settlement behaviours occur, as in shallow-water mussels. The Salerno paper you cite used shell size and the proportional composition of each shell type to classify specimens. It happens to use a different – somewhat simplistic but common – staging convention of assigning all specimens with juvenile dissoconch growth as being juveniles (there are several papers from Shawn Arellano's early-life stage studies on *G. childressi* which also adopt this definition of juvenile, e.g. <https://royalsocietypublishing.org/doi/10.1098/rspb.2013.3276>).

Your data demonstrate that there's a possibility that the "post-larvae" in the Salerno paper could have been pediveligers still in the process of settling (though a velum is a hard morphological trait to miss in a species with translucent valves) or early plantigrades following metamorphosis (i.e. what you term "plantigrades": this latter case is a very real possibility). Since you appear to be focused on the Salerno paper specifically

perhaps change this sentence to something that discusses the pitfalls of assigning early developmental stages based on shell lengths in deep-sea species:

"Our study shows that the use of shell characteristics alone as a means to determine the developmental stage of deep-sea mussels (e.g. Salerno et al 2005) cannot be relied upon, particularly for shell lengths typical of settling pediveligers and [early] plantigrades."

Line 306-309

This is actually common in shallow water mussels too, as you allude to in the next sentence. It is well known for other bivalve larvae / postlarvae as well (including *Mytilus*), because once competent to settle, larval size at settlement is entirely dependent on planktonic growth, in turn largely determined by transport time in the water column (with an unquantifiable food-supply component, obviously). Longer transport times are expected when habitat availability is poor (later in the season for coastal species). In fact quite extreme examples are now known to exist even in coastal species, which can, for example, lead to differences in PII at settlement of **122 microns** between the smallest and largest seasonal *means* for postlarval plantigrades, translating to a 300% or 3-fold difference in estimated masses, based on these seasonal means for *Prodissoconch II*, in Martel et al 2014
<https://doi.org/10.2983/035.033.0213>

Line 308

please change "...dependent and that these mussels are able to delay their metamorphosis while continuing to grow."

to something along the lines of

"...dependent, so providing further evidence of a capacity to delay metamorphosis while continuing to grow, as has been previously suggested for *G. chidressi*, based on larval growth experiments (Arellano et al 2009*).

* <https://www.journals.uchicago.edu/doi/10.1086/BBLv216n2p149>

Line 310

A more up-to-date reference on this (mentioned above) is the Martel ref, which should be included here. The Bayne et al 1965 reference demonstrates a capacity to delay metamorphosis under experimental conditions. Martel et al 2014 demonstrate that this is a real phenomenon and has a bearing on seasonal recruitment patterns.

Line 312

A good reference here would be Young et al 2012*, a dispersal modelling paper ground-truthed by larval developmental data on *Gigantidas chidressi* from Shawn Arellano's PhD work

* <https://doi.org/10.1093/icb/ics090>

Line 318

The vesicles are happy with their new home...

Elsewhere in the MS you say they are in the "stomach epithelium". Again, I believe its neither, but rather the main body of the digestive gland, which is currently not presented.

Line 320 change to "fuel energy-demanding metamorphoses"
metamorphosis isn't really sustained...

Line 326

An obvious comparison to made here is between *Bathymodiolus*/*Gigantidas* and other members of the *Bathymodiolinae*, instead of (or at least, in addition to) more distant shallow-water relatives. Laming et al 2014 and Laming et al 2015 describe ontogenetic patterns of development AND the acquisition of symbionts in two species, *Idas modiolaeformis*, and "*Idas*" *simpsoni**. Discussing how ontogenetic development and symbiont acquisition in the genera *Bathymodiolus* and *Gigantidas* compare with *Idas* or *Idas*-like mussels for which similar data are available would make for a much more meaningful discussion. It would provide an opportunity to speculate on the evolution of symbioses in bathymodioline mussels under various reducing conditions for example.

* Note: a putative new genus for this species, *Nypamodiolus*, has been proposed by Justine Thubaut and colleagues in 2013 but the formal description remains as-yet unpublished

Consider the differences:

1) Bathymodiolus / Gigantidas have large body sizes, are highly productive, live on highly energetic vents and seeps with much greater supply of reduced compounds (higher concentrations and greater longevity), possess intracellular symbioses with acquisition as plantigrades, and ultimately have a highly reduced linear gut

2) Idas (sensu lato) have much smaller sizes, are somewhat less productive, can live on seeps but are predominantly found on organic falls which are a more ephemeral with markedly lower sulphide concentrations, possess extracellular symbioses (less "evolved", but well adapted to this habitat?) and retain one to two loops in the gut (i.e. more than species in the current study's but less than true, heterotrophic mussels).

Line 326-328

The concomitant onset of gill hypertrophy in bathymodioline mussels and acquisition of symbionts, has also been described in *Idas modiolaeformis* by Laming et al 2014:

"...**hypertrophied gills**, a style sac and a recurrent loop (identified during dissection) in the microvilli-lined intestinal tract **were first recorded with certainty in Idas 4 (0.57 mm) and then in all larger individuals examined with histology...**" and "**The smallest SL at which putative bacteria were identified was 0.59 mm (Idas 7, by TEM, Fig. 5l, m) arranged in a thin extracellular monolayer only.**"

So, the appearance of bacteria in the gills of these smaller mussels also coincides with the onset hypertrophy. In that study, it wasn't clear whether the onset of hypertrophy paved the way for bacterial colonisation, or whether bacterial colonisation acted as a cue, initiating cellular hypertrophy in the gills (as in your hypothesis). While the data you present clearly indicates that hypertrophy and bacterial colonisation are tightly linked spatially and temporally, occurring on a cell-by-cell basis, no evidence is provided herein of the symbionts direct involvement as a cue or a mediator (or any other terms that suggest the presence of symbionts is a necessary prerequisite for the cytological / morphological changes to occur), though this is a very plausible working hypothesis for future manipulative experiments. So you need to be less deterministic when discussing this as it remains a hypothesis.

Line 328

maybe less wordy to say "The loss of cilia and microvilli during hypertrophy has been reported for many bacteria invading a wide range of epithelia"

Or at least change "remodelling" to "adaptation"

line 332

The near-linear rudimentary digestive system of adults is well known and pervasive in large bathymodioline mussels. What is new is that you provide evidence for a transition from an apparently fully functional gut in plantigrades and post larvae to a simple gut in young adults

Therefore you need to include previous knowledge on the adult gut

"Although it has been known for several decades that the symbionts of *Bathymodiolus* supply them with nutrition **and that adults possess a rudimentary gut only**, nothing was known about the development of the digestive system in these hosts.

Line 338

Does the gut not continue to develop and thus become more convoluted in *Mytilus*?

Also, reference please.

Line 339

I dislike the use of the words "remodel" or "streamlining", they are not particularly biological and while the occasional use of such words can be effective from a scientific communication perspective, overuse is not to be encouraged. Also in this context they are not particularly accurate. "Remodelling" suggests that the gut is actively adapted in the later stages of development (which I would argue against, for energy budget reasons, see below), while "streamlining" suggests the gut is somehow improved or made more efficient, which is of course the opposite of the reality.

In my opinion, the most parsimonious explanation is that this change is as a result of the cessation in further gut development (rather than anatomical rearrangement), and that the straightening of the gut is actually occurring passively, the intestinal loop being unfolded as the mussel increases substantially in length. I say

this because this would require minimal energy investment in comparison to a cryptic metamorphic process which would be energetically demanding, and because growth in mussels is posteriorly directed which would act to elongate and straighten this region of the gut.

Line 340-341 “Conspicuously, in *Bathymodiolus* this remodelling of the digestive system did not coincide with the first stages of symbiont colonization or metamorphosis, but rather occurred during the transition from post-larvae to the juvenile stage, well after metamorphosis...”

Well, the change takes place at different stages depending on which naming convention is used, so I'm not sold on this "transition to juveniles" argument. Better to say that it coincides with a notable increase in animal size, as mentioned above.

It *is* interesting that the cessation in gut development and establishment of symbiosis in the gill are concomitant processes: discovering how this is coordinated and to what degree this is mediated by the host and (presumably) influenced by the presence of bacteria would be a great leap forward for the field. Of course the ideal would be a squid-*Vibrio*-type experiment that tracked ontogenetic development after settlement in under the presence and absence of symbiont candidates to assess whether the acquisition process is an obligatory prerequisite for host development and if so, is their role simple to provide a cue for host signaling or are they mediating changes in the gill and/or whether these changes ultimately are required for gut development to stop. Sadly, such an experiment would face considerable obstacles, given the obvious technical and culturing constraints (both for the host and symbiont).

Also, remove “Conspicuously”, it is unnecessary.

Line 342-343 “We hypothesize that the streamlining of the digestive system is induced by the shift in nutrition of these hosts”

Hypothesising is all you can do for now but for what it's worth, it's a sound hypothesis in my opinion (though as I have mentioned, for me this would be a relatively passive process). The alternative, which is that the straightening of the gut is somehow genetically "hardwired" (now I'm using non-biological terms!), seems unlikely, given the relatively recent evolution of this group.

Also, the word streamlining is counter-intuitive

Line 346 – 349

you are “postulating” a scenario that has already been proposed before and yet it is presented here as a novel idea. I must therefore **insist** that you mention previous studies that have 1) already stated exactly this for bathymodiolins generally (Laming et al 2018) and 2) demonstrated it in smaller bathymodioline mussels empirically (i.e. Laming et al 2014; 2015).

To save you the time looking and for the benefit of the editor, I'll quote the various papers concerned

Laming et al 2014 on *Idas modiolaeformis*:

“... isolating the probable period of acquisition to the post-settlement phase of development. The **absence of bacteria in plantigrades, whose digestive system is well developed, indicates that planktotrophic larvae and post-larval plantigrades are both exclusively heterotrophic ...**

...This allometric [gill] development would increase the symbiont carrying capacity considerably. It is unknown whether heterotrophic nutrition remains important at this stage; however, the apparent **increase in the relative volumes of gill tissue versus digestive tissue** in larger specimens of the same species found upon carbonate crust (Gaudron et al. 2012) may indicate a **switch in nutritional mode from heterotrophic to predominantly chemosymbiotic with increasing size.**”

Laming et al 2015 on "*Idas simpsoni*:"

"Considering bacterial signals were **entirely absent from all epithelia in plantigrades of "I." simpsoni**, chemosymbiotic larval nutrition seems unlikely, though the pre-acquisition of a very low seeding population of bacteria cannot be entirely ruled out. Thus the **larvae of "I." simpsoni** are very likely to be **obligate heterotrophs**.

...Within a narrow range of SLs following their initial appearance in very small juveniles, symbionts in the current study appear to increase rapidly in abundance on the external, non-ciliated surfaces of gill epithelia reaching high abundances based on microscope observations (e.g. Fig. 4; Fig. 6B even at SLs 846 and 974 mm)..."

Then, since these mussels are subject to lower sulphide emissions:

“As young juveniles, the presence of a developing gut and a growing number of bacteria associated with the gill suggests a potential to ingest and assimilate particulate or dissolved organic matter as well as utilise possible symbiont-based nutrition...

...If the SOX symbionts found in specimens of “*I. simpsoni*” in the current study are indeed assimilated as a source of carbon and energy, then in highly sulphidic conditions the importance of an anatomically functional gut may be reduced. However, the release of reduced chemical compounds at reducing habitats is often highly heterogeneous (e.g. Le Bris et al., 2008), wherein mixotrophy involving filter-feeding and chemosynthetic symbiont nutrition might provide a means to ensure that a continuous energy resource remains available.”

and finally from the Laming et al 2018 review

"Adult bathymodiolins of most species rely on their bacterial partners for much of their nutrition, with some species retaining rudimentary digestive systems only. That said, growing evidence points to a retained capacity to filter feed (Page et al., 1991). Studies in aquaria have demonstrated the assimilation of carbon from algae, radio-labeled water-borne bacteria, and naturally-occurring plankton (Pile and Young, 1999). The retention of a fully-formed digestive system appears to be the norm in smaller-sized bathymodiolins from organic falls (Gustafson et al., 1998; Thubaut et al., 2013a; Laming et al., 2014, 2015a). In *I. modiolaeformis* and “*I. simpsoni*”, live examinations under a dissecting microscope reveal the mediated convection of debris in the vicinity of the gills in a way similar to that of shallow-water mussels (Supplementary Material 2). Given the earliest, post-settlement developmental stages of these species possess components of a functioning digestive system and are either aposymbiotic or display low abundance of symbiotic bacteria, the diets of juvenile bathymodiolins are likely to undergo a transition from heterotrophy-to-mixotrophy-to-chemosymbiosis with increasing size. This is supported by a model-based study examining *B. azoricus* from the relatively shallow hydrothermal vent site Menez Gwen (800 m), in which a shift toward greater reliance on symbionts during the growth of the host was hypothesized (Martins et al., 2008). Similar transitions are thought to take place in other chemosymbiotic molluscs (e.g., *Gigantopelta* spp., Chen et al., 2017)."

To completely ignore these studies findings and “postulate” a scenario that has already been proposed by both these and other studies (see also the output from the Deep-sea lab in the Azores which has produced several live-animal studies on trophic biology and immunology in *B. azoricus*) some five years ago, is an oversight at best, bordering on unethical.

Line 350

Consider changing “evolutionary strategy to reduce the energy” to “evolutionary adaptation to minimise the energy”. The term strategy in this context is suggestive of a master plan.

Line 353-355

yet again, smaller bathymodiolins from other reducing systems are not discussed. Aside from the aforementioned Laming et al studies that include data on retained looped intestines (one in *I. modiolaeformis* and two in “*I. simpsoni*”) there is also *I. iwaotakii* from wood (Thubaut et al 2013b <https://doi.org/10.1371/journal.pone.0069680>) and *Tamu fisheri* from seeps (Gustafson et al 1998 <https://bit.ly/341Mvaz>).

Line 355-357

Indeed, and this is also true for organic-fall species as organic falls are likely to occur in regions that see a greater overall flux of detritus, such as submarine canyons where the retention of a more functional looking gut (anatomically) could reflect active, ongoing selective pressures (i.e. a gut remains a requirement for survival because mussels are more mixotrophic), particularly if symbiont productivity is subject to intermittent or limited supplies of reduced-fluid emissions..

Line 357-359 This is the exactly the question that needs answering, as I mention above (i.e genetic predisposition *versus* environmental factors, even epigenesis?)

Line 361 Figure 4

I'm curious, what was the inspiration of the schematic in (e)? Maybe it was simply "convergent inspiration" but it feels **very** familiar...

For the editor:

https://www.frontiersin.org/files/Articles/385457/fmars-05-00282-HTML-r1/image_m/fmars-05-00282-g001.jpg

Developed from the schematic found on page 91 of the following PhD thesis:

https://tel.archives-ouvertes.fr/tel-01135209/file/pdf2star-1417090206-These_archivage_3160120.pdf

Line 369

You've overlooked the last author: please change to "Laming, Gaudron **and Duperron** [7],"

Line 379-380

Particle sorting is not coordinated by the gill when they are only rudimentary gill bars. This is performed by the highly ciliated foot.

Line 388-389

Sound reasoning and I would definitely agree.

However here, there is **yet another missed opportunity** to make comparisons with other reducing habitats characterised by lower reducing fluid emissions. Acquisition in these species of large bathymodioline is even more rapid than the already relatively fast acquisition of symbionts by "Idas" simpsoni on bones, which in turn was more rapid than acquisition in Idas modiolaeformis on wood falls.

This could be driven by the availability of proximate symbiont partners in the environment, likely influenced by the availability of reducing fluids (for free living candidates) and/or the densities of proximate compatible symbiotic species (mainly through lateral transmission from conspecifics). Consider: at vents there are both high emission rates and high densities of conspecifics, at carcass falls there are lower emission rates but still high densities of conspecifics, while on wood falls both emissions and conspecific densities are low.

Line 390-392

maybe add something about the availability of symbionts locally?

e.g. " This **is made possible through the immediate availability of symbiont candidates and** could reflect..." or something similar.

Line 400-401

Have you any sense as to why this might be and/or why MOX do not colonise cells independently?

Line 403/404 "early planktonic larval stages"

Please rephrase, as this reads like they are an early stage in planktonic development when they are of course the last stage.

Line 406-408

This is perfectly possible. However, it's worth noting that this would likely operate over fairly small spatial scales. The study you cite is also for a tubeworm but there is already evidence that mussel larvae employ vibrotaxis, chemotaxis with neurochemical signalling and respond to types of current flow (turbulent versus laminar) and substratum characteristics. Vents are likely to have a very particular soundscape and strong chemical and temperature cues from plume dispersion.

Line 409-410

As I have already mentioned repeatedly, an obvious opportunity to make comparisons between bathymodioline mussels from different reducing environments that are more or less chemosynthetically productive has been missed, despite the fact that this would enrich the discussion.

Line 412-414

Spatial scales yes. Temporal scales: not so much. The MAR is a slow spreading ridge and is not as ephemeral as has been postulated for vents in fast-spreading regions such as the EPR.

Also not "selective advantage" but "fitness advantage".

Line 421

Change to "are [*thought / believed*] to be triggered"

Line 429

remove "interwoven" and perhaps replace with "intrinsic" (or similar)

Finally:

Line 432 Supplementary materials

Some corrections are suggested below for the supplementary material:

First, the anatomical inaccuracies concerning the digestive glands in the CT data and labelling errors in the serial sections **must** be addressed.

Also, please use the term "commissures" not "commissure tissues" and "connectives" not "connective tissues". Strict terminology exists for neural biology: connective tissues are something different from what you are describing, which are connectives (a particular type of neurite bundle). For correct terminology in neurobiology see Richter et al 2010

<http://dx.doi.org/10.1186/1742-9994-7-29>

THE BAD NEWS: these terms are also incorrect in the CT 3D animations as well, so these annotations needs correcting, sorry!

Correct all other mentions of the lipid vesicle locations, as in main text.

In Figure S5:

Are the purple and white glands those that are visible at the base of the foot in these sections, abutting the pedal ganglion? If so, please label them. Also please see additional comments/annotation regarding digestive gland/divertula

In Figure S12

The composite image for b is flipped in it's horizontal axis for some reason...

In Figure S13

Caption states "The gill filaments of a *B. putose* plantigrade (c) is only colonized by SOX. a.a.m, anterior adductor muscle; ft, foot; gf, gill filament and m, mantle"

So, can you explain what I'm seeing then in the MOX image for c), because it doesn't look like background signal to me.

More generally, although self-evident to me (being very familiar with the techniques used in the study), the supplementary materials could benefit from a brief description of the data format (e.g. "tiff image stack") and the means by which such files are best viewed, as a reader not familiar with such image analyses and handling of files may struggle to view the supplementary material.

Appendix B

In their study, Franke et al. investigate symbiont acquisition in three deep-sea *Bathymodiolus* mussel species by means of various microscopic and imaging methods, including synchrotron radiation-based microtomography (SR μ CT). The authors show that the mussels are colonized by their chemoautotrophic symbionts during the plantigrade stage of larval development. Moreover, the study presents first time evidence that the earliest larval stages of *Bathymodiolus* mussels, pediveliger larvae, are symbiont-free. This is an important new finding which finally substantiates the previous assumption that *Bathymodiolus* symbionts are horizontally transmitted (i.e., acquired from the environment) rather than inherited from parents to offspring. The results thus contribute another valuable facet to our understanding of how marine invertebrate symbioses are established.

The paper is very well written, I enjoyed reading it. Images and particularly the supplementary videos are of high quality and present a real asset for the reader. The study has a clear research focus and the methods were chosen adequately to answer the research question. In the discussion, the findings are very plausibly interpreted and integrated into the bigger picture, i.e., mussel development and metamorphosis, mode of nutrition, and uptake of locally adapted symbionts. This work provides the basis for a number of interesting follow-up studies.

I have only a few minor comments and questions:

- Replicates and mussel species: In the sampling methods section I missed how many replicates were sampled of each mussel species. This important information is given later in the results, but could also be mentioned here (or at least state something like “refer to the results for replicate numbers”). You examined only 6 *B. childressi* specimens from the Gulf of Mexico, while all other > 250 individuals were from the MAR. This doesn’t really allow for conclusions on differences or similarities between mussels from both sites (and you are not attempting to draw such conclusions) – but this would be an interesting question for future studies with more replicates: Are there differences in symbiont acquisition between *Bathymodiolus* species from geographically distinct locations? (You could mention this in the outlook.)
- Biological replicates and symbiont colonization: Your finding that pediveliger larvae are aposymbiotic is based on two individuals (page 12, line 241) + 5 individuals (line 245), is that right? Shouldn’t this be made more transparent, considering that the aposymbiotic pediveligers are the most central and novel finding of this study? Also, when you say “In plantigrades 15% ... of ... cells were colonized by symbionts” (page 12, line 241-242), which means “all plantigrades had symbionts”, this is a bit misleading because it is based on only one individual, and seems to contradict the next sentence, where all 5 plantigrades were aposymbiotic. Please rephrase to make this clearer.
- Mode of nutrition and gill function: You could go into a bit more detail when explaining the proposed shift of functions of gills and digestive tract during morphological changes. How do the larvae feed before they start filter feeding? On page 11, line 211 you suggest that gills in plantigrades develop from purely respiratory organs to respiratory and filter feeding organs. But on page 18, line 346-348 you postulate that “the first life stages of *Bathymodiolus* rely on filter feeding”. That’s a bit contradictory. Also, when you mention that lipid vesicles serve as storage compounds (page 17, line 320) – does this mean these larvae only feed on internal lipid stores? Please provide some more explanation.
- Bacteria on larval shells (page 21, line 406): I assume that these bacteria were only visible in TEM and not identified as SOX and MOX by FISH? Why?

- Page 14, line 256: “Expanding our analyses to all three *Bathymodiolus* species...” – referencing Figure 2 in this sentence is misleading, since the figure only shows *B. puteoserpentis* (not all three species).
- Figure 4: I suggest to remove the background (a hydrothermal vent site?), because it is poorly discernable and diminishes readability of the illustration and text on the lower right
- Check for typos, spelling, etc., e.g.,
 - Abstract, line 33: period missing
 - Methods, line 120: 16S rRNA; line 146: “[22-25]van...”
 - Results, page 14, line 276: it should read “Figure 2b, e, h, j-m”
 - Figure 1 caption, line 201: “Note the different scale bars in a-c and d” is a bit confusing. Why not just “Note the different scale bars”?
 - Figure 3 caption, line 296: host tissue (singular)
 - Page 20, line 369: check Proceedings B reference format
 - Conclusion and outlook: Not sure if this is on purpose, but you are using “revealed” three times in the first two sentences. Might want to consider rephrasing.
 - Throughout: check consistency of your British English (colonization, visualization)

Appendix C

EDITOR COMMENT

Your manuscript has now been peer reviewed and the reviews have been assessed by an Associate Editor. The reviewers' comments (not including confidential comments to the Editor) and the comments from the Associate Editor are included at the end of this email for your reference. As you will see, the reviewers and the Editors have raised some concerns with your manuscript and we would like to invite you to revise your manuscript to address them.

Editor comments:

We have now obtained three expert reviews of your manuscript, and I am happy to tell you that all three reviewers liked your paper. My own reading of your manuscript corroborates the reviews: this is an interesting piece of work that could become publishable in Proceedings B.

However, although the reviews were generally very positive, one of the reviewers came up with a long list of insightful and useful suggestions, which should lead to an improved manuscript. The comments by the other two reviewers are fewer and perhaps easier to deal with, but they too would likely improve your paper. Briefly, all reviewers felt that the methods section could contain more detail in places, including a clearer description of the specimen handling workflow, and in some cases methods that are described in the results would be more suitable in the methods section. The methods and results surrounding replicates could be better explained. The discussion could be more developed with a fuller explanation of the proposed model. One reviewer also noted that the study organism should be considered two genera instead of one; if this is the case, the implications of this should be addressed. This reviewer also points out that more precise references should be made to original studies rather than review articles.

General answer to all reviews and the editor

We thank the reviewers and editor for their insightful and constructive suggestions on how to improve our manuscript originally titled “*Coming together – symbiont acquisition and early development in deep-sea Bathymodiolus mussels*”. In our revision, we addressed the reviewers' and editor's comments point by point, for details please see below.

We are very grateful for the extensive time and effort the reviewers and editor invested in our manuscript. We hope you will find that your comments, questions and suggestions helped us improve our manuscript. After addressing the reviewers' requests for more details on our methods, and their suggestions to include additional points in our discussion, we submitted our revised manuscript, but were informed that it exceeded the limits of Proceedings B by 1.5 pages or about 1300 words. We therefore had to move parts of the Methods to the Supplement, but kept critical information on specimen numbers and imaging approaches in the main text, including a table (Table 1) showing how many specimens were analysed using which method.

Maximilian Franke

Nikolaus Leisch

Nicole Dubilier

Referee 1

General observations

The manuscript has many strengths. Stand out among these are 1) its enterprising use of size-stratified series of specimens in three species of "Bathymodiolus" mussels, which is extremely rare in deep-sea biology (and, as the authors note, also highly fortuitous) and 2) the intelligent use of correlative approaches that allow some comparisons to be made across techniques. Sadly, a seriously underdeveloped discussion, a confused and at times vague methodology which lacks transparency and a narrative that infers results to be unprecedented when in places they are not, undermines the ingenious, intelligent use of samples. That said, I think the MS has the potential to be an outstanding piece of research. Several issues first need to be addressed.

1) The MS repeatedly focuses on what is and is not known about Bathymodiolus mussels. I'd like to draw attention to the fact that you are speaking of Bathymodiolus as if it were a monophyletic genus. Sadly this is not the case. Until quite recently, it was of course polyphyletic (or paraphyletic, depending on your point of view) in that it included two genera. Putative "Bathymodiolus" spp. that strictly belong in Gigantidas, are now accepted as such, in the face of overwhelming evidence that points to this. Some care is needed with this as the nomenclatural issues underlie very real divergent evolutionary histories (albeit relatively recent). This has ramifications for the discussion as on the one hand, you have two sister species capable of hybridisation (and in that situation, they possess phylogenetically indistinct symbionts), and on the other hand, a species from a separate genus, Gigantidas childressi. If you are concerned about continuity, you could simply state that you are deviating from the use of "Bathymodiolus" as the genus, since the species is now considered to be from the genus Gigantidas (see WORMS database) Also, changing the species name only in the MS is not sufficient, as this would fail to address the degree of relatedness among the three species, which is highly relevant to the ecology of symbiosis. Finally, please note that this problem also applies to the extensive supplementary materials so please ensure you make changes here as well

We agree with the reviewer and have changed the species name accordingly throughout the manuscript, please see below under comment "line 25" for our detailed responses.

2) There are too many instances where review articles are used to support quite specific statements. It's considered poor form to use a review, when the original studies discussed therein are more appropriate as references, since the original authors work is not given due credit. For example:

The role of symbionts in inducing developmental changes in the host in the squid-Vibrio symbiosis was described in 1994 and should be cited accordingly (<https://doi.org/10.1242/dev.120.7.1719>), rather than using a review paper published much later (McFall-Ngai 2014 reference, currently number 5 in the MS reference list) There are even instances where review papers that broadly deal with the appropriate subject area are used to support statements for which no evidence exists within said review (as in the Bright & Bulgheresi review cited in Line 55/56). This is lax, at best, and should be avoided. *Montgomery and McFall-Ngai 1994 <https://doi.org/10.1242/dev.120.7.1719>*

Point well taken! We have incorporated the advice from the reviewer as outlined in detail below under comment "line 55-56".

3) *Methods*: The methods are not clear regarding the workflow employed (particularly the order) for a given specimen fixation and could benefit from being reordered into paragraphs that follow the analytical timeline, rather than paragraphs grouping vaguely similar methods together. First, you need to define the term “morphological analyses” more explicitly for each fixation approach please. It may seem clear to you how the analyses were performed and in what order and upon which specimens for a given fixation. However, as a researcher who is very familiar with all the techniques individually and some (other) correlative approaches, I still find myself trying to piece together the order of analyses and decipher which specimens were subjected to what treatment. I know you have just put out a preprint on the “chemohisto- tomography” approach but the method here should be clearly formulated and only supported by the considerable supplementary materials that accompany this MS rather than being dependent on them. Another person needs to at least be able to follow the logic of the analysis. This is currently not possible. More serious still, fundamental methodological details concerning what analyses were carried out on which species that seriously influence the interpretation of your results are buried in the results section (see detailed comments) and the number of analyses in a given species is not detailed clearly. In short the MS lacks transparency and borders on misinformation at times.

We agree that our paper would profit from making our workflow clearer. We have introduced a new section describing the correlative workflow (see lines 178 – 191 in the revised manuscript), clarified parts of the methodology and have added the number of specimens and replication throughout the manuscript. Some parts have been moved from Results to Methods section, as suggested. For details, please see below under the comments “line 117, line 172-180 and line 256-257”.

Specific comments

The following comments deal with specific issues (and positive notes) in the MS. It is meant to be constructive and hope that the detail provided helps (I don't do short answers well). Disclaimer: you will probably have a good sense of the identity of this reviewer by the end.

Line 25

“three deep-sea Bathymodiolus species” The MS needs to be restructured in order to acknowledge that you are not looking at Bathymodiolus species exclusively, see later comments on this.

We agree with the reviewer and have changed the genus name of *Bathymodiolus childressi* to *Gigantidas childressi* throughout the manuscript and distinguished between the two habitats these genera come from. We have changed the text as follows: “Here, we collected early developmental stages of three deep-sea bathymodioline species from different habitats to identify when these acquire their symbionts and how their body plan adapts to a symbiotic lifestyle”.(Lines 24 – 27 in revised manuscript)

Line 32/33

The colonisation of other epithelia in Bathymodiolus spp. juveniles following the establishment of symbioses is also already documented (Wentrup et al 2013). It's the timing of acquisition

and the fact that gill tissues are among the first to be colonised rather than the last, that is novel to this study.

We agree and have reworded the sentence to emphasize the timing of acquisition: "*Symbiont colonisation began during metamorphosis from a planktonic to a benthic lifestyle, with the symbionts rapidly colonizing first the gills, the symbiotic organ of adults, followed by all other epithelia of their hosts.*" (Line 31 in revised manuscript)

*Line 33
a full stop is missing*

Change made.

*Line 33/34
"...symbiont densities in post-larvae reached those of adults" Careful.... In mytilids, post-larva and plantigrade are morphologically synonymous (but not functionally synonymous). They are terms used to describe individuals settling for the first time but employing markedly different locomotor behaviour. Early and Late plantigrades are terms to describe post-settlement individuals that have settled for the first time and the second time respectively (the latter, among adult conspecifics). Do you mean to define all post-settlement specimens that have yet to form a ventral groove as post-larvae ? I presume some of these individuals display dissoconch growth. Note that in deep-sea species, this is often considered the "juvenile dissoconch". If you have chosen to label all individuals that do not have a ventral groove on their gill as post-larvae, it flies in the face of many years of study on mussels, which use different staging conventions.*

We agree that definitions of developmental stages vary considerably in the literature. We thought it best to use terms that are widely used in one of the best-studied bivalves, the shallow-water mytilid *M. edulis*. We followed the reviewer's suggestions and removed post-larvae from our classifications, calling it now a plantigrade, and have updated our definitions: "*Because the names for larval stages of bivalves have not always been used consistently, we define them as follows: The earliest life stages in our study were at the last planktonic larval stage - the pediveliger. Once settled on the seafloor, the animal initiates its metamorphosis from a planktonic to a benthic lifestyle, and we refer to this stage as being in metamorphosis. While metamorphosing, the mussel degrades its velum, the larval feeding and swimming organ, and develops into a plantigrade [28]. During the plantigrade stage, the mussel secretes the adult shell and once the ventral groove of the gills, which transports particles to the labial palps, is formed, it enters the juvenile stage [29]. When the gonads are developed, the mussel is considered an adult [18].*" (Lines 140 – 149 in revised manuscript)

*Line 35
"This morphological change appears to be specific to Bathymodiolus" DO you mean within the mytilids or more generally? Be aware that similar morphological shifts have already been documented in Peltospirid gastropods as well.*

We have clarified this and limit this statement to mytilid mussels: "*Within the Mytilidae this morphological change appears to be specific to Bathymodiolus and Gigantidas and is likely linked to*

the decrease in importance of filter-feeding when these mussels switch to gaining their nutrition largely from their symbionts.” (Line 35 in revised manuscript)

Line 36/37

Indeed, though whether the change is mediated by the host or the symbionts still remains unclear.

Inferences to the contrary (e.g. bacteria inducing the cryptic metamorphosis in bathymodiolus juvenile from a functional gut and heterotrophy, to a linear ruduced gut and chemosymbiosis) can only ever be speculative without targeted experiments employing symbiont exposure versus aseptic conditions, akin to the squidvibrio symbiosis (I revisit this again later).

We agree with the reviewer that at this point we cannot judge which changes are mediated by the host and which by the symbionts.

Line 40/41

“By associating with microorganisms, animals can expand their metabolic capabilities and colonize habitats they could not live in on their own”

This is not accurate.

*First, not to be pedantic but the host's metabolic capabilities remain unchanged. Instead they benefit from the alternative metabolic capabilities of their symbionts and access to an efficient means of acquiring energy and a source of organic carbon. Second, engaging in symbioses does not "allow animals to colonise habitats they could not live in on their own". Instead, it typically provides a competitive fitness advantage, be it through: access to an energy source that would otherwise be inaccessible directly; the detoxification of a habitat; the realisation of a new (or augmentation of an existing) host trait; or several of these factors. It typically also allows increased somatic and reproductive growth (large sizes, high reproductive output). So the only way in which it aids in the colonisation of habitat is by increased larval output. In reducing environments, where the resource in question is clearly nutritional, symbioses allow host organisms to survive – or indeed thrive – in a habitat where availability of organic carbon for nutrition is otherwise limiting and critically, through efficient assimilation of and reliably access to a sustained food source without **expending energy on active feeding** (e.g. bacterial mats / biofilms, or in the case of a filter feeders POC as bacterial flocculates for example), providing a clear fitnesses advantage over heterotrophic feeding strategies (while potentially also allowing the host species to tolerate otherwise toxic habitat conditions). As a side note, the use of the word colonise here is also particularly counter intuitive for horizontally transmitted symbionts, since this often occurs in the absence of symbiont partners... (e.g. this study!). What allows animals to colonise a habitat is distinct from what allows them to survive/thrive in said habitat: In other words larval supply/settlement versus post-settlement mortality.*

Thus I'd ask that you be clearer about how you convey your message here please.

We followed the reviewer's suggestion and changed the sentence to the following: *“By associating with microbial symbionts, animals benefit from the metabolic capabilities of their symbionts and gain fitness advantages that allow them thrive in habitats they could not live in on their own [1].”* (Line 41 in revised manuscript)

Line 42

“bathymodiolin”

Not sure if this is a typo, but bathymodiolin is the noun, bathymodioline is the adjective. In other words it's either "bathymodiolins" or "bathymodioline mussels"

Thank you for pointing this out. We have changed this throughout the manuscript.

Line 43

Bathymodiolus

same issue as above (do you mean sensu stricto or sensu lato?)

We have revised this accordingly and it now reads *“Mussels of the genera Bathymodiolus and Gigantidas house chemosynthetic bacteria in their gills, in cells called bacteriocytes [2].”* (Line 45 in revised manuscript)

Line 45/46

“the bacteria use reduced compounds in the vent and seep fluids to gain energy for the fixation of carbon and provide nutrition to their hosts” consider changing to “the bacteria use reduced compounds in the vent and seep fluids as an energy source for carbon fixation, which in turn provides nutrition to their hosts” The sentence as it currently stands suggests that bacteria actively seek to provide energy to their hosts.

We have changed the text according to the reviewer's suggestion: *“In these nutritional symbioses, the bacteria use reduced compounds in the vent and seep fluids as an energy source for carbon fixation, which in turn provides nutrition to their hosts.”* (Line 46 in revised manuscript)

Lines 50 – 55

Any chance we could have some references to support these statements please?

We provided references and an example for vertical symbiont transmission in a marine invertebrate in lines 52 – 57 of the revised manuscript: *“The transmission of symbionts from one generation to the next plays a central role in the ecology and evolution of mutualistic associations [4]. Symbionts can be transmitted vertically from parent to offspring, intimately tying them to the reproduction and development of their host [4], as known from vesicomid clams [5]. Alternatively, in horizontal transmission, symbionts are recruited each generation anew from the environment, and the symbiotic partners are physically separate from each other before the symbiosis is established [4].”*

Line 54

change “entailing that” to “where”

Due to our revisions in response to the reviewer's comment for “lines 50 – 55” this sentence was rewritten.

Line 55/56

*“In many hosts that rely on horizontal transmission, the acquisition of symbionts **induces** drastic morphological and developmental changes”* While there are certainly data on bacterium-mediated changes in host physiology and on bacteria-induced settlement in larvae, data on demonstrable bacterium-mediated changes in host morphology and/or on bacteria being a cue for developmental changes remain limited (e.g. certain nematode species, some insects e.g. *Aedes*-associated *Wolbachia* and of course the squid *Euprymna scolopes*). In marine organisms in particular, there is very little empirical evidence to support this statement (that is, it may prove to be a widespread phenomenon in symbioses but we don't yet have the data to say so), with the notable exception of the squid *Euprymna scolopes* and its bioluminescent bacterial symbiont, *Vibrio fischeri*, for which the symbionts role in cuing changes in squid morphology is unequivocal.

Even in *Riftia pachyptila*, for which the process of horizontal transmission during host development has been documented, no evidence is proposed for the bacterial role in cuing / inducing host trophosome development (other than noting that the observed epithelial apoptosis in the host following infection echoes that seen in the *Euprymna-Vibrio* symbiosis during infection). Thus please change to “is believed to [induce / trigger / cue]... ” or something similar, to allude to the fact that we do not yet know which partner mediates this process in most symbioses and, please find a more suitable reference (perhaps one that is more specific) to provide evidence that this is known to occur in certain model species. You could, for example, write the following for remainder of this paragraph (multiple options provided in square brackets): “In hosts that rely on horizontal transmission, the acquisition of symbionts may even [induce / trigger / cue] morphological and developmental changes in the host, which can often be drastic and typically occur upon or after the arrival of symbiont partners. These can range from tissue rearrangement, known to be symbiont induced in the *Euprymna* squid-*Vibrio* symbiosis [REF: Montgomery and McFall-Ngai. 1994], to the largescale modification of existing organs, typically through hypertrophy (e.g. gill of the bathymodiolid *Idas modiolaeformis* REF: Laming et al 2014; oesophageal gland of the peltospirid gastropod *Gigantopelta chessoia* REF Chong et al 2018*) or the development of a novel symbiont-dedicated organ, such as the trophosome of the siboglinid *Riftia pachyptila* (REF Nussbaumer et al 2006) where, in these latter examples, the role of symbionts as mediators remains hypothetical, though probable.”

Finally, the Bright & Bulgheresi review does not provide any evidence to support this statement either, being predominantly concerned with the subject of inter-partner “dialogue” and migration of symbiont partners to target host organs during horizontal transmission (notwithstanding the probable veracity of your statement, given future evidence).

*Chong et al 2018 reference here: <https://doi.org/10.1098/rspb.2018.1099>

We thank the reviewer for this detailed comment and suggestion. We have improved our wording to be more specific. We changed the text to communicate more specifically what is known: “The transmission of symbionts from one generation to the next plays a central role in the ecology and evolution of mutualistic associations [4]. Symbionts can be transmitted vertically from parent to offspring, intimately tying them to the reproduction and development of their host [4], as known from vesicomyid clams [5]. Alternatively, in horizontal transmission, symbionts are recruited each generation anew from the environment, and the symbiotic partners are physically separate from each other before the symbiosis is established [4]. In many hosts that rely on horizontal transmission, the acquisition of symbionts triggers morphological and developmental changes in the host. These can range from tissue rearrangements, known to be symbiont-induced in the *Euprymna* squid-*Vibrio* symbiosis [6], to largescale modifications of host organs, typically through hypertrophy [7], or the development of a novel symbiont-housing organ, like the trophosome of the tubeworm *Riftia pachyptila* [8]. As most of these symbioses are uncultivable, it is still unclear if these developmental

changes are mediated by the host, actively induced by the symbiont or a mix of both." (Lines 52 – 65 in revised manuscript)

Line 65-67

*The findings of these studies are not fully disclosed here (and they are highly relevant to the current study). In addition, transmission electron microscopy was also used in Laming et al 2014, the study by Gaudron et al 2012 (here: doi: 10.1086/BBLv222n1p6) on symbiont transmission in *Idas modiolaeformis* is overlooked entirely and finally, given the definition of juvenile in the current MS deviates significantly from other papers on deep-sea mussel species, it might be prudent not to use the term juvenile here at all, as it is not synonymous with the juveniles in the current study.*

Thus the sentence should be rephrased as follows (or similar): "In two smaller-sized bathymodioline mussels that colonise organic falls and seeps, histological analyses, and fluorescence and transmission electron microscopy showed that both were aposymbiotic as newly settled plantigrades. Symbionts were first identified shortly after the onset of dissoconch shell growth, providing evidence for heterotrophic larval dispersal, post-metamorphic acquisition and horizontal transmission in these species (Laming et al 2014; 2015) These findings corroborated earlier, targeted fluorescent in-situ hybridisations (FISH) of adult reproductive tissues, which did not find any evidence of transovarian (vertical) bacterial transmission (Gaudron et al 2012)."

We have modified the text and incorporated the suggested literature as follows: *"It is assumed the symbionts are transmitted horizontally, based on phylogenetic studies that showed a lack of cospeciation between hosts and symbionts, as well as morphological studies that found no evidence for symbionts in the mussels' reproductive tissues [10-14]. In bathymodiolins of the genus *Idas*, commonly found at organic falls as well as seeps, the larvae remained aposymbiotic until they settled and developed the dissoconch shell, indicating a heterotrophic lifestyle in the larval dispersal phase, and horizontal acquisition of symbionts during or after metamorphosis [7, 15]."* (Lines 69 – 75 in revised manuscript)

Line 69

The term "late larval phase" is not accurate here. In Salerno et al 2005, the smallest stages referred to are "post-larvae" (i.e. post-metamorphic), which for deep-sea mussels is normally synonymous with plantigrade in shallow water mussels. Post-larvae, by definition, are "after larvae".*

Thus, they were not from the late larval phase, which would be the pediveliger, but rather the earliest postmetamorphic stage, i.e. either postlarvae (as stated in that study) or, following your classification system plantigrades / early postlarva without dissoconch,.

**By the way, the definitions for shallow water mytilids are more complicated, due to the propensity for byssus-drifting postlarvae that become temporarily resuspended and thus planktonic, making them functionally distinct from crawling plantigrades but not necessarily morphologically distinct (differences in byssus type being excepted). I revisit this distinction in later comments*

We thank the reviewer for pointing this out and have changed the text to: *"The earliest life stages described so far had undergone metamorphosis and were already colonized by symbionts, with a well-developed symbiotic habitus that was indistinguishable from adult mussels [16]."* (Line 78 in revised manuscript)

Line 79

“...“B”. *childressi* (also referred to as *Gigantidas childressi* e.g. [15]).”

In the face of overwhelming evidence from multiple papers (including some robust concatenated analyses), it is now probably more appropriate to refer to this species as Gigantidas childressi, accepted as such in WORMS.

Please see our answer above to comment “Line 25”.

Line 84

Suggestion for restructuring and rewording of sentence to be clearer and more succinct (multiple suggestions provided in square brackets):

“This approach allowed an integrative analysis of the processes involved in symbiont colonization and its effects on the host body plan, [from the organismal to the cellular level / from the whole animal down to single host and symbiont cells].

We revised the text as suggested by the reviewer: “*This approach allowed an integrative analysis of symbiont colonisation and its effects on the host body plan, from the whole animal down to single host and symbiont cells.*” (Line 93 in revised manuscript)

Line 104

Were measurements made of the Prodissoconch I and PII in all individuals or was only total shell length measured (i.e. equivalent to Prodissoconch II only in those specimens for which no dissoconch shell had developed)?

We measured total shell dimensions for all specimens. As mentioned above we had to move parts of the methods to the supplement. The information about the shell measurements are now located under Supplementary Methods b and read as follow: “*Specimens were photographed with a Nikon SMZ 25 stereomicroscope equipped with a Nikon Ds-Ri2 colour camera and the software NIS-Elements AR. Total shell dimensions were recorded as shown in Figure S1c (Table S2) and shell margin limits were identified by their unique colouration (Figure S1b and d)*”. (Lines 187 – 190 in the supplements)

Line 109

low-viscosity resin

LR white, I presume?

We did not use LR white, the resin we used is called “Agar low viscosity” resin.

Line 110

This is an ultramicrotome, not a microtome

The reviewer is correct, this was done on an ultramicrotome and we corrected the text accordingly, please see line 203 of the supplement.

Line 117

Until this point the MS reads like all methods were performed on all species (e.g. abstract, and the closing paragraph of the introduction). However, clearly FISH was only performed on B. puteoserpentis. In fact, examination of the supplementary materials (for which, by the way, I have created a separate section of corrections) suggests that B. puteoserpentis has been used as the model species for most analyses. I now see that this is ultimately alluded to in the results. You need to be MUCH clearer about which species have received which analyses in the main-text methods. To force a reader to dive into supplementary material to confirm this, is not acceptable and stating it in results is not appropriate. There is nothing inherently wrong with using one species as an example and making comparisons with other species, but it has to be clear that this is happening from the start. As a minimum you must provide details concerning the number of specimens actually analysed by various techniques, since the veracity and robustness of the findings can only be assessed knowing this information?

We agree, and while we had to move parts of the methods to the supplement due to length restrictions, we did add new section to the Methods of the main text that describes the correlative workflow in more detail (see lines 119 – 126), and a table that details the number of individuals from each species and developmental stage analysed (Table 1). We used *B. puteoserpentis* as the central pillar of this paper, because specimens from this species were the best-preserved and had the widest range of developmental stages.

For our correlative image approach (SR μ CT, LM and TEM), we examined all three mussel species. At the histological level (LM and TEM) all three species were similar in their morphology with respect to their development and symbiont acquisition (see Figure S15). FISH was only done on *B. puteoserpentis*, due to sample availability. Throughout the Results and Discussion, we drew all conclusions and interpretations based on our results from the three host species, unless specified otherwise.

Line 120

“probe”

It was a cocktail of three general probes, was it not? Worth mentioning, since this is a clever way to achieve more “universal” coverage.

Indeed, this is the typical EUBI-III probe mix. We mentioned this in the Supplement Table S3.

Line 137

abbreviation of STEM is not defined (Scanning Transmission Electron Microscopy, I presume?)

We have now removed the abbreviation completely.

Line 149

Is this an actual average value, or a colloquialism? If so you need a measure of variance and define which type (presumably a mean value?)

Due to length restrictions we had to move parts of the methods to the supplement. We changed the text accordingly to the suggestion: “SR μ CT models were used to ground truth the volume calculations

from histological section series by measuring sectioning-induced tissue compression, which varied between 1 and 7 % depending on the specimen (n = 3)." (Line 278 in the supplement).

Line 163

So, here is an example of where an honest inaccuracy veers towards misinformation. Figure 1 does not represent the results of your analysis of three species. It represents the results of a given (representative) specimen for each developmental stage from one species only subjected to either serial sectioning or SruCT and visualised in 3D following segmentation. While it might be representative of what you found in all three species (and I can't be sure if it is because the results and methodology do not provide sufficient information regarding the number of samples examined for each species /stage and by what means), it is not a suitable figure reference for this opening statement. The supplementary materials are the only data that truly encapsulate this statement. Note also previous comments about the fact that you have three species from two genera of large bathymodiolin mussel, not three species of *Bathymodiolus* and that of the three, the most distantly related *G. childressi*, which also inhabits a different reducing habitat in a separate oceanographic region in the Atlantic, is the species for which you have least samples, unless I misunderstood?

We have revised the legend of Figure 1 to clarify that the images show individual stages of *B. puteoserpentis*. The first sentence of the figure legend was revised to: "Three-dimensional visualisation of section series and SR μ CT measurements of *B. puteoserpentis* developmental stages based on analyses of four individuals for each stage." Furthermore, we adjusted the figure link at the corresponding text passage and included Figure S5 and S6 that show the comparison between *B. puteoserpentis*, *B. azoricus* and *G. childressi*."

Line 169 "post-larva"

Ok, so mytilid life stages... Postlarval forms are not so clear cut in mytilids as they are in other bivalves, and your use of these terms does not appear to follow the convention of many other shallow water studies: As pointed out in Baker and Mann 1997, in mussels, there exist "...two functional life history stages intermediate between the pediveliger larva and the sedentary juvenile. These two stages, which may either alternate or occur exclusively, include the benthic plantigrade (Carriker, 1961) and the planktonic postlarva (Sigurdsson et al., 1976)." So, the distinction between plantigrade and postlarva is basically a behavioural /functional one. The postlarva is post-metamorphic, despite also being planktonic. These post-larvae re-enter the plankton by way of byssus drifting using specialised gossamer-like threads that are structurally different from normal attachment byssuses. Plantigrades are of a similar size to the postlarvae described above. However they refer to mid- or postmetamorphic individuals not considered juveniles (juvenile size-range definitions vary a lot) that retain high levels of motility but are benthic, akin to your definition of plantigrade and postlarva combined, in the current study). Note that the term plantigrade is the norm for mussels, subdivided into early and late, based predominantly on whether they are primary settlers (away from adult conspecifics) or secondary settlers (gregarious settlement in association with adults). Post-larva is used more generally and has different meanings for different taxa. All that said, I agree that there are issues with consistency in the literature. The delineation between late plantigrades and small juveniles is also highly subjective, as you may probably discovered, since some use behavioural shifts / relocation of late plantigrades as the benchmark for being juvenile, others use onset of dissoconch growth (particularly in the deep sea, where we lack evidence for secondary settlement behaviour). The convention you are using appears to be based on Cannuel et al 2008., who basically invented their own definition because of these ambiguities. Just be aware that they aren't very consistent, as an earlier study by a subset of the authors on gill development in the Pacific Oyster *Crassostrea gigas* refers to all

postsettlement specimens as juveniles (Cannuel and Beninger 2006)!

<https://doi.org/10.1007/s00227-005-0228-6>

In deep-sea species, for the symbiont acquisition / developmental studies examining much smaller bathymodiolin (Idas-like) species, "juvenile" dissoconch shell development was used as a benchmark for being juvenile (shortly after which the first inflexion of gill lamellae was identified, but not the appearance of the ventral groove). Given the maximum size of your study species which is obviously much greater, I understand the choice to use later stages of gill development (i.e. appearance / formation the ventral groove) as a benchmark for being juvenile, as in Cannuel et al 2008. Just be aware that Shawn Arellano's work on Gigantidas – "Bathymodiolus" – childressi larval and reproductive biology also follows the "juvenile = dissoconch growth" staging convention.

In any case, I recommend that you re-examine the literature, as your stage definitions appear to be heavily influenced by Cannuel et al 2008, who deviate from standard terminology.

Regardless of your ultimate decision, you need to relate what you mean by each stage to what has been used in the past and you should provide a brief explanation of the biological reasoning behind this decision, as it currently reads as arbitrary. An obvious one relates to your earlier benchmark of velum absorption: i.e. shifts in filter-feeding strategy from predominantly velar (ending with pediveliger settlement and metamorphosis), to DIC and foot-driven (early / late plantigrades) and ultimately gill-driven, once the ventral groove has formed (juvenile onwards)? I personally would discourage the use of post-larva as a developmental stage, and use it in the more general context of post-larval processes.

We thank the reviewer for these insights and have revised our text and references in the paragraph that define the different life stages. We have removed the term post-larvae throughout the manuscript. Please see our comment above in response to comment "line 33/34" of the reviewer.

Line 170

"once the ventral groove of the gills is formed"

The biological significance of which is... (I know, you know, but the reading public may not)

We thank the reviewer for pointing this out and we added an explanation to the function of the ventral groove to the text, see line 144 of the revised manuscript and answer to the comment "line 33/34" above.

Line 172 -180

It feels a little like the number of individuals used for shell measurements is being stated explicitly for impact, while the number of specimens examined for EVERYTHING ELSE is not stated at all, presumably because only a very low number of individuals was actually examined using the correlative approach technique? You need to state here in the results or in the methods (of the main text) the number of individuals examined for each stage by each technique for each species. The lack of detail regarding this and the disorganised methodology must be improved if this MS is to be published in Proceedings B, or elsewhere

We agree with the reviewer that the number of individuals used for each technique should be explained more clearly. This information was provided in detail in Table S1, and we have also added a new table in the main text summarizing this information (Table 1, see below). We changed the corresponding sections in the Methods and Results accordingly.

Table 1 Overview of number of individuals analysed for each species, developmental stage, and imaging method

species	method	pediveliger	metamorphosis	plantigrade	juvenile	adult
B. puteoserpentis	SR μ CT / μ CT	-	-	-	1	-
	SR μ CT + serial sectioning	1	1	2	1	-
	serial sectioning + LM	2	2	-	-	-
	serial sectioning + TEM	2	2	3	-	-
	FISH	2	1	3	1	-
	total	7	6	8	3	-
B. azoricus	SR μ CT / μ CT	-	-	-	2	-
	SR μ CT + serial sectioning	-	-	1	-	-
	serial sectioning + LM	1	1	1	-	-
	serial sectioning + TEM	1	-	-	-	-
	total	2	1	2	2	-
G. childressi	SR μ CT / μ CT	-	-	-	-	-
	SR μ CT + serial sectioning + TEM	1	-	-	-	-
	serial sectioning + LM	-	2	2	-	-
	total	1	2	2	-	-

Line 190

palp or palps (they occur in pairs usually?)

Thanks, we changed “palp” to “palps” in the text, please see line 145 of the revised manuscript.

Line 190

“digestive glands”

currently your 3D data and the labelling of your serial sections indicate you don't fully appreciate the extent of the digestive glands, and have identified the main digestive ducts only (see comments below)

We agree and have changed the labels of the corresponding supplemental figures and 3D data from digestive glands to digestive ducts and added the digestive diverticula to the labels of Figure 1 and Figures S3, S4, S6, S9 and S10.

Line 191-192

was the style actually visible in histological sections or CT data? Also, please remove “an” before anal.

Even though it is hard to preserve the crystalline style, we were able to identify it in one specimen. We removed “an” before anal papillae.

Line 192

“epithelial cells of the stomach contained membrane-bound lipid vesicles”

*I may be mistaken but based on my experience with similar species at similar developmental stages, these aren't actually in the stomach epithelium. The CT segmentation in figure 1 indicates that the inclusions lie outside the region of the stomach entirely and are in fact closer to (but not inside) what appear to be the main digestive ducts in your "stomach and digestive gland" surface mesh. This part of the CT visualisation does not appear to include all the remaining digestive gland tissue (composed of a dense convoluted arrangement of tubules called diverticula) that is associated with these ducts, which would occupy the same space as the lipid inclusions (which I agree are likely to be vacuolar lipid stores, based on their appearance in serial sections). My interpretation is that these inclusions are almost certainly spread throughout these as-yet-unidentified digestive diverticula that communicate with the main ducts and thus the stomach (both of which appear to have been segmented correctly). This is confirmed unequivocally in the supplementary animations where the areas occupied by lipid inclusions match exactly the expected areas that digestive diverticula would be found (please see Laming et al 2014 and 2015 for analogous examples in Idas-like mussels, but also the work of the deep-sea team based in the Azores for examples from adult *B. azoricus*). Even in the toluidine-stained serial sections, there are examples where this "epithelium" would have to be unaccountable thick (aka it's actually developing digestive gland). Unfortunately it's not possible to see the fine-scale structure in these histological images to confirm this one way or another. In your defence, this oversight is understandable, as in my experience, digestive gland tissues typically display a less coherent histological arrangement in early post-settlement mussels. I also have experience with issues relating to thresholding of greyscale signal associated with digestive glands when using standard or synchrotron amplified CT scanning techniques, even with suitable contrast staining. This makes segmentation a headache and I'm not surprised to find that the tissue surrounding the inclusions was not attributed a clear function. However, this needs to be re-examined, with some greater care paid to the functional anatomy of this region. You need to either revisit the CT data and update the segmentation of the different tissues present (though this represents a lot of work as all the animations would need to be recreated too) or indicate clearly that the CT data only includes the **stomach and main digestive ducts of the digestive gland** and not the main mass (**diverticula**). Your supplementary tissue sections in which the digestive gland appears also need to be corrected as the labelling is not accurate (any tissue in this region that is not obviously the line of epithelial nuclei that surround the stomach lumen, in which the lipid stores appear) Finally, references to lipid inclusions stomach epithelial cells or digestive tract (you actually write both) must be changed.*

*For comparison, here are the main morphological features of *Mytilus edulis pediveliger* larvae.*

Large velum for swimming and feeding

Ciliated palp that sorts food particles

A few pairs of gill filaments

*Mouth, oesophagus, stomach with style sac and **large digestive gland**, simple intestine*

Thin mantle that secretes shell

Foot used in crawling and byssus secretion

Cerebral, pedal and visceral ganglia, sensory pigment spots, pedal statocysts

from:

*Gosling 2015. Reproduction, settlement and recruitment. *Marine Bivalve Molluscs*, 157–202.*

doi:10.1002/9781119045212.ch5

Thank you for your extensive comments here - and elsewhere - we appreciate the time and effort it cost you to help us improve our manuscript. We are pleased to let the reviewer know that both you and we are correct. The epithelial cells of the stomach contained lipid vesicles as shown now in the newly added Figure S7, but also a large proportion of these lipid vesicles were in the digestive diverticula. We corrected Figure 1 and Figure S3, S4, S6, S9 and S10 and Videos S1, S2, S3 and S4 accordingly.

Line 193

average = mean ? Also, you need a measure of variance around this value

We agree but due to length restrictions we had to move these details to the supplementary note 1d (line 82). The text in the manuscript reads now: "*The diverticula of the digestive glands and the epithelia of the stomach contained membrane-bound lipid vesicles (Figure 1, S6 and S7).*" (Line 169 in revised manuscript)

Line 210/211

*"These changes in gill morphology indicate a functional shift from a purely respiratory organ to a filter feeding and respiratory organ" I don't agree with your reasoning here. Increased inter-filamentary space doesn't speak to a change in function but rather an increase in the animals size and resulting spread of filaments; a necessary process to accommodate the budding of new filaments posteriorly. In fact, filter feeding relies on relatively tight packed filaments aligned to one another in order to work efficiently, since the ventral-ward transport of particles is achieved by the action of opposing lateral cilia between filaments, before being directly anteriorly along the ventral border of the gill. Overly wide gaps would actually interfere with the fluid dynamics of this process. Examination of live plantigrades in shallow water mussels confirms that at these sizes, gill coordination is poor, marked by random contractions rather than coordinated movement of cilia and that the ciliated foot plays a central role in current generation (I strongly recommend reading sections on larval-to-postlarval development found in "Marine Mussels: Their Ecology and Physiology" by Bayne, 1976 (see pages 94/95 in particular). I can also confirm that even in juveniles of the wood-colonising mussel *Idas argenteus* the foot remains highly ciliated and these cilia remain active, though the role of particle transport to the mouth by this stage is assumed by the gills, which possess ventral grooves.*

*For an example video of particle transport in a bathymodiolin mussel (in this case, "*Idas simpsoni*"), please see Supplementary material 2 of the 2018 Review article on the Lifecycle ecology of Bathymodioline mussels. available here:*

<https://www.frontiersin.org/articles/10.3389/fmars.2018.00282/full#supplementary-material>

We agree and clarified the text as follows:

"The organs involved in filter-feeding also changed. In pediveligers, the main feeding organ was the velum, which collects particles from the seawater and transports them to the oral labial palps and mouth. After the degradation of the velum, particle sorting was taken over by the highly ciliated foot (Figure S5, S8 and S9) in metamorphosing mussels, and by the gills in late plantigrades and juveniles." (Lines 183 – 188 in revised manuscript)

Line 211

"digestive tract"

Alarmingly, these vesicles appear to have moved from the "stomach epithelium" to the "digestive tract"! However, as mentioned elsewhere my feeling is that they are in fact located in as-yet-unidentified digestive diverticula, though higher resolution histological images would be useful to confirm this.

Please see our response above to comment: "line 192".

Line 212-214

If these are average percentage values then measures of variance are needed, unless some are calculated from < 3 specimens, in which case its more appropriate to provide percentage ranges. Regardless, the number of specimens examined for each stage should be stated Also: Why don't you use CT volume calculations more extensively in the main text? I eventually discovered in the supplementary materials that you had actually examined gut volumes for example but bafflingly this does not feature in the main text. For example you've missed a glaring opportunity to examine how relative gill-volume changes with development (as a proportion of total soft body volume) in the context of symbiont carrying capacity with increasing host size. This would be a welcome addition. Taken with a parallel calculation of the relative volume of the gut with increasing size, this would provide a more quantitative indication of the "importance" of each system in the trophic biology of the host throughout development and presumably provide support for some of your (valid) conclusions which relate to the shape and arrangement of the gut only.

This calculation is based on an n of 1 per developmental stage. For the lipid vesicles, the reduction over the development was striking and the SR μ CT reflected what we saw on the serial sections. We therefore included this data here. For more detailed comparisons of complex organs like gills and digestive system, more replicates would be needed. This data is shown in Table S6, but we prefer to avoid making absolute statements based on this data, and therefore have not included and discussed this in the main text.

Line 214

"no longer present in juveniles and adults"

This is a very nice finding! It's very satisfying to see the transition from lipid-supported heterotrophy, to mixotrophy, to predominantly chemosymbiotic nutritional modes.

We thank the reviewer for this comment.

Line 221 – 246

*A lot of this section is methodology and does not belong in the results, except only to reiterate the method used (eg. see bold text) Central to an accurate assessment of symbiont colonization and symbiont-mediated morphological changes was our combined approach of correlative SR μ CT, FISH, and light and microscopy (Video S5, Figure S10). This allowed us to rapidly screen whole animals yet achieve the resolution needed to identify the colonization of single eukaryotic cells by symbiotic bacteria. **We first searched for a morphological characteristic that was visible using light microscopy and reliably revealed the presence of symbionts in host cells. Previous studies [28, 29] showed that in juvenile and adult Bathymodiolus mussels, the morphology of epithelial cells colonized by symbionts is remodelled: i) The microvilli that cover all epithelial cells are lost (known as microvillar effacement), and ii) epithelial cells become hypertrophic (swollen) compared to aposymbiotic cells. We identified a third characteristic change in epithelial cells colonized by symbionts that has not received much attention, namely the loss of cilia (Figure 2 c, f and i). We tested if these three morphological characteristics had predictive power for symbiont colonization by analysing 1813 epithelial cells from a subset of six B. puteoserpentis individuals (2 pediveliger, 1 plantigrade, 3 post-larvae) and comparing LM images of these cells with their***

correlated TEM images (average cell count per sample 302). Our analyses showed that all cells predicted to have symbionts in the LM dataset were indeed colonized in the TEM dataset, and likewise, all cells predicted to be aposymbiotic were free of symbionts (FigureS10). **We next used our verified morphological characters to reveal the onset of colonization in *B. puteoserpentis* in the subset of six mussel individuals.** All pediveliger cells were free of symbionts ($n = 797$ host cells in 2 pediveliger). In plantigrades 15% and in post-larvae 23% of all analysed gill, mantle, foot and retractor muscle epithelia cells were colonized by symbionts ($n = 336$ host cells in 1 plantigrade; $n = 680$ host cells in 3 post-larvae). **Finally, we expanded our analysis to LM analyses of serially sectioned individuals,** and these confirmed our results from the correlative dataset (all pediveligers were aposymbiotic, $n=5$; 50% of the plantigrades were colonized by symbionts, $n=6$; all post-larvae were colonized by symbionts, $n=5$).

We agree and have addressed this as described in our reply to comment “line 117”.

Line 222 “symbiont-mediated”

Remove this term: You do not provide evidence that the changes in the gill are symbiont mediated (though this is of course quite possible), just that the acquisition of symbionts coincides with the onset of hypertrophy and an apparent cessation in further gut development. The mechanism is not demonstrated, only an observation of the pattern.

We agree with the reviewer and have revised this sentence as following:

“Central to an accurate assessment of symbiont colonization and symbiont-mediated morphological changes was our correlative approach, which combined SR μ CT, light and electron microscopy (Video S5, Figure S2) and was complemented with FISH.” (Line 195 in revised manuscript).

Line 224-226

In my view this is the main strength of the MS.

We thank the reviewer for these kind words.

Line 229

consider changing “remodelled” to something less suggestive of interior design e.g. “fundamentally altered”

Revised as suggested: “Previous studies [30, 31] showed that in juvenile and adult *Bathymodiolus* mussels, the morphology of epithelial cells colonized by symbionts is fundamentally altered:...” (Line 201 in revised manuscript)

Line 233 – 239

Credit, where credit is due: this is ingenious and a stand-out aspect of the MS. That said this is One question: the example provided of the correlative light-electron microscopy approach in Figure S10 concerns the presence of MOX symbionts only (which is fine as they are always accompanied by SOX symbionts, right?). However, as you likely know, MOX are considerably larger than SOX (exemplified by your TEM images). Was this predictive power thus restricted to the presence of MOX symbionts and if so, how do you know that epithelial cells infected by

SOX symbionts only haven't been overlooked by this extrapolative approach? In other words, is the degree of hypertrophy / loss of microvilli/cilia similar in cells colonised by SOX only comparable to those with SOX/MOX?

We thank the reviewer for these kind comments. For our predictive approach based on correlative light and electron microscopy, we did not use the MOX as a diagnostic trait for symbiont colonization. Instead, we screened for changes of the host cells that we had previously verified to be diagnostic with EM, namely the loss of microvilli and cilia. At the EM level, we could readily identify SOX cells as highlighted in Figure 2. Cilia and microvilli were also absent in gill cells that were only colonized by SOX (see Figure 2). Therefore, we did not have to rely on the presence of large MOX to identify symbiont colonisation. While the level of hypertrophy was directly related to the amount of symbionts present, loss of microvilli and cilia was not. We were able to identify eukaryotic cells that were colonised either by only SOX, SOX and MOX already at the light microscopy level.

Line 239-240

These sorts of details must be also be provided in the methodology as mentioned above

We agree. Please refer to comment "line 117" for more details on how we changed the Methods and Results sections.

Line 241

"In plantigrades" change to "In the plantigrade measured" (since n = 1)

Also if the percentage value for "post-larvae" is based on the three specimens, is it a mean? If so provide measure of variance or alternatively provide the range of percentages calculated for the three specimens

We realized that the number of analysed specimens in our original manuscript was not correct. We investigated two mussels that were under ongoing metamorphosis with the correlative imaging approach, not only one (Table 1). Therefore, we changed the text to: "*In the metamorphosing mussels examined with our correlative approach 1 – 15% (n = 488 host cells in 2 metamorphosing mussels) and in plantigrades 21 - 26% (n = 680 host cells in 3 plantigrades) of all analysed gill, mantle, foot and retractor muscle epithelia cells were colonized by symbionts.*" We changed the mean of colonized host cells in plantigrades to a range of colonized cells, as three specimens were investigated. (Line 217 in revised manuscript)

Line 250

Are color overlays drawn by eye based on bacteria ultrastructure I presume, rather than formally identified by a truly correlative approach (e.g. CLEM-based FISH?). Given the focus of correlative approaches in the MS, this needs to be clear.

Yes, color overlays in this figure are based on ultrastructural identification of the typical bacterial morphology. We wanted to help the reader easily distinguish between host cells that were aposymbiotic and those that were colonized by symbionts. The original figure without color overlays is provided in the Supplement (Figure S14). Additionally we added the information that the color overlays were added for visibility reasons in the legend of Figure 2.

Line 256 – 257

“Expanding our analyses to all three Bathymodiolus species, we found that the pediveligers of all three host species were aposymbiotic” Here I finally get a sense of the approach used, but you don't explicitly express it. In fact, A lot of unspoken methodology appears to be alluded to in this one sentence. Effectively what you are saying, I presume, is that histological sections of the other two species were interrogated in the same way using the same LM cell morphologies, based on the validated method used in B. puteoserpentis? Was this carried out manually, by eye or using a semi-automated, machine- learning-type approach? Please correct me if I am wrong, but this suggests that no actual correlative microscopy with TEM, SRuCT or FISH was performed in the other two species, correct? This is not clear in the methods AT ALL, which is where this information needs to be detailed (in the main text, not the supplementary materials).It changes the interpretation of the entire MS and calls the reliability of your analyses for the other two species into doubt (particularly when this revelation is buried in the results). If the above interpretation is correct, I have very genuine concerns about this extrapolative approach, in the absence of any cross validation using either (S)TEM or FISH in the remaining species. If you have carried out such validation, evidence of this needs to be included. If not, then the narrative of the MS must change to reflect the degree of confidence you have in the results for each species. I would also expect to see the word "putative" (or equivalent) being used, since these are inter-host inferences of symbiont distributions / acquisition patterns in multiple epithelia in each host, based solely (unless I've misunderstood) on comparative histology. This is made more troubling by specific concerns: For B. azoricus, while this comparative approach should work well, as the appearance of hypertrophied, nonciliated, microvilli-devoid, epithelial cells in this species is likely to be very similar to that of it's sister species, it has come to my attention that you may have used different tissue thicknesses and / or embedding procedures for this species (based on the tell-tale differences in tissue appearance in the Supplementary figures). This doesn't necessarily mean the proposed comparative approach won't work if carried out manually by eye. It does pose a problem if the approach is somehow automated. For G. childressi, I see three possible issues. First, there are far fewer specimens used (i believe?), the section quality is either poorer or the staining is less differentiated in some sections (e.g. Figure S2) and the host species is less closely related and as a seep species, the relative proportions of MOX and SOX symbionts is likely to heavily biased to MOX. Can you provide assurances/evidence that these species have bacteria-colonised hypertrophied epithelial cells that are roughly equivalent to those found in B. puteoserpentis please. The whole-animal histological section comparisons provided in the Supplementary materials do not really demonstrate this well

We thank the reviewer for this thoughtful comment and have made several changes throughout the manuscript, from the Methods to the Result section, to address them. To sum them up here:

The correlative imaging approach of SR μ CT, LM and TEM was performed on *B. puteoserpentis*, *B. azoricus* and *G. childressi*. For LM analyses, all samples of all three species were embedded and sectioned in the same way, with the same section thickness. Due to sample availability, FISH was only performed on *B. puteoserpentis* as previously stated in the Methods section, Figure 3 and now specifically stated again in Table 1.

The analysis of cell morphologies was done manually, and no automated algorithm or machine learning was applied. All three species showed the same characteristics, with colonized host cells displaying a hypertrophic habitus and devoid of microvilli and cilia. To support these statements we have added an additional Supplementary Figure S15, which highlights that these characteristics are a useful marker for determining the colonization status in all three species.

One of the reasons the *B. azoricus* data looks different is that these specimens were fixed with a different fixative / buffer combination that resulted in a much poorer preservation than the other two species. We added this information into the Supplementary Methods a (line 181) and Supplementary Table S1. Additionally, two automated microscopes were used for these analyses, as described in the Supplementary Methods f (lines 257 - 259), which may have also contributed to a slightly different appearance between specimens.

Line 257-259

This is interesting, but I'm glad the speculation around this is cautious since I believe there are plenty of free living SOX bacteria at vents and seeps, with some evidence for free-living MOX bacteria in seeps (at vents its less clear) and unless I have missed something in the literature, the positive identification of known symbiont OTUs in the environment, as free-living bacteria, remains elusive.

We totally agree that for now we can only speculate if these bacteria are symbionts or not.

Line 265

Reference needed

As suggested by the reviewer we provided references here (Fisher et al. 1987, Wentrup et al. 2014 and Fiala-Medioni et al. 1997), please see line 238 of the revised manuscript.

Line 274

"The good preservation"

This sounds innately odd in English, as a native speaker. Consider changing to "superior". Also, I believe it was also the "better coverage of developmental stages" too, was it not?

We changed the text according to the suggestions of the reviewer: *"The superior preservation of B. puteoserpentis specimens allowed us to analyse the process of symbiont colonisation in this species in more detail."* (Line 249 in revised manuscript)

Line 276

Please change to "... a few bacteria in gill, mantle and foot epithelial cells..."

Changed as suggested, please see line 251 in the revised manuscript.

Line 283-285

This has proven notoriously difficult to capture in still-image analyses in deep-sea species and is a credit to your sampling effort for this species. Well done!

Thank you!

Line 289-290

Worth providing a possible explanation as to why this might be in the discussion?

Unfortunately, at this time we have no data and therefore could only speculate on the molecular mechanism at the time of host cell colonisation. As host-microbe recognition is a complex topic, and not in the scope of this study, we decided to not include any speculation on this topic given the word limits of the journal.

Line 292

Figure 3

Great FISH images. Question though: Is there additional MOX signal in the centre of the foot in b, in a region where nuclei densities are lowest (i.e. potentially not epithelium, but more internal) and in the lumen of the stomach in a?

Thank you for these insightful questions. We interpreted the signal in the MOX channel, which is visible in the area of the foot, as non-specific signal. In comparison, the signal is weaker than the signal from the colonized epithelia, and we often record autofluorescence from this area. The reviewer correctly observed that there appear to be signals from MOX symbionts in the digestive system. We are currently investigating if these signals are really from MOX, but this needs further analyses and is outside the scope of this study.

Line 303

"Bivalves" is a very large group of animals and this is actually taxon specific. It's well established that what works for staging oysters does not hold true for mussels, for example. As alluded to already, in mussels and many other bivalves groups it's actually a combination of shell size and morphology and environmental context that's normally used to estimate developmental stage, in the absence of internal anatomy (i.e. relative proportions of PII and Dissoconch growth and – in mussels – whether plantigrades are settling away from, or among conspecific adults). However, in bathmodioline species, it's not yet known if primary and secondary settlement behaviours occur, as in shallow-water mussels. The Salerno paper you cite used shell size and the proportional composition of each shell type to classify specimens. It happens to use a different – somewhat simplistic but common – staging convention of assigning all specimens with juvenile dissoconch growth as being juveniles (there are several papers from Shawn Arellano's early-life stage studies on *G. childressi* which also adopt this definition of juvenile, e.g. <https://royalsocietypublishing.org/doi/10.1098/rspb.2013.3276>). Your data demonstrate that there's a possibility that the "post-larvae" in the Salerno paper could have been pediveligers still in the process of settling (though a velum is a hard morphological trait to miss in a species with translucent valves) or early plantigrades following metamorphosis (i.e. what you term "plantigrades": this latter case is a very real possibility). Since you appear to be focused on the Salerno paper specifically perhaps change this sentence to something that discusses the pitfalls of assigning early developmental stages based on shell lengths in deep-sea species: "Our study shows that the use of shell characteristics alone as a means to determine the developmental stage of deep-sea mussels (e.g. Salerno et al 2005) cannot be relied upon, particularly for shell lengths typical of settling pediveligers and [early] plantigrades."

We thank the reviewer for these suggestions and adapted the text accordingly: "Our study shows that the use of shell characteristics alone to determine developmental stages of deep-sea mussels is not

reliable (e.g. [16]), particularly for shell lengths of settling pediveligers and early plantigrades.” (Line 270 in revised manuscript)

Line 306-309

*This is actually common in shallow water mussels too, as you allude to in the next sentence. It is well known for other bivalve larvae / postlarvae as well (including Mytilus), because once competent to settle, larval size at settlement is entirely dependent on planktonic growth, in turn largely determined by transport time in the water column (with an unquantifiable food-supply component, obviously). Longer transport times are expected when habitat availability is poor (later in the season for coastal species). In fact quite extreme examples are now known to exist even in coastal species, which can, for example, lead to differences in PII at settlement of **122 microns** between the smallest and largest seasonal means for postlarval plantigrades, translating to a 300% or 3-fold difference in estimated masses, based on these seasonal means for *Prodissoconch II*, in Martel et al 2014*

<https://doi.org/10.2983/035.033.0213>

We thank the reviewer for pointing us towards the Martel 2014 paper, and have now included it in our manuscript.

Line 308

*please change “...dependent and that these mussels are able to delay their metamorphosis while continuing to grow.” to something along the lines of “...dependent, so providing further evidence of a capacity to delay metamorphosis while continuing to grow, as has been previously suggested for *G. childressi*, based on larval growth experiments (Arellano et al 2009*). * <https://www.journals.uchicago.edu/doi/10.1086/BBLv216n2p149>*

We have adjusted the text in line 274 of the revised manuscript and it now reads: “*This inconsistency in shell lengths between developmental stages indicates that metamorphosis begin is not dependent on size, and provides further evidence for the ability of mussels to delay metamorphosis while continuing to grow, as previously suggested for *G. childressi* [33].*”

Line 310

A more up-to-date reference on this (mentioned above) is the Martel ref, which should be included here. The Bayne et al 1965 reference demonstrates a capacity to delay metamorphosis under experimental conditions. Martel et al 2014 demonstrate that this is a real phenomenon and has a bearing on seasonal recruitment patterns.

We thank the reviewer for pointing us towards this reference and have incorporated it into the text.

Line 312

A good reference here would be Young et al 2012, a dispersal modelling paper ground-truthed by larval developmental data on *Gigantidas chidressi* from Shawn Arellano's PhD work * <https://doi.org/10.1093/icb/ics090>*

Another good reference, also now included, thank you.

Line 318

The vesicles are happy with their new home...

Elsewhere in the MS you say they are in the "stomach epithelium". Again, I believe its neither, but rather the main body of the digestive gland, which is currently not presented.

Please refer to our answer above to comment: "line 192".

Line 320

change to "fuel energy-demanding metamorphoses" metamorphosis isn't really sustained...

We agree and have changed the text as suggested: "*In early developmental stages of mussels from the Mytilidae, lipid vesicles serve as storage compounds to fuel the energy-demanding process of metamorphosis [18] and they likely have a similar function in Bathymodiolus and Gigantidas.*" (Line 287 in revised manuscript)

Line 326

An obvious comparison to made here is between Bathymodiolus/Gigantidas and other members of the Bathymodiolinae, instead of (or at least, in addition to) more distant shallow-water relatives. Laming et al 2014 and Laming et al 2015 describe ontogenetic patterns of development AND the acquisition of symbionts in two species, Idas modiolaeformis, and "Idas" simpsoni. Discussing how ontogenetic development and symbiont acquisition in the genera Bathymodiolus and Gigantidas compare with Idas or Idas-like mussels for which similar data are available would make for a much more meaningful discussion. It would provide an opportunity to speculate on the evolution of symbioses in bathymodioline mussels under various reducing conditions for example.*

** Note: a putative new genus for this species, Nypamodiolus, has been proposed by Justine Thubaut and colleagues in 2013 but the formal description remains as-yet unpublished Consider the differences:*

1) Bathymodiolus / Gigantidas have large body sizes, are highly productive, live on highly energetic vents and seeps with much greater supply of reduced compounds (higher concentrations and greater longevity), possess intracellular symbioses with acquisition as plantigrades, and ultimately have a highly reduced linear gut

2) Idas (sensu lato) have much smaller sizes, are somewhat less productive, can live on seeps but are predominantly found on organic falls which are a more ephemeral with markedly lower sulphide concentrations, possess extracellular symbioses (less "evolved", but well adapted to this habitat?) and retain one to two loops in the gut (i.e. more than species in the current study's but less than true, heterotrophic mussels).

We thank the reviewer for these points and have included the suggested references. As the reviewer points out, the discussion about the effects of different reducing conditions on the evolution of bathymodioline symbioses would be speculative. The speculative nature of such a discussion, together with the length restrictions of articles in this journal led us to decide against including such a discussion.

Line 326-328

The concomitant onset of gill hypertrophy in bathymodioline mussels and acquisition of symbionts, has also been described in *Idas modiolaeformis* by Laming et al 2014: "...**hypertrophied gills**, a style sac and a recurrent loop (identified during dissection) in the microvilli-lined intestinal tract **were first recorded** with certainty in *Idas* 4 (**0.57 mm**) and then in all larger individuals examined with histology..." and "The smallest SL at which putative bacteria were identified was **0.59 mm** (*Idas* 7, **by TEM**, Fig. 5l, m) arranged in a thin extracellular monolayer only." So, the appearance of bacteria in the gills of these smaller mussels also coincides with the onset hypertrophy. In that study, it wasn't clear whether the onset of hypertrophy paved the way for bacterial colonisation, or whether bacterial colonisation acted as a cue, initiating cellular hypertrophy in the gills (as in your hypothesis). While the data you present clearly indicates that hypertrophy and bacterial colonisation are tightly linked spatially and temporally, occurring on a cell-by-cell basis, no evidence is provided herein of the symbionts direct involvement as a cue or a mediator (or any other terms that suggest the presence of symbionts is a necessary prerequisite for the cytological / morphological changes to occur), though this is a very plausible working hypothesis for future manipulative experiments. So you need to be less deterministic when discussing this as it remains a hypothesis

We thank the reviewer for this comment and agree that it remains a hypothesis that the symbionts induce hypertrophy. We also agree that it is correct if we write that hypertrophy is tightly linked to symbiont colonisation, and remains unclear if the host or the symbionts mediate this morphological change. We have changed the text in line 296 of the revised manuscript as follows: "If the symbionts of *Bathymodiolus* and *Gigantidas* actively induce these cellular changes or if these are a response by the host to symbiont colonisation remains unresolved, but our data shows that these processes were tightly linked spatially and temporally."

Line 328

maybe less wordy to say "The loss of cilia and microvilli during hypertrophy has been reported for many bacteria invading a wide range of epithelia" Or at least change "remodelling" to "adaptation"

We revised this sentence to: "Observations of effacement of cilia and microvilli have been reported for a wide range of bacteria that invade epithelia, particularly pathogens, and for these it also remains to be shown if the bacteria or the host drive these processes [39-41]." (Line 377 in revised manuscript)

line 332

The near-linear rudimentary digestive system of adults is well known and pervasive in large bathymodioline mussels. What is new is that you provide evidence for a transition from an apparently fully functional gut in plantgrades and post larvae to a simple gut in young adults Therefore you need to include previous knowledge on the adult gut "Although it has been known for several decades that the symbionts of *Bathymodiolus* supply them with nutrition **and that adults possess a rudimentary gut only**, nothing was known about the development of the digestive system in these hosts.

We agree and have revised accordingly: "Although it has been known for several decades that the symbionts of *Bathymodiolus* and *Gigantidas* mussels supply their host with nutrition, and that adults possess only a rudimentary gut, nothing was known about the development of the digestive system in these mussels." (Line 304 in revised manuscript)

Line 338

Does the gut not continue to develop and thus become more convoluted in Mytilus?

Also, reference please.

We thank the reviewer for this comment. With this sentence we wanted to emphasise that in mytilids such as *M. edulis*, such drastic morphological changes after metamorphosis, like the straightening of the digestive system, as we observed in our study, are not known. Besides the size of different organs of the digestive system, the overall blueprint in *M. edulis* is similar between late plantigrades, juveniles and adult mussels (Bayne 1971, Eggermont et al. 2020 and Purdie 1887). We changed the sentence to: *“This transformation is striking, as in mytilids like M. edulis, such drastic morphological changes after metamorphosis are not known [42].”* (Line 310 in revised manuscript)

Line 339

I dislike the use of the words "remodel" or "streamlining", they are not particularly biological and while the occasional use of such words can be effective from a scientific communication perspective, overuse is not to be encouraged. Also in this context they are not particularly accurate. "Remodelling" suggests that the gut is actively adapted in the later stages of development (which I would argue against, for energy budget reasons, see below), while "streamlining" suggests the gut is somehow improved or made more efficient, which is of course the opposite of the reality. In my opinion, the most parsimonious explanation is that this change is as a result of the cessation in further gut development (rather than anatomical rearrangement), and that the straightening of the gut is actually occurring passively, the intestinal loop being unfolded as the mussel increases substantially in length. I say this because this would require minimal energy investment in comparison to a cryptic metamorphic process which would be energetically demanding, and because growth in mussels is posteriorly directed which would act to elongate and straighten this region of the gut.

We removed the words “remodel” and “streamlining” throughout the manuscript and changed this sentence to: *“In Bathymodiolus and Gigantidas, the straightening of the digestive system did not coincide with the first stages of symbiont colonisation or metamorphosis, but rather occurred during the transition from the plantigrade to the juvenile stage, well after metamorphosis and only when these hosts had become fully colonized by their symbionts (Figure 4).”* (Line 312 in revised manuscript)

Line 340-341

“Conspicuously, in Bathymodiolus this remodelling of the digestive system did not coincide with the first stages of symbiont colonization or metamorphosis, but rather occurred during the transition from post-larvae to the juvenile stage, well after metamorphosis...” Well, the change takes place at different stages depending on which naming convention is used, so I'm not sold on this "transition to juveniles" argument. Better to say that it coincides with a notable increase in animal size, as mentioned above. It is interesting that the cessation in gut development and establishment of symbiosis in the gill are concomitant processes: discovering how this is coordinated and to what degree this is mediated by the host and (presumably) influenced by the presence of bacteria would be a great leap forward for the field. Of course the ideal would be a squid-Vibrio-type experiment that tracked ontogenetic development after settlement in under the presence and absence of symbiont candidates to

assess whether the acquisition process is an obligatory prerequisite for host development and if so, is their role simple to provide a cue for host signaling or are they mediating changes in the gill and/or whether these changes ultimately are required for gut development to stop. Sadly, such an experiment would face considerable obstacles, given the obvious technical and culturing constraints (both for the host and symbiont). Also, remove "Conspicuously", it is unnecessary.

We agree with the reviewer that it would be a great step forward if we could gain a better understanding of whether developmental processes are driven by the host, by the symbiont or both. We also agree with the reviewer that experiments for determining these processes are currently not feasible given that bathymodioline symbionts have not yet been cultured. What we have done in our study is pinpoint when the digestive system changed from a looped to a straight form, namely after symbiont colonisation began and metamorphosis was completed. Based on our definitions of developmental stages (see our answer above to comment "line 33/34"), this corresponds to when the mussel was in transition from the plantigrade to the juvenile stage.

Line 342-343

"We hypothesize that the streamlining of the digestive system is induced by the shift in nutrition of these hosts" Hypothesising is all you can do for now but for what it's worth, it's a sound hypothesis in my opinion (though as I have mentioned, for me this would be a relatively passive process). The alternative, which is that the straightening of the gut is somehow genetically "hardwired" (now I'm using non-biological terms!), seems unlikely, given the relatively recent evolution of this group. Also, the word streamlining is counter-intuitive

We agree with the reviewer that for now we can only speculate about this, and also agree that our hypothesis is likely.

Line 346 – 349

*you are "postulating" a scenario that has already been proposed before and yet it is presented here as a novel idea. I must therefore **insist** that you mention previous studies that have 1) already stated exactly this for bathymodiolins generally (Laming et al 2018) and 2) demonstrated it in smaller bathymodioline mussels empirically (i.e. Laming et al 2014; 2015). To save you the time looking and for the benefit of the editor, I'll quote the various papers concerned Laming et al 2014 on *Idas modiolaeformis*: "... isolating the probable period of acquisition to the post-settlement phase of development. The **absence of bacteria in plantigrades, whose digestive system is well developed, indicates that planktotrophic larvae and post-larval plantigrades are both exclusively heterotrophic ...** This allometric [gill] development would increase the symbiont carrying capacity considerably. It is unknown whether heterotrophic nutrition remains important at this stage; however, the apparent **increase in the relative volumes of gill tissue versus digestive tissue** in larger specimens of the same species found upon carbonate crust (Gaudron et al. 2012) may indicate a **switch in nutritional mode from heterotrophic to predominantly chemosymbiotic with increasing size.**" Laming et al 2015 on "*Idas simpsoni*: "Considering bacterial signals were **entirely absent from all epithelia in plantigrades of "I." simpsoni**, chemosymbiotic larval nutrition seems unlikely, though the pre-acquisition of a very low seeding population of bacteria cannot be entirely ruled out. Thus the **larvae of "I." simpsoni** are very likely to be **obligate heterotrophs**. ... Within a narrow range of SLs following their initial appearance in very small juveniles, symbionts in the current study appear to increase rapidly in abundance on the external, non-ciliated surfaces of gill epithelia reaching high*

abundances based on microscope observations (e.g. Fig. 4; Fig. 6B even at SLs 846 and 974 mm)...” Then, since these mussels are subject to lower sulphide emissions: “As young juveniles, the presence of a developing gut and a growing number of bacteria associated with the gill suggests a potential to ingest and assimilate particulate or dissolved organic matter as well as utilise possible symbiont-based nutrition... ..If the SOX symbionts found in specimens of “*I.*” *simpsoni* in the current study are indeed assimilated as a source of carbon and energy, then in highly sulphidic conditions the importance of an anatomically functional gut may be reduced. However, the release of reduced chemical compounds at reducing habitats is often highly heterogeneous (e.g. Le Bris et al., 2008), wherein mixotrophy involving filter-feeding and chemosynthetic symbiont nutrition might provide a means to ensure that a continuous energy resource remains available.” and finally from the Laming et al 2018 review “Adult bathymodioline mussels of most species rely on their bacterial partners for much of their nutrition, with some species retaining rudimentary digestive systems only. That said, growing evidence points to a retained capacity to filter feed (Page et al., 1991). Studies in aquaria have demonstrated the assimilation of carbon from algae, radio-labeled water-borne bacteria, and naturally-occurring plankton (Pile and Young, 1999). The retention of a fully-formed digestive system appears to be the norm in smaller-sized bathymodioline mussels from organic falls (Gustafson et al., 1998; Thubaut et al., 2013a; Laming et al., 2014, 2015a). In *I. modiolaeformis* and “*I.*” *simpsoni*, live examinations under a dissecting microscope reveal the mediated convection of debris in the vicinity of the gills in a way similar to that of shallow-water mussels (Supplementary Material 2). Given the earliest, post-settlement developmental stages of these species possess components of a functioning digestive system and are either aposymbiotic or display low abundance of symbiotic bacteria, the diets of juvenile bathymodioline mussels are likely to undergo a transition from heterotrophy-to-mixotrophy-to-chemosymbiosis with increasing size. This is supported by a model-based study examining *B. azoricus* from the relatively shallow hydrothermal vent site Menez Gwen (800 m), in which a shift toward greater reliance on symbionts during the growth of the host was hypothesized (Martins et al., 2008). Similar transitions are thought to take place in other chemosymbiotic molluscs (e.g., *Gigantopelta* spp., Chen et al., 2017).” To completely ignore these studies findings and “postulate” a scenario that has already been proposed by both these and other studies (see also the output from the Deep-sea lab in the Azores which has produced several live-animal studies on trophic biology and immunology in *B. azoricus*) some five years ago, is an oversight at best, bordering on unethical.

We thank the reviewer for this comment. When referring to “*Bathymodiolus*” in our original manuscript, we focussed on the early life stages of the three species we analysed, but did not make general statements about all bathymodioline species. We never disputed (or discussed) that adult bathymodioline mussels likely complement their nutrition with filter-feeding. As Laming et al. point out themselves in their 2018 review in the section in which they discuss bathymodioline mussels from vents and seeps (called Bathymodiolinae clade in their review): “*However, as no aposymbiotic specimens have ever been found, the exact timing of symbiont acquisition remains unresolved*”. We are therefore in good agreement with Laming et al. (2018) in our conclusion that for symbiotic vent and seep mussels, no data about how their larvae acquire nutrition was available prior to our study. We have revised this section to more clearly communicate these points, as follows: “*We hypothesize that the straightening of the digestive system in the vent and seep mussels we analysed here was induced by their shift from filter feeding to gaining nutrition from their symbionts.*” (Line 320 in revised manuscript)

Line 350

Consider changing “*evolutionary strategy to reduce the energy*” to “*evolutionary adaptation to minimise the energy*”. The term *strategy* in this context is suggestive of a master plan.

We changed our text as suggested: “*The straightening of the digestive tract could be an evolutionary adaptation to minimise the energy needed for maintaining the digestive tract once the majority of nutrition is gained through intracellular digestion of symbionts in the bacteriocytes.*” (Line 322 in revised manuscript)

Line 353-355

*yet again, smaller bathymodioline species from other reducing systems are not discussed. Aside from the aforementioned Laming et al studies that include data on retained looped intestines (one in *I. Modiolaeformis* and two in "*I. simpsoni*") there is also *I. iwaotakii* from wood (Thubaut et al 2013b <https://doi.org/10.1371/journal.pone.0069680>) and *Tamu fisheri* from seeps (Gustafson et al 1998 <https://bit.ly/341Mvaz>).*

Line 355-357

Indeed, and this is also true for organic-fall species as organic falls are likely to occur in regions that see a greater overall flux of detritus, such as submarine canyons where the retention of a more functional looking gut (anatomically) could reflect active, ongoing selective pressures (i.e. a gut remains a requirement for survival because mussels are more mixotrophic), particularly if symbiont productivity is subject to intermittent or limited supplies of reduced-fluid emissions..

We thank the reviewer for these two comments. We revised our text and added more examples to this discussion point. Now it should be clearer that some bathymodioline mussels keep their complex digestive system and others reduce it.

*“Not all bathymodioline mussels have a straightened digestive system. *Idas* intestines are complex with one or more loops [7, 15, 44]. These small mussels are commonly found at organic falls, which typically have higher inputs of organic matter than vents and seeps. Also, symbiont abundances are lower in *Idas* than in the large vent and seep bathymodioline mussels, indicating that *Idas* mussels may depend more on filter feeding than their larger relatives. Intriguingly, the relatively large adults of the bathymodioline genus *Vulcanidas* have a pronounced looped intestine despite high symbiont abundances [45]. *Vulcanidas* inhabits relatively shallow hydrothermal vents close to the photic zone (140 m compared to 1000–2500 m depth for the mussels in our study) where more organic material from the photic zone is available for their nutrition. It is thus likely that filter feeding plays a greater role in the nutrition of *Idas* and *Vulcanidas* than in the three species analysed in this study. These findings raise the question whether the straightening of the digestive tract is a conserved developmental trait in only some genera, or if the environment and availability of organic matter and other non-symbiotic food sources drives the morphology of the digestive tract in bathymodioline mussels.”* (Line 426 of the revised manuscript)

Line 357-359

This is the exactly the question that needs answering, as I mention above (i.e genetic predisposition versus environmental factors, even epigenesis?)

We agree with the reviewer that this is one of the most interesting follow up questions but it exceeds the scope of this study.

Line 361 Figure 4

*I'm curious, what was the inspiration of the schematic in (e)? Maybe it was simply "convergent inspiration" but it feels **very** familiar... For the editor:*

https://www.frontiersin.org/files/Articles/385457/fmars-05-00282-HTML-r1/image_m/fmars-05-00282-g001.jpg Developed from the schematic found on page 91 of the following PhD thesis: https://tel.archives-ouvertes.fr/tel-01135209/file/pdf2star-1417090206-These_archivage_3160120.pdf

Given that the objective of our figure was the same as that of Laming et al., namely to provide the reader with a diagram of the lifecycle of deep-sea mussels, it is inherent that portrayals of mussel life stages will share basic similarities. In our opinion, our figure differs substantially from that of Laming et al.

Line 369

*You've overlooked the last author: please change to "Laming, Gaudron **and Duperron** [7],"*

We changed the authors accordingly.

Line 379-380

Particle sorting is not coordinated by the gill when they are only rudimentary gill bars. This is performed by the highly ciliated foot.

We thank the reviewer for this comment. We changed the text to make it clearer that we hypothesize that the gills of Bathymodiulus mussels change their function in a similar manner as the gills of *M. edulis* based on their similar morphology. After the velum is degraded, the gill changes from being a purely respiratory organ in pediveligers to becoming responsible for water current production in late metamorphosing mussels and then for sorting particles in late plantigrades and juveniles. *"Given that the developmental changes in the morphology of the bathymodiolin gills are so similar to those of M. edulis [7, 15, 18, 29], it is likely that the functional roles of the gills in Bathymodiulus change in a similar manner as in M. edulis: The pediveligers gills of Bathymodiulus are used for respiration only, and then become responsible for water current generation in late metamorphosing mussels, and the sorting of particles in plantigrades and juveniles. We hypothesize that this functional shift of the gills plays an important role in enabling the symbionts to adhere to the gill epithelium and initiate symbiont colonisation."* (Line 352 in revised manuscript)

Line 388-389

*Sound reasoning and I would definitely agree. However here, there is **yet another missed opportunity** to make comparisons with other reducing habitats characterised by lower reducing fluid emissions. Acquisition in these species of large bathymodiolin is even more rapid than the already relatively fast acquisition of symbionts by "Idas" simpsoni on bones, which in turn was more rapid than acquisition in Idas modiolaeformis on wood falls. This could be driven by the availability of proximate symbiont partners in the environment, likely influenced by the availability of reducing fluids (for free living candidates) and/or the densities of proximate compatible symbiotic species (mainly through lateral transmission from conspecifics). Consider: at vents there are both high emission rates and high densities of conspecifics, at carcass falls there are lower emission rates but still high densities of conspecifics, while on wood falls both emissions and conspecific densities are low.*

We agree that a number of factors, including symbiont abundances within hosts, the availability of energy sources for the symbionts, and the abundances of free-living symbiont life stages will affect

colonization rates. Given the word limits for this manuscript and the speculative nature of the role these factors play, we prefer not to expand our discussion of this paragraph.

Line 390-392

maybe add something about the availability of symbionts locally? e.g. “ This is made possible through the immediate availability of symbiont candidates and could reflect...” or something similar.

We kept this unchanged, please see above our answer to comment “Line 388-389” for details.

Line 400-401

Have you any sense as to why this might be and/or why MOX do not colonise cells independently?

We thank the reviewer for this interesting question, but can only speculate for now. We could envision that the SOX symbionts prime the host cell in a way that allows the MOX to then colonize the host cell. On the other hand, we know that in mussels that only harbour MOX, these symbionts can colonize host cells on their own. We'd love to understand the molecular and cellular processes behind this!

Line403/404

“early planktonic larval stages” Please rephrase, as this reads like they are an early stage in planktonic development when they are of course the last stage.

We thank the reviewer for this comment. Due to length restrictions we had to move this part to the supplementary discussion. The text reads now as follows: “*We therefore assume that these late planktonic larval stages were in the process of settling on the sea floor.*” (Line 154 in the supplement).

Line 406-408

This is perfectly possible. However, it's worth noting that this would likely operate over fairly small spatial scales. The study you cite is also for a tubeworm but there is already evidence that mussel larvae employ vibriotaxis, chemotaxis with neurochemical signalling and respond to types of current flow (turbulent versus laminar) and substratum characteristics. Vents are likely to have a very particular soundscape and strong chemical and temperature cues from plume dispersion.

We thank the reviewer for providing this additional information. We incorporated some of the information into the text to clarify the different strategies larvae could use to settle and begin metamorphosis. Due to length restrictions this whole paragraph was moved to the supplementary discussion: “*Although all individuals in this study were collected from mussel beds on the sea floor, the pediveligers were always aposymbiotic. We therefore assume that these late planktonic larval stages were in the process of settling on the sea floor. Mussel larvae can react to cues like current flow, water chemistry and temperature, and substratum characteristics that could play a critical role in detecting vent plumes and inducing settlement [2-4]. At smaller spatial scales, bacteria may also play a role in inducing settlement and metamorphosis in marine invertebrates [5]. It is tempting to speculate that the symbiont-like bacteria we observed on the larval shells, or microorganisms in the*

mussel beds, including possible free-living forms of the symbionts, play a role in inducing the settlement and metamorphosis of Bathymodiolus mussels.” (lines 153 – 162 in the supplement).

Line 409-410

As I have already mentioned repeatedly, an obvious opportunity to make comparisons between bathymodioline mussels from different reducing environments that are more or less chemosynthetically productive has been missed, despite the fact that this would enrich the discussion.

We agree with the reviewer and have incorporated examples for mussels living in reducing environments: “*Our analyses revealed that B. puteoserpentis, B. azoricus and G. childressi larvae are aposymbiotic during their planktonic phase and first acquire their symbionts when they transition to a benthic lifestyle. The timing of symbiont acquisition and major developmental changes was similar in all three species, and, with the exception of the digestive system, corresponds to observations of early life stages of Idas [7, 9, 15]. These similarities in developmental biology and symbiont acquisition suggest that these traits are conserved in bathymodiolins. The acquisition of symbionts after settlement allows these mussels to recruit locally adapted symbionts. As the geochemistry of vent and seep environments varies strongly across spatial scales [49], recruiting locally adapted symbiont populations would confer a strong fitness advantage to these hosts.*” (Line 382 in revised manuscript)

Line 412-414

Spatial scales yes. Temporal scales: not so much. The MAR is a slow spreading ridge and is not as ephemeral as has been postulated for vents in fast-spreading regions such as the EPR.

Also not “selective advantage” but “fitness advantage”.

We changed “*selective advantage*” to “*fitness advantage*” and removed the reference to temporal scales, please see line 391 of the revised manuscript.

Line 421

Change to “are [thought / believed] to be triggered”

We changed the text accordingly, please see line 486 in revised manuscript.

Line 429

remove “interwoven” and perhaps replace with “intrinsic” (or similar)

We changed the text to: “... and further our understanding of the **entwined** dialogue between animal hosts and their microbial symbionts.” (Line 407 in revised manuscript)

Finally:

Line 432

Supplementary materials

Some corrections are suggested below for the supplementary material:

First, the anatomical inaccuracies concerning the digestive glands in the CT data and labelling errors in the serial sections **must** be addressed.

Also, please use the term “commissures” not “commissure tissues” and “connectives” not “connective tissues”. Strict terminology exists for neural biology: connective tissues are something different from what you are describing, which are connectives (a particular type of neurite bundle). For correct terminology in neurobiology see Richter et al 2010 <http://dx.doi.org/10.1186/1742-9994-7-29>

THE BAD NEWS: these terms are also incorrect in the CT 3D animations as well, so these annotations needs correcting, sorry! Correct all other mentions of the lipid vesicle locations, as in main text.

In Figure S5:

Are the purple and white glands those that are visible at the base of the foot in these sections, abutting the pedal ganglion? If so, please label them. Also please see additional comments/annotation regarding digestive gland/divertula

In Figure S12

The composite image for b is flipped in it's horizontal axis for some reason...

In Figure S13

Caption states “The gill filaments of a *B. puteoserpentis* planigrade (c) is only colonized by SOX. a.a.m, anterior adductor muscle; ft, foot; gf, gill filament and m, mantle”

We thank the reviewer for their detailed and helpful comments for the supplement. We addressed all suggestions above and adjusted the corresponding text, figures and supplementary videos.

So, can you explain what I'm seeing then in the MOX image for c), because it doesn't look like background signal to me.

The reviewer is correct that there is a fluorescence signal from the MOX symbionts. This signal is not in the gill (which is only colonized by SOX) but from the foot tissue. The cells of the foot epithelia are colonized by SOX and MOX. We revised the figure legend to make this clear.

More generally, although self-evident to me (being very familiar with the techniques used in the study), the supplementary materials could benefit from a brief description of the data format (e.g. "tiff image stack") and the means by which such files are best viewed, as a reader not familiar with such image analyses and handling of files may struggle to view the supplementary material.

We thank the reviewer for their suggestion, we have added a readme file containing the information to access this data, to the public repository which stores the data. The file reads as follows: “To open the image data stored in this project please download the free image processing software Fiji (<https://imagej.net/software/fiji/>). After installation, the image data can either be opened via drag and drop into the status bar or by going to file  import  image sequence  chose the location of the images. The image data in this project is stored as .tif files. Some of these files are saved as one image stack (one tif file) containing multiple z-planes (these can be easily opened via drag and drop) and others are saved as image sequences of multiple .tif files (for these files please use the import function).”

Referee 2

In their study, Franke et al. investigate symbiont acquisition in three deep-sea Bathymodiolus mussel species by means of various microscopic and imaging methods, including synchrotron radiationbased microtomography (SR μ CT). The authors show that the mussels are colonized by their chemoautotrophic symbionts during the plantigrade stage of larval development. Moreover, the study presents first time evidence that the earliest larval stages of Bathymodiolus mussels, pediveliger larvae, are symbiont-free. This is an important new finding which finally substantiates the previous assumption that Bathymodiolus symbionts are horizontally transmitted (i.e., acquired from the environment) rather than inherited from parents to offspring. The results thus contribute another valuable facet to our understanding of how marine invertebrate symbioses are established.

The paper is very well written, I enjoyed reading it. Images and particularly the supplementary videos are of high quality and present a real asset for the reader. The study has a clear research focus and the methods were chosen adequately to answer the research question. In the discussion, the findings are very plausibly interpreted and integrated into the bigger picture, i.e., mussel development and metamorphosis, mode of nutrition, and uptake of locally adapted symbionts. This work provides the basis for a number of interesting follow-up studies.

We thank the reviewer for their kind words.

I have only a few minor comments and questions:

*Replicates and mussel species: In the sampling methods section I missed how many replicates were sampled of each mussel species. This important information is given later in the results, but could also be mentioned here (or at least state something like “refer to the results for replicate numbers”). You examined only 6 *B. childressi* specimens from the Gulf of Mexico, while all other > 250 individuals were from the MAR. This doesn't really allow for conclusions on differences or similarities between mussels from both sites (and you are not attempting to draw such conclusions) – but this would be an interesting question for future studies with more replicates: Are there differences in symbiont acquisition between Bathymodiolus species from geographically distinct locations? (You could mention this in the outlook.)*

We agree that the method section needed improvement. We added a new section with the details of the correlative workflow and an overview table that includes how many individuals from which species were analyzed with which method (Table 1). We also agree that it would not be wise to focus on similarities and differences between the MAR and Gulf of Mexico species, given that we only had six specimens for the latter. However, in the six *G. childressi* individuals we examined, the timing of symbiont acquisition was similar to that of MAR mussels, namely after metamorphosis in plantigrades.

We also revised the Discussion, and addressed the question if bathymodioline mussels from different locations acquire symbionts similarly: “*Our analyses revealed that *B. puteoserpentis*, *B. azoricus* and *G. childressi* larvae are aposymbiotic during their planktonic phase and first acquire their symbionts when they transition to a benthic lifestyle. The timing of symbiont acquisition and major developmental changes was similar in all three species, and, with the exception of the digestive system, corresponds to observations of early life stages of *Idas* [7, 9, 15]. These similarities in developmental biology and symbiont acquisition suggest that these traits are conserved in bathymodiolins.*” (Line 382 of the revised manuscript)

Biological replicates and symbiont colonization: Your finding that pediveliger larvae are aposymbiotic is based on two individuals (page 12, line 241) + 5 individuals (line 245), is that right? Shouldn't this be made more transparent, considering that the aposymbiotic

pediveligers are the most central and novel finding of this study? Also, when you say “In plantigrades 15% ... of ... cells were colonized by symbionts” (page 12, line 241-242), which means “all plantigrades had symbionts”, this is a bit misleading because it is based on only one individual, and seems to contradict the next sentence, where all 5 plantigrades were aposymbiotic. Please rephrase to make this clearer.

We agree with the reviewer that it is important to make the number of individuals we investigated clearer. We analysed nine individuals (4 pediveliger, 2 plantigrade and 3 post-larvae) with our correlative imaging approach (Table 1). This correlative approach formed the basis for our light microscopy analyses of 23 additional individuals (4 pediveliger, 6 metamorphosing mussels, 6 plantigrades, 4 juveniles and 3 adults), in which we identified morphological characteristics of the host cells (hypertrophy, loss of cilia and microvilli) indicative of symbiont colonisation. The correlative imaging approach was complemented with FISH analysis of seven additional *B. puteoserpentis* specimens. We revised the Method section and added Table 1 with the numbers of individuals analysed by which method.

The new method section of the main text reads as follows:

“Section series were screened by eye to identify individual host cells and predict symbiont colonisation state based on three morphological characteristics: hypertrophy, loss of microvilli and loss of cilia (Table 1, Supplementary Methods i). The location and symbiont colonization state of the analysed host cells was marked using Cell Counter in Fiji [26]. The analysed sections were re-sectioned and the same fields of view were recorded with TEM (Figure S2). To validate the LM-based predictions, symbionts were identified in TEM images based on their morphology (Video S5).” (Lines 120 – 126 in revised manuscript)

Furthermore, we added a more detailed description of the correlative work flow to the supplementary methods, which reads as follows: *“We screened section series of 32 specimens from the three analysed species (Table 1) using light microscopy and by eye identified individual cells and predicted their symbiont colonisation status based on the presence / absence of the morphological characteristics hypertrophy, loss of microvilli and loss of cilia. The location of the cells and their colonization status was marked using Cell Counter in Fiji [14]. Semi-thin sections of nine specimens were re-sectioned, the same fields of view were recorded with TEM, and LM and TEM data was overlaid (Figure S2). To validate the LM-based predictions, symbionts were identified in TEM images based on their morphology. We performed this workflow on samples imaged with either both SR μ CT and section series, or samples only prepared as section series (Video S5). We complemented this approach with FISH on sections of seven additional *B. puteoserpentis* specimens.” (Lines 289 – 299 of the supplement)*

Mode of nutrition and gill function: You could go into a bit more detail when explaining the proposed shift of functions of gills and digestive tract during morphological changes. How do the larvae feed before they start filter feeding? On page 11, line 211 you suggest that gills in plantigrades develop from purely respiratory organs to respiratory and filter feeding organs. But on page 18, line 346-348 you postulate that “the first life stages of Bathymodiolus rely on filter feeding”. That’s a bit contradictory. Also, when you mention that lipid vesicles serve as storage compounds (page 17, line 320) – does this mean these larvae only feed on internal lipid stores? Please provide some more explanation.

We thank the reviewer for this comment and have clarified the text accordingly: *“In the digestive gland and stomach epithelia, the number and volume of lipid vesicles decreased from 12.8% of the soft-body volume in pediveligers, to 3.9% in metamorphosing mussels, 1.5% in plantigrades, and were no longer present in juveniles and adults (Figure 1 a-d and Table S6). The organs involved in filter-*

feeding also changed. In pediveligers, the main feeding organ was the velum, which collects particles from the seawater and transports them to the oral labial palps and mouth. After the degradation of the velum, particle sorting was taken over by the highly ciliated foot (Figure S5, S8 and S9) in metamorphosing mussels, and by the gills in late plantigrades and juveniles.” (Lines 180 – 188 in revised manuscript)

In response to the reviewer's question about lipid vesicles: *Bathymodiolus* larvae are planktotrophic and any yolk provided from the parents is limited and only sufficient to last until the larvae have developed their own digestive system and feeding apparatus. The pediveliger feeds by collecting particles with the velum and during the filter-feeding planktonic larval stage, energy storages in form of lipid vesicles are built up. These lipid vesicles serve as storage compounds for the highly energy consuming process of metamorphosis (Please see supplement lines 58 – 63). For the change of feeding over the development please refer to lines 253 – 262 of the revised manuscript.

Bacteria on larval shells (page 21, line 406): I assume that these bacteria were only visible in TEM and not identified as SOX and MOX by FISH? Why?

We thank the reviewer for this comment. The reviewer is correct that we only identified the bacteria on the mussel's shell via TEM and not FISH. Glutaraldehyde used for TEM fixation provides better fixation compared to the paraformaldehyde fixation used for FISH samples. Furthermore, the resolution of TEM allowed us to detect single- to few bacteria, while the strong autofluorescence of the shell made the detection of single bacteria with FISH extremely difficult. We added the following sentence to the legend of Figure S16: “FISH analyses of these shell-associated bacteria were not possible because of the strong autofluorescence of the shell.”

Page 14, line 256: “Expanding our analyses to all three Bathymodiolus species...” – referencing Figure 2 in this sentence is misleading, since the figure only shows B. puteoserpentis (not all three species).

We agree with the reviewer. We added a new Supplement Figure S15 that shows data from *B. azoricus* and *G. childressi*. Furthermore, we revised the text and now refer to both figures: “Our correlative workflow revealed that pediveliger were aposymbiotic in all three mussel species (Figure 2 a, d and g and S15 a and c).” (Line 229 in revised manuscript)

Figure 4: I suggest to remove the background (a hydrothermal vent site?), because it is poorly discernable and diminishes readability of the illustration and text on the lower right

Good point, thank you! We have removed the background and agree that the figure is now much easier to read.

Check for typos, spelling, etc., e.g.,

o Abstract, line 33: period missing

o Methods, line 120: 16S rRNA; line 146: “[22-25]van...”

o Results, page 14, line 276: it should read “Figure 2b, e, h, j-m”

o Figure 1 caption, line 201: “Note the different scale bars in a-c and d” is a bit

confusing. Why not just “Note the different scale bars”?

o Figure 3 caption, line 296: host tissue (singular)

o Page 20, line 369: check Proceedings B reference format

o Conclusion and outlook: Not sure if this is on purpose, but you are using “revealed” three times in the first two sentences. Might want to consider rephrasing.

o Throughout: check consistency of your British English (colonization, visualization)

We thank the reviewer for pointing out typos and spelling mistakes. We corrected them as suggested.

Referee: 3

Comments to the Author(s)

This is an important contribution to our understanding of morphological development in coordination with the timing of symbiont acquisition in a widespread and highly-studied group of chemosymbiotic mussels from deep-sea hydrothermal vents and cold seeps. Through meticulous study of an impressive number of specimens, and using a clever combination of microscopy techniques, the authors have provided clear descriptions of how organs develop from the late larval to the adult stage (through metamorphosis), and have managed to establish, for the first time, the timing and process of symbiont uptake in this group of bivalves. The investigative approach used combines detailed reconstructions of organs in whole specimens along with histological analysis, FISH and TEM, which allow confirmation of symbiont type, presence, and apparent endocytosis and intracellular lysis in various organs. In particular, the combination of histological images and reconstructed tomographs, as presented in the supplementary videos, provide exceptional detail and a clear understanding of the processes described in the text.

*The analysis performed also yielded important data with regards to relationships between developmental stage and shell size in *Bathymodiolus*, as well as the relative timing of acquisition of sulfur-oxidizing and methanotrophic symbionts in *B. puteoserpentis*. Interestingly, the analysis revealed how the morphology of the digestive tract changes during development, likely due to a change in host nutrient acquisition modes. The observation of some SOX- and MOX-like bacterial morphotypes on the shells of *B. puteoserpentis* pediveliger and postlarvae is also of interest and should stimulate further work.*

The text is clear, logical and sufficient detail is provided to enable non-experts to understand the study. The figures are of exceptional quality, and I particularly appreciate the highly informative Figure 4 which elegantly summarizes the study's findings.

This study is relevant to multiple fields of study (from animal-microbe symbiosis to marine ecology, developmental biology, and evolution, for example) and should inspire further research in this exciting area.

We appreciate the kind words by the reviewer.

I have a few minor comments:

- *line 33: period missing at end of sentence.*
- *line 36: these mussels still rely to some extent on filter-feeding in the adult stage. Perhaps reword to something like: "is likely linked to the decrease in importance of filter-feeding as intracellular digestion of symbionts becomes possible."*
- *lines 42-43: bathymodiolins also occur in other habitats (whale falls and wood falls)*
- *line 69: "late larval phase": perhaps the term "post-larval" would be more appropriate here?*
- *line 77 (and 175): perhaps specify that these "very early, aposymbiotic life stages" were at the post-settlement stage (I was left with the impression that you had collected planktonic stages).*
- *line 463: "and" instead of "und"*

Supplementary materials:

- *line 44: nervous*
- *line 149: "immersed" rather than "dissolved"*
- *line 253 (legend, Fig S2): add v.g. and o.l.p.*

We thank the reviewer for the comments and corrected them as suggested.

- *line 111: how did you mount semi-thin sections on a resin block? Using an adhesive?*

We now explain how we mounted our sections in the Supplementary Methods c, and have provided a reference:

“For TEM, semi-thin sections were mounted on a freshly trimmed resin block by placing the semi-thin section on applying a drop of Milli Q on the resin block and placing the semi-thin section on the drop of Milli Q. After drying the resin block for 1 h at 40 °C the section was fully adhered to the resin block. Ultra-thin (70 nm) sections were cut on a Leica UC7 ultramicrotome (Ultracut UC7 Leica Microsystem, Austria) and mounted on formvar-coated slot grids (Agar Scientific, United Kingdom) [7]. Sections were contrasted with 0.5% aqueous uranyl acetate (Science Services, Germany) for 20 min and with 2% Reynold’s lead citrate for 6 min.” (Lines 205 – 212 in the supplement)

- *line 117: were these PFA-fixed specimens decalcified prior to dehydration and embedding?*

Yes, the samples were decalcified using EDTA. We have revised the corresponding section in the methods: *“Paraformaldehyde fixed samples were decalcified (Supplementary Methods e) and DOPE-FISH was performed on embedded and sectioned samples using general and specific probes (Table S3, Supplementary Methods d) [22].”* (Line 106 in revised manuscript)

Manuscript with Track changes

Coming together – symbiont acquisition and early development in deep-sea bathymodioline mussels

Maximilian Franke^{1,3}, Benedikt Geier¹, Jörg U. Hammel², Nicole Dubilier^{1,3*}
and Nikolaus Leisch^{1*}

¹ Max Planck Institute for Marine Microbiology, Celsiusstr. 1, 28359 Bremen, Germany

² Helmholtz-Zentrum Hereon, Institute of Materials Physics, Max-Planck-Str. 1, 21502 Geesthacht, Germany

³ MARUM—Zentrum für Marine Umweltwissenschaften, University of Bremen, Leobener Str. 2, 28359 Bremen, Germany

*Corresponding authors

Nikolaus Leisch, e-mail: nleisch@mpi-bremen.de

Nicole Dubilier, e-mail: ndubilie@mpi-bremen.de

Max-Planck-Institute for Marine Microbiology

Celsiusstr.1, D-28359 Bremen, Germany

Phone: 0049 (0)4212028

Fax: 0049 (0)4212028760

Keywords: larvae, aposymbiotic, morphology, 3D, symbiosis, host microbe interaction, invertebrates, bivalves, anatomy

Abstract

How and when symbionts are acquired by their animal hosts has a profound impact on the ecology and evolution of the symbiosis. Understanding symbiont acquisition is particularly challenging in deep-sea organisms because early life stages are so rarely found. Here, we collected early developmental stages of three deep-sea bathymodioline species from different habitats to identify when these acquire their symbionts and how their body plan adapts to a symbiotic lifestyle. These mussels gain their nutrition from chemosynthetic bacteria, allowing them to thrive at deep-sea vents and seeps worldwide. Correlative imaging analyses using synchrotron-radiation based micro-tomography together with light, fluorescence and electron microscopy revealed that the pediveliger larvae were aposymbiotic. Symbiont colonisation began during metamorphosis from a planktonic to a benthic lifestyle, with the symbionts rapidly colonizing first the gills, the symbiotic organ of adults, followed by all other epithelia of their hosts. Once symbiont densities in plantigrades reached those of adults, the host's intestine changed from the looped anatomy typical for bivalves to a straightened form. Within the Mytilidae this morphological change appears to be specific to *Bathymodiolus* and *Gigantidas* and is likely linked to the decrease in importance of filter-feeding when these mussels switch to gaining their nutrition largely from their symbionts.

Introduction

Mutualistic interactions between hosts and their microbiota play a fundamental role in the ecology and evolution of animal phyla. By associating with microbial symbionts, animals benefit from the metabolic capabilities of their symbionts and gain fitness advantages that allow them thrive in habitats they could not live in on their own [1]. Prime examples for such symbioses are bathymodioline mussels, which occur worldwide at cold seeps, hot vents, and whale and wood falls in the deep sea. Mussels of the genera *Bathymodiolus* and *Gigantidas* house chemosynthetic bacteria in their gills, in cells called bacteriocytes [2]. In these

nutritional symbioses, the bacteria use reduced compounds in the vent and seep fluids as an energy source for carbon fixation, which in turn provides nutrition to their hosts. Two types of symbionts dominate bathymodioline mussels, sulphur-oxidizing (SOX) symbionts, whose main source of energy are reduced sulphur compounds, and methane-oxidizing symbionts (MOX), which gain their energy from oxidizing methane [3].

The transmission of symbionts from one generation to the next plays a central role in the ecology and evolution of mutualistic associations [4]. Symbionts can be transmitted vertically from parent to offspring, intimately tying them to the reproduction and development of their host [4], as known from vesicomid clams [5]. Alternatively, in horizontal transmission, symbionts are recruited each generation anew from the environment, and the symbiotic partners are physically separate from each other before the symbiosis is established [4]. In many hosts that rely on horizontal transmission, the acquisition of symbionts triggers morphological and developmental changes in the host. These can range from tissue rearrangements, known to be symbiont-induced in the *Euprymna* squid-*Vibrio* symbiosis [6], to largescale modifications of host organs, typically through hypertrophy [7], or the development of a novel symbiont-housing organ, like the trophosome of the tubeworm *Riftia pachyptila* [8]. As most of these symbioses are uncultivable, it is still unclear if these developmental changes are mediated by the host, actively induced by the symbiont or a mix of both.

Although bathymodioline mussels have been studied for over 40 years, very little is known about how their symbionts are transmitted, at which developmental stage the symbionts colonize the mussels, and how symbiont acquisition affects the development and body plan of the mussels [9]. It is assumed the symbionts are transmitted horizontally, based on phylogenetic studies that showed a lack of cospeciation between hosts and symbionts, as well as morphological studies that found no evidence for symbionts in the mussels' reproductive tissues [10-14]. In bathymodiolins of the genus *Idas*, commonly found at organic falls as well as seeps, the larvae remained aposymbiotic until they settled and

developed the dissoconch shell, indicating a heterotrophic lifestyle in the larval dispersal phase, and horizontal acquisition of symbionts during or after metamorphosis [7, 15].

Given the high abundance of *Bathymodiolus* and *Gigantidas* at vents and seeps worldwide, it is surprising that definitive evidence of an aposymbiotic early life stage of these mussels is lacking. The earliest life stages described so far had undergone metamorphosis and were already colonized by symbionts, with a well-developed symbiotic habitus that was indistinguishable from adult mussels [16]. As the early life stages of aposymbiotic *Bathymodiolus* and *Gigantidas* mussels have not yet been described, fundamental questions in the acquisition of symbionts in these bathymodioline genera have remained unanswered, including at which developmental stage the mussels acquire their symbionts, if the SOX and MOX symbionts colonize their hosts at the same time, and which developmental changes occur in the mussels at the onset of symbiont colonisation.

In this study, we were fortuitous in discovering very early, aposymbiotic life stages of three bathymodioline species, two from hydrothermal vents on the Mid-Atlantic Ridge (MAR), *Bathymodiolus puteoserpentis* and *B. azoricus*, and one from cold seeps in the Gulf of Mexico, *Gigantidas childressi* (originally described as "*B.* *childressi*" [17]). We used a correlative imaging approach by combining synchrotron-radiation based micro-computed tomography (SR μ CT), correlative light (LM) and transmission electron microscopy (TEM), and complemented it with fluorescence in situ hybridization (FISH) to analyse the early life stages and compare them to their shallow-water relative *Mytilus edulis* [18]. This approach allowed an integrative analysis of symbiont colonisation and its effects on the host body plan, from the whole animal down to single host and symbiont cells.

Methods

Sampling and fixation

Deep-sea mussels were collected from the sea floor with remotely operated vehicles (Table S1). *M. edulis* were collected in the Baltic Sea at a site close to Kiel, Germany (Table S1).

Samples were preserved for morphological, FISH and TEM analysis (Supplementary Methods a). All specimens were photographed and shell dimensions and shell margin limits were recorded as shown in Figure S1b, c and d (Supplementary Methods b, Table S2).

Sample preparation

For histological analysis, all samples were post-fixed, embedded, serial sectioned and stained with a toluidine blue and sodium tetraborate solution. For TEM, semi-thin sections were re-sectioned according to [19] (Supplementary Methods c). Paraformaldehyde fixed samples were decalcified and DOPE-FISH performed on embedded and sectioned samples using general and specific probes (Table S3, Supplementary Methods d, e) [20]. Details of the light, fluorescence and electron microscopes used are found in Supplementary Methods f.

SR μ CT measurements

SR μ CT datasets were recorded at the DESY using the P05 beamline of PETRA III, operated by the Helmholtz-Zentrum Hereon (Geesthacht, Germany [21]). The x-ray microtomography setup at 15–30 keV and 5X to 40X magnification was used to scan resin-embedded samples with attenuation contrast and uncontrasted samples in PBS-filled capillaries [22] with propagation-based phase contrast. Scan parameters are summarized in Table S4. The tomography data were processed with custom scripts implemented in the ASTRA toolbox [23-25] (Supplementary Methods g).

Correlative workflow for SR μ CT, light and electron microscopy, and FISH

Section series were screened by eye to identify individual host cells and predict symbiont colonisation state based on three morphological characteristics: hypertrophy, loss of microvilli and loss of cilia (Table 1, Supplementary Methods i). The location and symbiont colonization state of the analysed host cells was marked using Cell Counter in Fiji [26]. The analysed sections were re-sectioned and the same fields of view were recorded with TEM

(Figure S2). To validate the LM-based predictions, symbionts were identified in TEM images based on their morphology (Video S5).

Table 1 Overview of number of individuals analysed for each species, developmental stage, and imaging method

species	method	pediveliger	metamorphosis	plantigrade	juvenile	adult
B. puteoserpentis	SR μ CT / μ CT	-	-	-	1	-
	SR μ CT + serial sectioning	1	1	2	1	-
	serial sectioning + LM	2	2	-	-	-
	serial sectioning + TEM	2	2	3	-	-
	FISH	2	1	3	1	-
	total		7	6	8	3
B. azoricus	SR μ CT / μ CT	-	-	-	2	2
	SR μ CT + serial sectioning	-	-	1	-	-
	serial sectioning + LM	1	1	1	-	-
	serial sectioning + TEM	1	-	-	-	-
	total		2	1	2	2
G. childressi	SR μ CT / μ CT	-	-	-	-	1
	SR μ CT + serial sectioning + TEM	1	-	-	-	-
	serial sectioning + LM	-	2	2	-	-
	total		1	2	2	-

Image processing and 3D visualisation

Microscopy images were adjusted and figures composed using Fiji and Adobe Photoshop and Illustrator 2021. LM-images were stitched and aligned with TrackEM2 [27] in Fiji. Amira 2020.2 (ThermoFisher Scientific) was used to generate 3D models from LM and μ CT datasets. Co-registration between μ CT, LM and TEM datasets was carried out after [19] (Supplementary Methods j).

Results

We analysed developmental stages of *B. puteoserpentis*, *B. azoricus* and *G. childressi* ranging from aposymbiotic pediveligers to symbiotic adults, to determine at which stage the symbionts colonize their hosts, and the developmental modifications that these mussels have evolved to adapt to their symbiotic lifestyle (Figure 1, S3 and S4). Because the names for larval stages of bivalves have not always been used consistently, we define them as follows: The earliest life stages in our study were at the last planktonic larval stage - the

pediveliger. Once settled on the seafloor, the animal initiates its metamorphosis from a planktonic to a benthic lifestyle, and we refer to this stage as being in metamorphosis. While metamorphosing, the mussel degrades its velum, the larval feeding and swimming organ, and develops into a plantigrade [28]. During the plantigrade stage, the mussel secretes the adult shell and once the ventral groove of the gills, which transports particles to the labial palps, is formed, it enters the juvenile stage [29]. When the gonads are developed, the mussel is considered an adult [18].

Identification of developmental stages

We measured the shell lengths of 259 specimens: 129 *B. puteoserpentis*, 124 *B. azoricus* and 6 *G. childressi* individuals. We assume that these had already settled or were in the process of settling, as we collected them from mussel beds on the sea floor. The specimens ranged from 370 μm to 4556 μm shell length (Table S2 and S5). The earliest developmental stages were pediveligers, with shell lengths of 366 - 465 μm . Developmental stages could only be identified through detailed analyses of the mussels' soft body anatomy (Figure S1a and b). These analyses revealed that the shell sizes of 58 pediveligers and mussels in metamorphosis overlapped with those of the smallest plantigrade stages (Figure S1 f).

Morphological characterization of *Bathymodiolus* and *Gigantidas* developmental stages

For our morphological analyses, we analysed 39 individuals (Table 1). *B. puteoserpentis* specimens were best preserved and covered the widest range of developmental stages. We therefore focussed our detailed morphological analyses on *B. puteoserpentis*, and compared these with selected *B. azoricus* and *G. childressi* stages (Figure S3 and S4). In the following, we describe the shared morphological features of all three species unless specified otherwise.

The pediveliger larvae were characterized by the presence of a velum, a fully developed digestive system, a foot with two pairs of retractor muscles and two gill baskets (Table S6,

Figure 1 a, S5 a and d and Video S1). The digestive system consisted of the mouth, oral labial palps, oesophagus, stomach, two digestive glands, gastric shield, the style sac with crystalline style, mid gut, s-shaped looped intestine, and anal papillae (Figure 1a and S6). The diverticula of the digestive glands and the epithelia of the stomach contained membrane-bound lipid vesicles (Figure 1, S6 and S7). The gill baskets on each side of the foot (Figure 1a and S6) consisted of three to four single gill filaments in *B. puteoserpentis* and five in *B. azoricus* and *G. childressi* (Figure S3 b and c). These filaments form the descending lamella of the inner demibranch in later life stages. For further details, see Supplementary Note 1.

Figure 1 Three-dimensional visualisation of section series and SRμCT measurements of *B. puteoserpentis* developmental stages based on analyses of four individuals for each stage. Note the different scale bars.

In mussels undergoing metamorphosis, the first steps from a planktonic to a benthic lifestyle were visible in the degradation of the velum (Figure S5) and the appearance of byssus threads (Figure S8). Rearrangements of all organs occurred in this stage, for example, the alignment of the growth axis of the gill ‘basket’ with the length axis of the mussel (Figure 1 a-c, S9 and Video S2). The number of gill filaments increased by one in all species (Figure S6). Furthermore, gill filaments separated from each other, increasing the gaps between

them from 47 μm to 120 μm (Figure 1 c). In the digestive gland and stomach epithelia, the number and volume of lipid vesicles decreased from 12.8% of the soft-body volume in pediveligers, to 3.9% in metamorphosing mussels, 1.5% in plantigrades, and were no longer present in juveniles and adults (Figure 1 a-d and Table S6). The organs involved in filter-feeding also changed. In pediveligers, the main feeding organ was the velum, which collects particles from the seawater and transports them to the oral labial palps and mouth. After the degradation of the velum, particle sorting was taken over by the highly ciliated foot (Figure S5, S8 and S9) in metamorphosing mussels, and by the gills in late plantigrades and juveniles. The plantigrade stage began once the mussels secreted the dissoconch and completed metamorphosis (Figure S1 and S10). As the mussels transitioned from the plantigrade to the juvenile stage, the digestive system straightened (Figure 1c-d, S11 and Video S3 and S4). This morphological change was most prominent in the intestine, which went from a looped to a straight shape and remained straight in all later developmental stages (Figure 1c-d and S11). For further details see Supplementary Note 2.

Establishment of the symbiosis

Central to an accurate assessment of symbiont colonization and symbiont-mediated morphological changes was our correlative approach, which combined SR μ CT, light and electron microscopy (Video S5, Figure S2) and was complemented with FISH. This allowed us to rapidly screen whole animals, yet achieve the resolution needed to identify the colonization of single eukaryotic cells by symbiotic bacteria. We first searched for a morphological characteristic that was visible using light microscopy and reliably revealed the presence of symbionts in host cells. Previous studies [30, 31] showed that in juvenile and adult *Bathymodiolus* mussels, the morphology of epithelial cells colonized by symbionts is fundamentally altered: i) The microvilli that cover all epithelial cells are lost (known as microvillar effacement), and ii) epithelial cells become hypertrophic (swollen) compared to aposymbiotic cells. We identified a third characteristic change in epithelial cells colonized by symbionts that has not received much attention, namely the loss of cilia (Figure 2 c, f and i).

We tested if these three morphological characteristics had predictive power for symbiont colonisation by analysing 1965 epithelial cells from a subset of seven *B. puteoserpentis* individuals (Table 1) and comparing LM images of these cells with their correlated TEM images. Our analyses revealed that all cells predicted to have symbionts in the LM dataset were indeed colonized in the TEM dataset, and likewise, all cells predicted to be aposymbiotic were free of symbionts (Figure S2). Our approach allowed us to identify host cells that were colonised by only a few SOX symbionts based on the absence of microvilli and cilia, indicating that these are lost immediately after the first symbionts colonise host cells (Figure 2 e). We next used our verified morphological characters to reveal the onset of symbiont colonisation in *B. puteoserpentis* in this subset of seven mussels. All pediveliger cells were free of symbionts ($n = 797$ host cells in 2 pediveliger). In the metamorphosing mussels examined with our correlative approach 1 – 15% ($n = 488$ host cells in 2 metamorphosing mussels) and in plantigrades 21 - 26% ($n = 680$ host cells in 3 plantigrades) of all analysed gill, mantle, foot and retractor muscle epithelia cells were colonized by symbionts.

Figure 2 Symbionts first colonize *Bathymodiolus puteoserpentis* during metamorphosis.. TEM micrographs of gills (a–c), mantle (d–f) and foot (g–i) epithelial tissues of pediveligers (a, d and g), metamorphosing mussels (b, e and h) and plantigrades (c, f and i). SOX (yellow) and MOX symbionts (magenta) are highlighted with colour overlays for visibility reasons (a–i). All epithelial tissues of pediveligers were aposymbiotic (a, d and g). Colonisation by the SOX and MOX symbionts was first observed during metamorphosis (b, e and h). In plantigrades, all epithelial tissues were colonized by both symbiont types (c, f and i). Dashed boxes indicate regions in which symbionts were in the process of colonizing epithelial tissue (shown magnified in j – m). ci, cilia; mi, microvilli; MOX, methane-

oxidizing symbiont; nu, nucleus; phl, phagolysosome; SOX, sulphur-oxidizing symbiont. For raw image data see Supplement Figure S14.

We then expanded our analyses to LM on another 10 *B. puteoserpentis* individuals (Table 1). These analyses confirmed our results from the correlative dataset: All additional pediveligers (n=3) and metamorphosing mussels were aposymbiotic (n=3), all additional plantigrades (n=2) and juveniles (n=2) were colonized by symbionts. Finally, we performed FISH on another seven *B. puteoserpentis* individuals (Table 1). These analyses corroborated our LM data on the timing of symbiont colonisation, with all pediveliger aposymbiotic and symbiont colonization beginning at metamorphosis (Figure 3, S12 and S13).

Our correlative workflow revealed that pediveliger were aposymbiotic in all three mussel species (Figure 2 a, d and g and S15 a and c). Interestingly, two of the aposymbiotic *B. puteoserpentis* pediveligers had bacterial morphotypes similar to the SOX and MOX symbionts attached to the outside of their shell (Figure S16). Mussels that were undergoing metamorphosis were the earliest developmental stage in which we found symbionts in all three host species, with a shell length of 432 μm in the smallest *B. puteoserpentis* individual, 510 μm in *B. azoricus*, and 383 μm in *G. childressi*. In these metamorphosing mussels, we observed symbionts in epithelial cells of the gill filaments, mantle, foot and retractor muscle (Figure 2 b, e, h and j–m and S15). Once symbiont colonisation began, gill tissue morphology was similar to that of adult mussels [30-32]: the majority of gill cells were symbiont-containing bacteriocytes without microvilli or cilia, whereas the only gill cells without symbionts were those at the ventral ends of the gill filaments and at the frontal-to-lateral zones along the length of the filaments, as well as the intercalary cells (Figure S6, S9 and S10). In addition to the gills, the epithelial tissues of the mantle, foot and retractor muscles also had symbionts in all three host species (Figure 2 c, f and i and S15). We never observed other bacteria besides the two symbionts in any of the developmental stages, including the intranuclear parasite that infects these mussels, based on FISH analyses with symbiont-specific and eubacterial probes of *B. puteoserpentis*, and TEM analyses of

symbiont morphology of *B. puteoserpentis*, *B. azoricus* and *G. childressi* specimen (Figure 2, 3 S12, S13 and S15).

The superior preservation of *B. puteoserpentis* specimens allowed us to analyse the process of symbiont colonisation in this species in more detail. We first found evidence of symbiont colonisation in metamorphosing mussels that had only a few bacteria in gill, mantle and foot epithelial (Figure 2 b, e, h, j-m). Bacterial density per gill cell was the lowest in metamorphosing mussels with 13.7% (± 6.3) of the host cell area occupied by symbionts, and steadily increased in later developmental stages, reaching up to 29.0% (± 5.0) in plantigrades and 32.1% (± 6.1) in adult mussels (Table S7). With the onset of symbiont colonisation, we observed phagolysosomal digestion of the symbionts in all epithelial cells, even those with only very few symbionts (Figure 2 b – c and e – f). The process of symbiont colonisation appeared to be extremely rapid. Nearly all metamorphosing mussels had either no symbionts at all, or all of their epithelial tissues were colonized. In only two out of six individuals, we occasionally observed SOX and MOX symbionts that were not completely engulfed by the host's apical cell membrane, which we interpreted as on-going colonisation (Figure 2 j – m). These first steps in colonisation were particularly common in mantle epithelial cells, while in the same specimen the gill epithelial cells were already fully colonized (Figure 2b, e and f). Furthermore, in host cells where colonisation was ongoing, we observed that these were colonized only by SOX (Figure 2 e), or by both SOX and MOX (Figure 2 h), but we never observed host cells only colonized by MOX symbionts.

Figure 3 SOX and MOX symbionts colonize all epithelial tissues in *Bathymodiolus puteoserpentis* plantigrades. False-coloured FISH images show probes specific for SOX (yellow, BMARt-193) and MOX symbionts (magenta, BMARm-845) and host nuclei stained with DAPI (cyan). Sagittal cross sections of two individuals (**a** and **b**) show SOX and MOX symbionts in the gill, foot and mantle epithelia. Schematic drawings of the anatomy are provided in the top right corners. To visualize host tissue autofluorescence is shown in grey in **a** and **b**. Dashed boxes indicate magnified regions of the colonized gill, mantle and foot region shown in **c–h**. ft, foot; m, mantle; gf, gill filament; p.a.m, posterior adductor muscle; st, stomach.

Discussion

Post-metamorphosis development in *Bathymodiolus* and *Gigantidas* deviates from the mytilid blueprint

Our study shows that the use of shell characteristics alone to determine developmental stages of deep-sea mussels is not reliable (e.g. [16]), particularly for shell lengths of settling pediveligers and early plantigrades. Our analyses of shell lengths in developmental stages of the three mussel species revealed an overlap in size of 50 μm between pediveligers and plantigrades. This inconsistency in shell lengths between developmental stages indicates that metamorphosis begin is not dependent on size, and provides further evidence for the ability of mussels to delay metamorphosis while continuing to grow, as previously suggested for *G. childressi* [33]. Such a delay could be due to a lack of settlement cues or limited nutrition, similar to what is known from *M. edulis* [34]. Delaying metamorphosis would favour dispersal, potentially leading to an increase in geographic distribution and the colonisation of new and remote habitats [35].

The pre-metamorphosis development of *Bathymodiolus* and *Gigantidas* mussels is similar to that of their close relatives from the genus *Idas* [7, 15] and shallow water mytilids such as *M. edulis* [18, 29, 36]. The pediveligers of *Bathymodiolus*, *Gigantidas*, *Idas* and *Mytilus* have a large velum, foot, mantle epithelium, digestive system, central nerve system and two preliminary gill baskets consisting of three to five gill buds [7, 15, 18]. During metamorphosis, the velum is degraded, organs within the mantle cavity are rearranged and the lipid vesicles in the digestive diverticula and stomach are reduced. In early developmental stages of mussels from the *Mytilidae*, lipid vesicles serve as storage compounds to fuel the energy-demanding process of metamorphosis [18] and they likely have a similar function in *Bathymodiolus* and *Gigantidas*. Furthermore, these lipid vesicles could provide energy for movement of the pediveligers during their searches for sites to settle [37, 38].

Although early development appears to be conserved across *Bathymodiolus*, *Gigantidas*, *Idas* and their shallow water relatives, marked differences occur as soon as symbiont colonisation begins in the deep-sea mytilids. All colonised epithelial cells lost not only their microvilli but also their cilia, and developed a hypertrophic habitus, as previously shown for gill bacteriocytes [7, 30, 31]. If the symbionts of *Bathymodiolus* and *Gigantidas* actively induce these cellular changes or if these are a response by the host to symbiont colonisation remains unresolved, but our data shows that these processes were tightly linked spatially and temporally. Furthermore, our correlative analyses demonstrate that these cell surface modifications serve as reliable markers for the state of symbiont colonisation. Observations of effacement of cilia and microvilli have been reported for a wide range of bacteria that invade epithelia, particularly pathogens, and for these it also remains to be shown if the bacteria or the host drive these processes [39-41].

Although it has been known for several decades that the symbionts of *Bathymodiolus* and *Gigantidas* mussels supply their host with nutrition, and that adults possess only a rudimentary gut, nothing was known about the development of the digestive system in these mussels. We observed a straightening of the digestive system after completion of metamorphosis in all three species (Figure 4). The stomach and the intestine straightened and the digestive system changed from the complex looped type found in *Mytilus* to the straight type seen in adults of bathymodioline mussels (Table S8). This transformation is striking, as in mytilids like *M. edulis*, such drastic morphological changes after metamorphosis are not known [42]. In *Bathymodiolus* and *Gigantidas*, the straightening of the digestive system did not coincide with the first stages of symbiont colonisation or metamorphosis, but rather occurred during the transition from the plantigrade to the juvenile stage, well after metamorphosis and only when these hosts had become fully colonized by their symbionts (Figure 4).

In general, the morphology of an animal's gastrointestinal tract reflects its food sources. Animals that digest complex foods possess enlarged compartments and lengthened

gastrointestinal structures to slow down the flow of digested material and increase the breakdown of complex molecules [43]. We hypothesize that the straightening of the digestive system in the vent and seep mussels we analysed here was induced by their shift from filter feeding to gaining nutrition from their symbionts. The straightening of the digestive tract could be an evolutionary adaptation to minimise the energy needed for maintaining the digestive tract once the majority of nutrition is gained through intracellular digestion of symbionts in the bacteriocytes.

Figure 4 Summary of symbiont colonisation and development of *Bathymodiolus* and *Gigantidas* mussels. Pediveliger are aposymbiotic (a), and symbiont colonisation begins in metamorphosing mussels (b). By the plantigrade (c) and juvenile (d) stage, all epithelial tissues are fully colonized by symbionts. The digestive system is reduced between the plantigrade and juvenile stages, changing from a looped to a straight morphology. (e) Schematic of the hypothetical life cycle of bathymodioline mussels indicating the aposymbiotic pelagic developmental stages and the symbiotic benthic developmental stages. ft, foot; gf, gill filament; in, intestine; l.i, lipid inclusion; oe, oesophagus.

Not all bathymodiolins have a straightened digestive system. *Idas* intestines are complex with one or more loops [7, 15, 44]. These small mussels are commonly found at organic falls, which typically have higher inputs of organic matter than vents and seeps. Also, symbiont abundances are lower in *Idas* than in the large vent and seep bathymodiolins,

indicating that *Idas* mussels may depend more on filter feeding than their larger relatives. Intriguingly, the relatively large adults of the bathymodioline genus *Vulcanidas* have a pronounced looped intestine despite high symbiont abundances [45]. *Vulcanidas* inhabits relatively shallow hydrothermal vents close to the photic zone (140 m compared to 1000–2500 m depth for the mussels in our study) where more organic material from the photic zone is available for their nutrition. It is thus likely that filter feeding plays a greater role in the nutrition of *Idas* and *Vulcanidas* than in the three species analysed in this study. These findings raise the question whether the straightening of the digestive tract is a conserved developmental trait in only some genera, or if the environment and availability of organic matter and other non-symbiotic food sources drives the morphology of the digestive tract in bathymodioline mussels.

Symbiont colonisation begins during the plantigrade stage as soon as the velum is degraded

As recently highlighted, how and when *Bathymodiolus* and *Gigantidas* mussels acquire their symbionts has remained, as yet, unclear [9]. Here we used correlative imaging analyses to reveal that the early developmental stages of two *Bathymodiolus* and one *Gigantidas* species were aposymbiotic, and narrowed the window of symbiont acquisition to mussels undergoing metamorphosis (Figure 4). Our analyses revealed that the symbionts colonize the gills only after the velum is lost. As long as the velum is present and active, particles from the surrounding seawater are either transported directly into the digestive system or expelled from the mussel. In *M. edulis*, once the velum is degraded the gill filaments separate further from each other and take over the task of sorting food particles and generating a water current [18, 29]. Beginning in the late plantigrade stage of *Bathymodiolus*, we identified a similar increase of space between the gill filaments. Given that the developmental changes in the morphology of the bathymodiolin gills are so similar to those of *M. edulis* [7, 15, 18, 29], it is likely that the functional roles of the gills in *Bathymodiolus* change in a similar manner as in *M. edulis*: The pediveliger gills of

Bathymodiolus are used for respiration only, and then become responsible for water current generation in late metamorphosing mussels, and the sorting of particles in plantigrades and juveniles. We hypothesize that this functional shift of the gills plays an important role in enabling the symbionts to adhere to the gill epithelium and initiate symbiont colonisation.

The initial colonisation of the host by their symbionts appears to be rapid, based on our observation that 22 out of 24 *B. puteoserpentis* individuals were either completely aposymbiotic or fully colonized, and the first stages of symbiont colonisation were only visible in two metamorphosing mussels. As we were working with preserved samples that represent a snapshot of development, the chance of observing a process depends on how often it occurs and how long it takes. The less frequent or the faster a process happens, the smaller the chance of observing it. We therefore conclude that symbiont colonisation occurs rapidly in *B. puteoserpentis*, given the small percentage of individuals in which we observed the first steps of colonisation. Symbiont colonization seems to be even more rapid than in *I. simpsoni* and *I. modiolaeformis* [7, 15]. This could reflect the importance for *Bathymodiolus* mussels to quickly acquire symbionts once they have nearly consumed their internal energy reserves and settled in an environment that lacks energy-rich planktonic nutrition [46].

We found symbionts in epithelial cells of the gills, mantle, foot, and retractor muscle in the plantigrades and juveniles of all three bathymodioline species similar to previous studies [16, 47, 48]. Previous work suggested that the symbionts first colonize the mantle epithelia, and from there colonize gill cells, as the first gill filaments are formed from mantle tissues [47].

Our data contradicts this assumption, as the gills had already begun to develop in pediveligers before the onset of symbiont colonisation. Furthermore, in two *B. puteoserpentis* specimen we detected fully colonized gills, while the mantle tissue was still in the process of being colonized. Our findings indicate that symbionts first colonize gill cells before colonizing other epithelial tissues, and that the SOX symbionts colonize individual host cells first, before the MOX.

Our analyses revealed that *B. puteoserpentis*, *B. azoricus* and *G. childressi* larvae are aposymbiotic during their planktonic phase and first acquire their symbionts when they transition to a benthic lifestyle. The timing of symbiont acquisition and major developmental changes was similar in all three species, and, with the exception of the digestive system, corresponds to observations of early life stages of *Idas* [7, 9, 15]. These similarities in developmental biology and symbiont acquisition suggest that these traits are conserved in bathymodiolins. The acquisition of symbionts after settlement allows these mussels to recruit locally adapted symbionts. As the geochemistry of vent and seep environments varies strongly across spatial scales [49], recruiting locally adapted symbiont populations would confer a strong fitness advantage to these hosts. Indeed, recent studies have revealed that *Bathymodiolus* mussels host multiple strains of symbionts that vary in key functions, such as the use of energy and nutrient sources, electron acceptors and viral defence mechanisms [50, 51]. By acquiring their symbionts from the sites where they settle, bathymodioline mussels can establish symbioses with those strains that are best adapted to the local environment.

Conclusion and outlook

Our correlative imaging workflow revealed the intricate developmental processes from the subcellular to the whole animal scale that are thought to be triggered when deep-sea mussels acquire their symbionts. Furthermore, we identified the narrow window in which symbiont acquisition begins and showed the morphological changes of the digestive system following symbiont uptake. Given that we never observed bacterial morpho- or phylotypes other than the known SOX and MOX symbionts, even in the earliest larval life stages, strong recognition mechanisms must ensure this high specificity. Now that we have identified when and how symbiont colonisation occurs in *Bathymodiolus* and *Gigantidas*, a spatial and temporal transcriptomic approach could shed light on the underlying molecular mechanisms of symbiont recognition, acquisition and maintenance, and further our understanding of the entwined dialogue between animal hosts and their microbial symbionts.

Data accessibility

LM-data, μ CT-data and supplementary videos are available on figshare, see Table S1 for DOIs.

Authors' contributions

M.F., **N.L.** and **N.D.** conceived this study. **M.F.** and **N.L.** wrote the manuscript, with support from **N.D.**, and contributions and revisions from all other co-authors. **M.F.** performed the light and fluorescence microscopy, analysed the light, TEM, fluorescence and μ CT datasets, reconstructed the 3D models, designed the figures, and produced the videos. **B.G.** helped with the 3D reconstructions and μ CT-measurements. **J.U.H.** performed the μ CT measurements with help from **B.G.** and **M.F.** and reconstructed the μ CT-datasets. **N.L.** did the TEM re-sectioning and measurements.

Competing interests

We declare no competing interests.

Funding

Funding was provided by the Max Planck Society, the MARUM Cluster of Excellence 'The Ocean Floor' (Deutsche Forschungsgemeinschaft (German Research Foundation) under Germany's Excellence Strategy - EXC-2077 – 39074603), a Gordon and Betty Moore Foundation Marine Microbial Initiative Investigator Award (grant no. GBMF3811 to N.D.) and a European Research Council Advanced Grant (BathyBiome, Grant 340535 to N.D.). μ CT measurements were performed at the DESY under the proposal IDs: 20170337 and 20180295.

Acknowledgements

We thank the captains, crew members and ROV pilots of the cruises M126, M82-3 and NA58. We are grateful to Christian Borowski and Stéphane Hourdez for their valuable contributions to collecting mussel larvae and Wiebke Ruschmeier for her help in the lab. We thank all involved in supporting us at the DESY at the P05 beamline of PETRA III (Helmholtz-Zentrum Hereon, Geesthacht, Germany). We also thank Benjamin Cooper (Max-Planck Institute for Experimental Medicine, Göttingen) for preliminary sample preparation,

and Bernhard Ruthensteiner (Zoologische Staatssammlung München) and Frank Melzner (GEOMAR, Kiel) for fruitful discussions.

References

- [1] McFall-Ngai, M., Hadfield, M.G., Bosch, T.C.G., Carey, H.V., Domazet-Lošo, T., Douglas, A.E., Dubilier, N., Eberl, G., Fukami, T., Gilbert, S.F., et al. 2013 Animals in a bacterial world, a new imperative for the life sciences. *Proc. Natl. Acad. Sci. USA* **110**, 3229-3236. (doi:10.1073/pnas.1218525110).
- [2] Dubilier, N., Bergin, C. & Lott, C. 2008 Symbiotic diversity in marine animals: the art of harnessing chemosynthesis. *Nat. Rev. Microbiol.* **6**, 725-740. (doi:10.1038/nrmicro1992).
- [3] DeChaine, E. & Cavanaugh, C.M. 2005 Symbioses of methanotrophs and deep-sea mussels (*Mytilidae: Bathymodiolinae*). *Prog Mol Subcell Biol.* **41**, 227-249. (doi:10.1007/3-540-28221-1_11).
- [4] Bright, M. & Bulgheresi, S. 2010 A complex journey: transmission of microbial symbionts. *Nat. Rev. Microbiol.* **8**, 218-230. (doi:10.1038/nrmicro2262).
- [5] Endow, K. & Ohta, S. 1990 Occurrence of bacteria in the primary oocytes of vesicomid clam *Calptogena soyoae*. *Mar. Ecol. Prog. Ser.* **64**. (doi:10.3354/meps064309).
- [6] Montgomery, M.K. & McFall-Ngai, M. 1994 Bacterial symbionts induce host organ morphogenesis during early postembryonic development of the squid *Euprymna scolopes*. *Development* **120**, 1719-1729.
- [7] Laming, S.R., Duperron, S., Cunha, M.R. & Gaudron, S.M. 2014 Settled, symbiotic, then sexually mature: adaptive developmental anatomy in the deep-sea, chemosymbiotic mussel *Idas modiolaeformis*. *Mar. Biol.* **161**, 1319-1333. (doi:10.1007/s00227-014-2421-y).
- [8] Nussbaumer, A.D., Fisher, C.R. & Bright, M. 2006 Horizontal endosymbiont transmission in hydrothermal vent tubeworms. *Nature* **441**, 345-348. (doi:10.1038/nature04793).
- [9] Laming, S.R., Gaudron, S.M. & Duperron, S. 2018 Lifecycle Ecology of Deep-Sea Chemosymbiotic Mussels: A Review. *Front. Mar. Sci.* **5**. (doi:10.3389/fmars.2018.00282).
- [10] Fontanez, K.M. & Cavanaugh, C.M. 2014 Evidence for horizontal transmission from multilocus phylogeny of deep-sea mussel (*Mytilidae*) symbionts. *Environ. Microbiol.* **16**, 3608-3621. (doi:10.1111/1462-2920.12379).
- [11] Won, Y.J., Hallam, S.J., O'Mullan, G.D., Pan, I.L., Buck, K.R. & Vrijenhoek, R.C. 2003 Environmental Acquisition of Thiotrophic Endosymbionts by Deep-Sea Mussels of the Genus *Bathymodiolus*. *Appl. Environ. Microbiol.* **69**, 6785-6792. (doi:10.1128/aem.69.11.6785-6792.2003).
- [12] Won, Y.J., Jones, W.J. & Vrijenhoek, R.C. 2008 Absence of cospeciation between deep-sea mytilids and their thiotrophic endosymbionts. *J. Shellfish Res.* **27**, 129-138. (doi:10.2983/0730-8000(2008)27[129:AOCBDM]2.0.CO;2).
- [13] Russell, S.L., Pepper-Tunick, E., Svedberg, J., Byrne, A., Ruelas Castillo, J., Vollmers, C., Beinart, R.A. & Corbett-Detig, R. 2020 Horizontal transmission and recombination maintain forever young bacterial symbiont genomes. *PLoS Genet.* **16**, e1008935. (doi:10.1371/journal.pgen.1008935).
- [14] Gaudron, S.M., Demoyencourt, E. & Duperron, S. 2012 Reproductive Traits of the Cold-Seep Symbiotic Mussel *Idas modiolaeformis*: Gametogenesis and Larval Biology. *Biol. Bull.* **222**, 6-16. (doi:10.1086/bblv222n1p6).
- [15] Laming, S.R., Duperron, S., Gaudron, S.M., Hilario, A. & Cunha, M.R. 2015 Adapted to change: The rapid development of symbiosis in newly settled, fast-maturing chemosymbiotic mussels in the deep sea. *Mar. Environ. Res.* **112**, 100-112. (doi:10.1016/j.marenvres.2015.07.014).
- [16] Salerno, J.L., Macko, S.A., Hallam, S.J., Bright, M., Won, Y.J., McKiness, Z. & Van Dover, C.L. 2005 Characterization of symbiont populations in life-history stages of mussels from chemosynthetic environments. *Biol. Bull.* **208**, 145-155. (doi:10.2307/3593123).

- [17] Gustafson, R.G., Turner, R.D., Lutz, R.A. & Vrijenhoek, R.C. 1998 A new genus and five new species of mussels (Bivalvia, Mytilidae) from deep-sea sulfide/hydrocarbon seeps in the Gulf of Mexico. *Malacologia* **40**, 63-112.
- [18] Bayne, B.L. 1971 Some morphological changes that occur at the metamorphosis of the larvae of *Mytilus edulis*. In *The Fourth European Marine Biology Symposium* (pp. 259-280).
- [19] Handschuh, S., Baeumler, N., Schwaha, T. & Ruthensteiner, B. 2013 A correlative approach for combining microCT light and transmission electron microscopy in a single 3D scenario. *Front. Zool.* **10**. (doi:10.1186/1742-9994-10-44).
- [20] Stoecker, K., Dorninger, C., Daims, H. & Wagner, M. 2010 Double labeling of oligonucleotide probes for fluorescence in situ hybridization (DOPE-FISH) improves signal intensity and increases rRNA accessibility. *Appl. Environ. Microbiol.* **76**, 922-926. (doi:10.1128/AEM.02456-09).
- [21] Wilde, F., Ogurreck, M., Greving, I., Hammel, J.U., Beckmann, F., Hipp, A., Lottermoser, L., Khokhriakov, I., Lytaev, P., Dose, T., et al. 2016 Micro-CT at the imaging beamline P05 at PETRA III. *AIP Conference Proceedings* **1741**, 030035. (doi:10.1063/1.4952858).
- [22] Geier, B., Franke, M., Ruthensteiner, B., Porras, M.Á.G., Gruhl, A., Wörmer, L., Moosmann, J., Hammel, J.U., Dubilier, N., Leisch, N., et al. 2019 Correlative 3D anatomy and spatial chemistry in animal-microbe symbioses: developing sample preparation for phase-contrast synchrotron radiation based micro-computed tomography and mass spectrometry imaging. In *SPIE Optical Engineering + Applications* (eds. B. Müller & G. Wang), SPIE.
- [23] Moosmann, J., Ershov, A., Weinhardt, V., Baumbach, T., Prasad, M.S., LaBonne, C., Xiao, X., Kashef, J. & Hofmann, R. 2014 Time-lapse X-ray phase-contrast microtomography for in vivo imaging and analysis of morphogenesis. *Nat. Protoc.* **9**, 294. (doi:10.1038/nprot.2014.033).
- [24] van Aarle, W., Palenstijn, W.J., De Beenhouwer, J., Altantzis, T., Bals, S., Batenburg, K.J. & Sijbers, J. 2015 The ASTRA Toolbox: A platform for advanced algorithm development in electron tomography. *Ultramicroscopy* **157**, 35-47. (doi:10.1016/j.ultramic.2015.05.002).
- [25] van Aarle, W., Palenstijn, W.J., Cant, J., Janssens, E., Bleichrodt, F., Dabrovolski, A., De Beenhouwer, J., Joost Batenburg, K. & Sijbers, J. 2016 Fast and flexible X-ray tomography using the ASTRA toolbox. *Opt. Express* **24**, 25129-25147. (doi:10.1364/oe.24.025129).
- [26] Schindelin, J., Arganda-Carreras, I., Frise, E., Kaynig, V., Longair, M., Pietzsch, T., Preibisch, S., Rueden, C., Saalfeld, S., Schmid, B., et al. 2012 Fiji: an open-source platform for biological-image analysis. *Nat. Methods* **9**, 676-682. (doi:10.1038/nmeth.2019).
- [27] Cardona, A., Saalfeld, S., Schindelin, J., Arganda-Carreras, I., Preibisch, S., Longair, M., Tomancak, P., Hartenstein, V. & Douglas, R.J. 2012 TrakEM2 Software for Neural Circuit Reconstruction. *PloS one* **7**, e38011. (doi:10.1371/journal.pone.0038011).
- [28] Baker, P. & Mann, R. 1997 The postlarval phase of bivalve mollusks: a review of functional ecology and new records of postlarval drifting of Chesapeake Bay bivalves. *Bull. Mar. Sci.* **61**, 409-430.
- [29] Cannuel, R., Beninger, P.G., Mc Combie, H. & Boudry, P. 2009 Gill Development and Its Functional and Evolutionary Implications in the Blue Mussel *Mytilus edulis*. *Biol. Bull.* **217**, 173-188. (doi:10.1086/BBLv217n2p173).
- [30] Fisher, C.R., Childress, J.J., Oremland, R.S. & Bidigare, R.R. 1987 The importance of methane and thiosulfate in the metabolism of the bacterial symbionts of two deep-sea mussels. *Mar. Biol.* **96**, 59-71. (doi:10.1007/BF00394838).
- [31] Wentrup, C., Wendeberg, A., Schimak, M., Borowski, C. & Dubilier, N. 2014 Forever competent: deep-sea bivalves are colonized by their chemosynthetic symbionts throughout their lifetime. *Environ. Microbiol.* **16**, 3699-3713. (doi:10.1111/1462-2920.12597).
- [32] Fiala-Medioni, A. & Le Pennec, M. 1987 Trophic structural adaptations in relation to the bacterial association of bivalve molluscs from hydrothermal vents and subduction zones. In *Symposium on marine symbioses. 1 (1987)* (pp. 63-74, Balaban, Philadelphia, .
- [33] Arellano, S.M. & Young, C.M. 2009 Spawning, development, and the duration of larval life in a deep-sea cold-seep mussel. *Biol. Bull.* **216**, 149-162. (doi:10.1086/BBLv216n2p149).

- [34] Martel, A., Tremblay, R., Toupoint, N., Olivier, F. & Myrand, B. 2014 Veliger Size at Metamorphosis and Temporal Variability in Prodissoconch II Morphometry in the Blue Mussel (Mytilus edulis): Potential Impact on Recruitment. *J. Shellfish Res.* **33**, 443-455. (doi:10.2983/035.033.0213).
- [35] Young, C.M., He, R., Emllet, R.B., Li, Y., Qian, H., Arellano, S.M., Van Gaest, A., Bennett, K.C., Wolf, M., Smart, T.I., et al. 2012 Dispersal of deep-sea larvae from the intra-American seas: simulations of trajectories using ocean models. *Integr. Comp. Biol.* **52**, 483-496. (doi:10.1093/icb/ics090).
- [36] Bayne, B.L. 1965 Growth and the delay of metamorphosis of the larvae of *Mytilus edulis* (L.). *Ophelia* **2**, 1-47. (doi:10.1080/00785326.1965.10409596).
- [37] Breusing, C., Biastoch, A., Drews, A., Metaxas, A., Jollivet, D., Vrijenhoek, Robert C., Bayer, T., Melzner, F., Sayavedra, L., Petersen, Jillian M., et al. 2016 Biophysical and Population Genetic Models Predict the Presence of “Phantom” Stepping Stones Connecting Mid-Atlantic Ridge Vent Ecosystems. *Curr. Biol.* **26**, 2257-2267. (doi:10.1016/j.cub.2016.06.062).
- [38] Distel, D.L., Baco, A.R., Chuang, E., Morrill, W., Cavanaugh, C. & Smith, C.R. 2000 Do mussels take wooden steps to deep-sea vents? *Nature* **403**, 725-726. (doi:10.1038/35001667).
- [39] Kaper, J.B., Nataro, J.P. & Mobley, H.L.T. 2004 Pathogenic *Escherichia coli*. *Nat. Rev. Microbiol.* **2**, 123-140. (doi:10.1038/nrmicro818).
- [40] Quarmby, L.M. 2004 Cellular Deflagellation. In *Int. Rev. Cytol.* (pp. 47-91, Academic Press).
- [41] Tubiash, H.S., Chanley, P.E. & Leifson, E. 1965 Bacillary Necrosis, a Disease of Larval and Juvenile Bivalve Mollusks I. Etiology and Epizootiology. *J. Bacteriol.* **90**, 1036.
- [42] Eggermont, M., Cornillie, P., Dierick, M., Adriaens, D., Nevejan, N., Bossier, P., Van den Broeck, W., Sorgeloos, P., Defoirdt, T. & Declercq, A.M. 2020 The blue mussel inside: 3D visualization and description of the vascular-related anatomy of *Mytilus edulis* to unravel hemolymph extraction. *Sci. Rep.* **10**, 6773. (doi:10.1038/s41598-020-62933-9).
- [43] Karasov, W.H. & Douglas, A.E. 2013 Comparative digestive physiology. *Compr Physiol* **3**, 741-783. (doi:10.1002/cphy.c110054).
- [44] Thubaut, J., Corbari, L., Gros, O., Duperron, S., Couloux, A. & Samadi, S. 2013 Integrative Biology of *Idas iwaoakii* (Habe, 1958), a ‘Model Species’ Associated with Sunken Organic Substrates. *PLOS ONE* **8**, e69680. (doi:10.1371/journal.pone.0069680).
- [45] Von Cosel, R. & Marshall, B.A. 2010 A new genus and species of large mussel (Mollusca: Bivalvia: *Mytilidae*) from the Kermadec Ridge. *Records of the Musuem of New Zealand Te Papa* **21**, 15.
- [46] Page, H., Fisher, C. & Childress, J. 1990 Role of filter-feeding in the nutritional biology of a deep-sea mussel with methanotrophic symbionts. *Mar. Biol.* **104**, 251-257. (doi:10.1007/BF01313266).
- [47] Streams, M.E., Fisher, C.R. & Fiala-Médioni, A. 1997 Methanotrophic symbiont location and fate of carbon incorporated from methane in a hydrocarbon seep mussel. *Mar. Biol.* **129**, 465-476. (doi:10.1007/s002270050187).
- [48] Wentrup, C., Wendeberg, A., Huang, J.Y., Borowski, C. & Dubilier, N. 2013 Shift from widespread symbiont infection of host tissues to specific colonization of gills in juvenile deep-sea mussels. *ISME J* **7**, 1244-1247. (doi:10.1038/ismej.2013.5).
- [49] Desbruyères, D., Almeida, A., Biscoito, M., Comtet, T., Khripounoff, A., Le Bris, N., Sarradin, P.M. & Segonzac, M. 2000 A review of the distribution of hydrothermal vent communities along the northern Mid-Atlantic Ridge: dispersal vs. environmental controls. In *Island, Ocean and Deep-Sea Biology* (eds. M.B. Jones, J.M.N. Azevedo, A.I. Neto, A.C. Costa & A.M.F. Martins), pp. 201-216. Dordrecht, Springer Netherlands.
- [50] Ansorge, R., Romano, S., Sayavedra, L., Porras, M.Á.G., Kupczok, A., Tegetmeyer, H.E., Dubilier, N. & Petersen, J. 2019 Functional diversity enables multiple symbiont strains to coexist in deep-sea mussels. *Nat. Microbiol* **4**, 2487-2497. (doi:10.1038/s41564-019-0572-9).

[51] Romero Picazo, D., Dagan, T., Ansorge, R., Petersen, J.M., Dubilier, N. & Kupczok, A. 2019 Horizontally transmitted symbiont populations in deep-sea mussels are genetically isolated. *ISME J* **13**, 2954-2968. (doi:10.1038/s41396-019-0475-z).